# Variational Deep Learning via Implicit Regularization

**Jonathan Wenger**[†]
Columbia University

**Beau Coker**[†]
Columbia University

**Juraj Marusic**
Columbia University

**John P. Cunningham**
Columbia University

## Abstract

Modern deep learning models generalize remarkably well in-distribution, despite being overparametrized and trained with little to no *explicit* regularization. Instead, current theory credits *implicit* regularization imposed by the choice of architecture, hyperparameters, and optimization procedure. However, deep neural networks can be surprisingly non-robust, resulting in overconfident predictions and poor out-of-distribution generalization. Bayesian deep learning addresses this via model averaging, but typically requires significant computational resources as well as carefully elicited priors to avoid overriding the benefits of implicit regularization. Instead, in this work, we propose to regularize variational neural networks solely by relying on the *implicit bias of (stochastic) gradient descent*. We theoretically characterize this inductive bias in overparametrized linear models as generalized variational inference and demonstrate the importance of the choice of parametrization. Empirically, our approach demonstrates strong in- and out-of-distribution performance without additional hyperparameter tuning and with minimal computational overhead.

## 1 Introduction

The success of deep learning across many application domains is, on the surface, remarkable, given that deep neural networks are usually overparameterized and trained with little to no *explicit* regularization. The generalization properties observed in practice have been explained by *implicit* regularization instead, resulting from the choice of architecture [1], hyperparameters [2, 3], and optimizer [4–10]. Notably, the corresponding inductive biases often require no additional computation, in contrast to enforcing a desired inductive bias through explicit regularization.

In the last two decades, there has been an increasing focus on improving the reliability and robustness of deep learning models via (approximately) Bayesian approaches [11] to improve performance on out-of-distribution data [12], in continual learning [13], and in sequential decision-making [14]. However, despite its promise, in practice, Bayesian deep learning can suffer from issues with prior elicitation [15], can be challenging to scale [16], and explicit regularization via a prior combined with approximate inference may result in pathological inductive biases and uncertainty [17–20].

In this work, we demonstrate both theoretically and empirically how to exploit the implicit bias of optimization for approximate inference in probabilistic neural networks, thus regularizing training implicitly rather than explicitly via the prior. This not only narrows the gap to how standard neural networks are trained, but also reduces the computational overhead of training compared to variational inference. More specifically, we propose to learn a variational distribution over the weights of a deep neural network by maximizing the *expected* log-likelihood in analogy to training via maximum likelihood in the standard case. However, in contrast to variational Bayes, there is *no explicit regularization* via a Kullback-Leibler divergence to the prior. Surprisingly, we show theoretically and empirically that training this way does not cause uncertainty to collapse away from the training data, if initialized and parametrized correctly. More so, for overparametrized linear models we rigorously characterize the implicit bias of SGD as generalized variational inference with a 2-Wasserstein regularizer penalizing deviations from the prior. Figure 1 illustrates our approach on a toy example.

---

[†]Equal contribution.

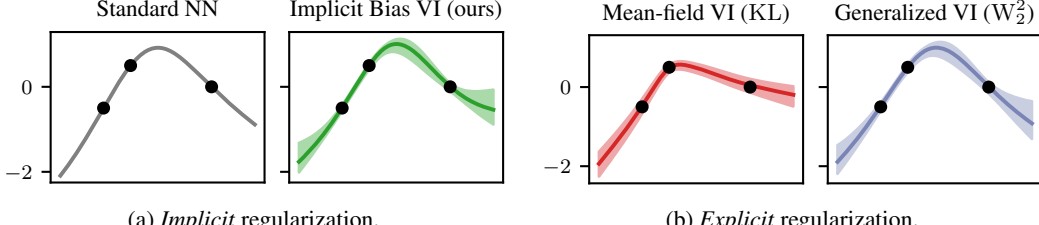

(a) *Implicit* regularization.        (b) *Explicit* regularization.

Figure 1: *Variational deep learning via implicit regularization.* Neural networks generalize well without explicit regularization due to implicit regularization from the architecture and optimization. We can exploit this implicit bias for variational deep learning, removing the computational overhead of explicit regularization and narrowing the gap to deep learning practice. As illustrated for a two-hidden layer MLP and proven rigorously for overparametrized linear models in Theorems 1 and 2, the implicit bias of (S)GD in variational networks (see (a)) can be understood as generalized variational inference with a 2-Wasserstein regularizer (see (b)). This differs from the standard ELBO objective with a KL divergence to the prior as used for example in mean-field VI (see (b)).

**Contributions**    In this work, we propose a new approach to Bayesian deep learning that generalizes robustly by exploiting the implicit regularization of (stochastic) gradient descent. We fully characterize this implicit bias for regression (Theorem 1) and binary classification (Theorem 2) in overparameterized linear models, generalizing results for non-probabilistic models and drawing a rigorous connection to generalized Bayesian inference. We also demonstrate the importance of the parametrization for the inductive bias and its impact on hyperparameter choice. In several benchmarks, we demonstrate competitive performance to state-of-the-art baselines for Bayesian deep learning, at minimal computational overhead compared to standard neural networks. Finally, we provide an open-source implementation of our approach as a standalone library: `inferno`.

## 2    BACKGROUND

Given a training dataset $(\boldsymbol{X}, \boldsymbol{y}) = \{(\boldsymbol{x}_n, y_n)\}_{n=1}^{N}$ of input-output pairs, supervised learning seeks a function $f_{\boldsymbol{w}}(\boldsymbol{x})$ to predict the corresponding output $y(\boldsymbol{x})$ for a test input $\boldsymbol{x}$. The parameters $\boldsymbol{w} \in \mathbb{R}^{P}$ of the function are typically trained via empirical risk minimization, i.e.

$$\boldsymbol{w}_{\star} \in \arg\min_{\boldsymbol{w}} \ell_r(\boldsymbol{w}) \qquad \text{with} \qquad \ell_r(\boldsymbol{w}) = \ell(\boldsymbol{y}, f_{\boldsymbol{w}}(\boldsymbol{X})) + \lambda r(\boldsymbol{w}), \tag{1}$$

where the loss $\ell(\boldsymbol{y}, f_{\boldsymbol{w}}(\boldsymbol{X}))$ encourages fitting the training data and the regularizer $r(\boldsymbol{w})$, given some $\lambda > 0$, discourages overfitting, which can lead to poor generalization on test data.

**Implicit Bias of Optimization**    One remarkable observation in deep learning is that training over-parametrized neural networks ($P > N$) with gradient descent without *explicit* regularization can nonetheless lead to effective (in-distribution) generalization, despite there being many global minima of the loss corresponding to functions $f_{\boldsymbol{w}}$ which achieve zero training error [21]. This can be explained by the optimizer, initialization, and parametrization *implicitly* regularizing the optimization problem $\arg\min_{\boldsymbol{w}} \ell(\boldsymbol{y}, f_{\boldsymbol{w}}(\boldsymbol{X}))$, thereby preferring certain global minima [e.g. 4, 5, 7, 22, 23]. Nonetheless, deep neural networks can be surprisingly brittle when predicting *out-of-distribution*, often displaying overconfidence and a significant drop in generalization performance.

**Bayesian Deep Learning**    Approximate Bayesian techniques like the Laplace approximation [24–26], stochastic weight averaging [27, 28], deep ensembles [29], and variational approaches [30–33] attempt to address the aforementioned shortcomings of deep learning by learning a distribution over functions as opposed to merely a point estimate. The idea being that a weighted combination of models, all of which achieve low training error, generalizes more robustly while at the same time providing uncertainty quantification.

**Variational Inference**    In Bayesian inference this weighted combination is defined by the posterior distribution $p(\boldsymbol{w} \mid \boldsymbol{X}, \boldsymbol{y}) \propto p(\boldsymbol{y} \mid \boldsymbol{X}, \boldsymbol{w})p(\boldsymbol{w})$ over weights, induced by a likelihood $p(\boldsymbol{y} \mid \boldsymbol{w})$ and

a choice of prior $p(\boldsymbol{w})$ that expresses an explicit preference for some models over others. Approximating the posterior with $q_{\boldsymbol{\theta}}(\boldsymbol{w}) \approx p(\boldsymbol{w} \mid \boldsymbol{X}, \boldsymbol{y})$ by maximizing a lower bound to the log-evidence leads to the following variational optimization problem [34]:

$$\boldsymbol{\theta}_{\star} \in \arg\min_{\boldsymbol{\theta}} \ell_r(\boldsymbol{\theta}) \qquad \text{s.t.} \qquad \ell_r(\boldsymbol{\theta}) = \mathbb{E}_{q_{\boldsymbol{\theta}}(\boldsymbol{w})}(-\log p(\boldsymbol{y} \mid \boldsymbol{w})) + \mathrm{KL}(q_{\boldsymbol{\theta}}(\boldsymbol{w}) \parallel p(\boldsymbol{w})). \qquad (2)$$

Equation (2) is an instance of the empirical risk minimization objective in Equation (1), with the key difference that one optimizes over variational parameters $\boldsymbol{\theta}$ of a family of distributions $q_{\boldsymbol{\theta}}(\boldsymbol{w}) \in Q$. If that family includes the posterior, $q_{\boldsymbol{\theta}}(\boldsymbol{w}) = p(\boldsymbol{w} \mid \boldsymbol{X}, \boldsymbol{y})$ is the unique global minimum. In the case of a potentially misspecified prior or likelihood, the variational formulation (2) can be generalized to arbitrary loss functions $\ell$ and statistical distances D to the prior [35–37], such that

$$\ell_r(\boldsymbol{\theta}) = \mathbb{E}_{q_{\boldsymbol{\theta}}(\boldsymbol{w})}(\ell(\boldsymbol{y}, f_{\boldsymbol{w}}(\boldsymbol{X}))) + \lambda \, \mathrm{D}(q_{\boldsymbol{\theta}}, p). \qquad (3)$$

## 3 Variational Deep Learning via Implicit Regularization

Our overarching goal is to enable deep neural networks to generalize robustly out-of-distribution without sacrificing their in-distribution performance, at minimal computational overhead. We approach this goal within the framework of Bayesian deep learning, by learning a distribution over neural networks $f_{\boldsymbol{w}}$, induced by a parametrized variational distribution $q_{\boldsymbol{\theta}}(\boldsymbol{w})$ over its weights. However, rather than approximating the Bayesian posterior, which trades off training error against an explicit, a priori preference for certain models, we enforce that *all models have zero training error* while using *implicit* regularization to weight them. Doing so preserves the implicit regularization of the optimizer, which determines the generalization performance of neural networks to a substantial degree, rather than purely relying on explicit regularization induced by the prior. Importantly, this approach leads to robust out-of-distribution generalization, while providing uncertainty quantification at small computational overhead over standard deep learning.

### 3.1 Training via the Expected Loss

We propose to train a variational neural network defined by an architecture $f_{\boldsymbol{w}}$ and a variational distribution over weights $q_{\boldsymbol{\theta}}(\boldsymbol{w})$ by *minimizing the expected loss* $\bar{\ell}(\boldsymbol{\theta})$ in analogy to how deep neural networks are usually trained. In other words, the optimal variational parameters are given by

$$\boldsymbol{\theta}_{\star} \in \arg\min_{\boldsymbol{\theta}} \underbrace{\mathbb{E}_{q_{\boldsymbol{\theta}}(\boldsymbol{w})}(\ell(\boldsymbol{y}, f_{\boldsymbol{w}}(\boldsymbol{X})))}_{:=\bar{\ell}(\boldsymbol{\theta})} + \lambda \mathrm{D}(q_{\theta}, p). \qquad (4)$$

At first glance removing the divergence term from the variational objective in Eq. (3) seems problematic because the new objective is clearly minimized when the variational distribution is a point mass at the minimum loss solution, i.e. $q_{\boldsymbol{\theta}_{\star}}(\boldsymbol{w}) = \delta_{\boldsymbol{w}_{\star}}(\boldsymbol{w})$ where $\boldsymbol{w}_{\star} \in \arg\min_{\boldsymbol{w}} \ell(\boldsymbol{y}, f_{\boldsymbol{w}}(\boldsymbol{X}))$. This seemingly defeats the point of a Bayesian deep learning framework, given that there is no variability in predictions on test data. Moreover, the new objective no longer involves a prior distribution, ostensibly removing the ability to manually favor some models over others entirely. The key to understanding our approach is that, in the overparameterized setting, a point mass is only one of many optima corresponding to distributions $q_{\boldsymbol{\theta}_{\star}}(\boldsymbol{w})$, and it is the implicit bias of the optimization procedure that chooses among them. As we will see, if one trains an overparametrized linear model via the expected loss using (stochastic) gradient descent, this implicit bias can be explicitly characterized to depend on the initialization.

### 3.2 Implicit Bias of SGD as Generalized Variational Inference

Assume we train an overparametrized linear model with a Gaussian variational family via the expected loss. For an appropriate learning rate sequence, (stochastic) gradient descent converges to a global minimum $\boldsymbol{\theta}_{\star}^{\mathrm{GD}} \in \arg\min \bar{\ell}(\boldsymbol{\theta})$ of the training objective. As we show in Section 4, if SGD is *initialized to the prior*, i.e. $q_{\boldsymbol{\theta}_0}(\boldsymbol{w}) = p(\boldsymbol{w})$, its implicit bias can be understood as selecting the distribution over models with zero training error that is closest to the prior in 2-Wasserstein distance:

$$q_{\boldsymbol{\theta}_{\star}^{\mathrm{GD}}} = \arg\min_{\substack{q_{\boldsymbol{\theta}} \\ \text{s.t. } \boldsymbol{\theta} \in \arg\min \bar{\ell}(\boldsymbol{\theta})}} \mathrm{W}_2^2(q_{\boldsymbol{\theta}}, p).$$

Therefore, we can interpret the implicit bias of (S)GD when training a variational linear model as performing *generalized variational inference*. More precisely, the above is equivalent to $q_{\theta^{\text{GD}}_*}$ minimizing the objective in Equation (3) for a certain regularization strength, but with a regularizer that is *not* a KL divergence as it would be for standard variational inference, but rather a 2-Wasserstein distance to the prior. This characterization directly generalizes results for (non-probabilistic) models, where the implicit bias of SGD selects minima that are close to the initialization in Euclidean distance [5, 21]. We therefore call our method Implicit Bias Variational Inference (IBVI). From a practical perspective, by exploiting the implicit regularization of SGD, rather than performing generalized variational inference directly, we no longer need to compute the regularizer explicitly or allocate memory for the prior hyperparameters.[1]

Section 4 provides a detailed version of the regression result introduced here and proves a similar result for binary classification. Our experiments in Section 5 focus on the application to deep neural networks, where we generally expect the implicit regularization to be more complex.

### 3.3 COMPUTATIONAL EFFICIENCY

In practice, we minibatch the expected loss both over training data and parameter samples $w_m$ drawn from the variational distribution $q_\theta(w)$ such that

$$\bar{\ell}(\theta) = \mathbb{E}_{q_\theta(w)}(\ell(y, f_w(X))) \approx \frac{1}{N_b M} \sum_{n=1}^{N_b} \sum_{m=1}^{M} \ell(y_n, f_{w_m}(x_n)). \tag{5}$$

The training cost is primarily determined by two factors. The number of parameter samples $M$ we draw for each evaluation of the objective, and the variational family, which determines the number of additional parameters of the model and the cost for sampling a set of parameters in each forward pass. We wish to keep the overhead compared to a vanilla deep neural network as small as possible.

**Training With A Single Parameter Sample ($M = 1$)** When drawing fewer parameter samples $w_m$ the training objective in Eq. (5) becomes noisier similar to using a smaller batch size. This is concerning since the optimization procedure may not converge given this additional noise. However, one can *train with a single parameter sample only*, simply by reducing the learning rate appropriately, as we show experimentally in Figure 2 and Section S3.2. Therefore, given a set of sampled parameters, the cost of a forward and backward pass is identical to a standard neural network (up to the overhead of the covariance parameters). When using fewer parameter samples in the expected loss, training is unstable unless the learning rate is chosen sufficiently small. For a fixed number of optimizer steps this decreases performance, but either training for more steps, or using momentum closes this gap.

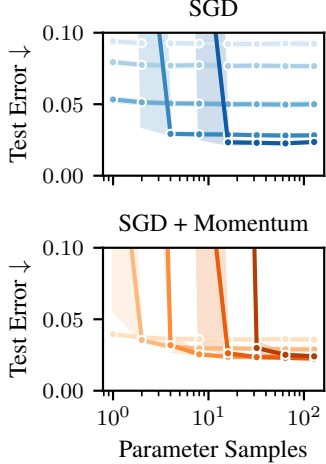

Figure 2: *Training with a single parameter sample given a small enough learning rate.* Lighter color shades correspond to smaller learning rates. See also Section S3.2.

**Variational Family and Covariance Structure** We choose a Gaussian variational distribution $q_\theta(w)$ over (a subset of the) weights of the neural network. While at first glance this may seem restrictive, there is ample evidence that variational families in deep neural networks do not need to be complex to be expressive [38, 39]. In fact, in analogy to deep feedforward NNs with ReLU activations being universal approximators [40], one can show that Bayesian neural networks with ReLU activations and at least one Gaussian hidden layer are universal conditional distribution approximators, meaning they can approximate any continuous conditional distribution arbitrarily well [39]. As we show in Section 4, training an overparametrized linear model with SGD via the expected loss amounts to generalized variational inference *if the covariance is factorized*, i.e. $\Sigma = SS^\top$ where $S \in \mathbb{R}^{P \times R}$ is a dense matrix with rank $R \leq P$. The implicit bias of SGD for arbitrary parametrizations of the covariance matrix remains an open problem. Throughout our experiments we use Gaussian layers with factorized covariances for all architectures.

---

[1]We only need them to initialize the optimizer after which we can free up the allocated memory.

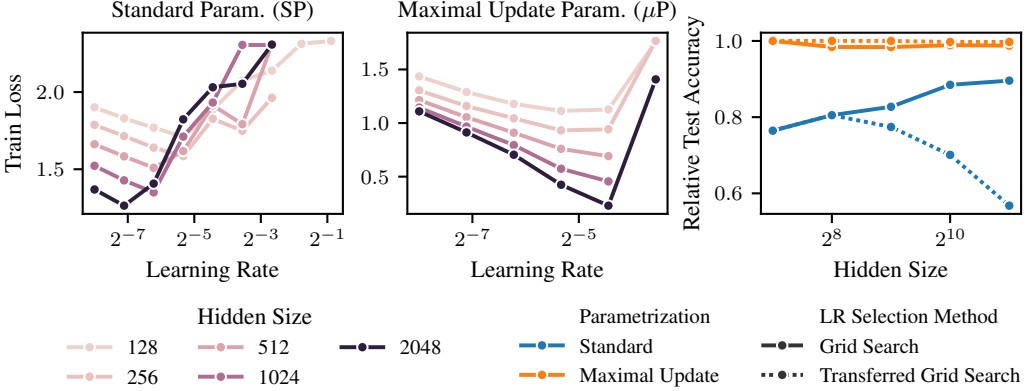

Figure 3: *Hyperparameter Transfer.* When scaling the size of a neural network, one has to re-tune the hyperparameters, such as the learning rate, when using the standard parametrization (SP). The same is true for probabilistic networks as we show here on CIFAR-10 (left). However, when using our proposed extension of the maximal update parametrization ($\mu$P) [41] to probabilistic networks, one can tune the learning rate on a small model and achieve optimal generalization for larger models by "transferring" the optimal learning rate from a smaller model (center and right).

### 3.4 Parametrization, Feature Learning and Hyperparameter Transfer

The inductive bias of SGD depends on the initialization and choice of *parametrization*, a bijective map $\rho : \Theta' \to \Theta$ reparametrizing a (variational) model such that $f_{\boldsymbol{\theta}} \equiv f_{\rho(\boldsymbol{\theta}')}$. When training deep neural networks, it is not unusual to use layer-specific learning rates. These can be absorbed into the weights of the model and the initialization, meaning they effectively just define a different parametrization [Lemma J.1, 41]. While parameterization is well-studied for non-probabilistic deep learning, it has been identified as one of the "grand challenges of Bayesian computation" [42].

In deep learning, the "standard parameterization" (SP) initializes the weights of a neural network randomly from a distribution with variance $\propto 1/\texttt{fan\_in}$ (e.g., as in Kaiming initialization, the PyTorch default) and makes no further adjustments to the forward pass or learning rate. In contrast, the maximal update parametrization ($\mu$P) [43] ensures feature learning even as the width of the network tends to infinity. In addition, under $\mu$P, hyperparameters like the learning rate, can be tuned on a small model and transferred to a large-scale model [41].

Given our interpretation of training via the expected loss as generalized variational inference with a prior implied by the parametrization and initialization, a natural question is whether we can extend $\mu$P to the variational setting and thus inherit its inductive bias. In the probabilistic setting, feature learning occurs when the *distribution* over hidden units changes from initialization. At any point during training, the $i$th hidden unit in layer $l$ is a function of four random variables: the variational mean and covariance parameters $(\boldsymbol{\mu}, \boldsymbol{S})$, Gaussian noise $\boldsymbol{z}$, and the previous layer hidden units:

$$\boldsymbol{h}_i^{(l)}(\boldsymbol{x}) = \boldsymbol{W}_i \boldsymbol{h}^{(l-1)}(\boldsymbol{x}) = (\boldsymbol{\mu}_i + \boldsymbol{S}_i \boldsymbol{z}) \boldsymbol{h}^{(l-1)}(\boldsymbol{x}). \tag{6}$$

The parameters are random because of the stochasticity in the initialization and/or optimization procedure, while the noise is randomly drawn during each forward pass. Since the $\boldsymbol{S}_i \boldsymbol{z}$ term is a sum over $R$ terms, where $R$ is the rank of $\boldsymbol{S} \in \mathbb{R}^{P \times R}$, applying the central limit theorem we propose scaling this term by $R^{-1/2}$ and then applying $\mu$P to the mean and covariance parameters. In practice, we implement the scaling via an adjustment to the covariance initialization and learning rate. Section S2 in the supplement provides empirical investigating of this scaling, demonstrating feature learning in the last hidden layer as the width is increased.

Figure 3 demonstrates that our proposed maximal update parametrization enables hyperparameter transfer in a probabilistic model. We train two-hidden-layer MLPs on CIFAR10, using a low rank covariance in the final two layers. Under standard parametrization (left panel), the learning rate that results in the smallest training loss decreases with hidden size. In contrast, under $\mu$P (middle panel), it remains the same across hidden sizes. The right panel of Fig. 3 demonstrates the practical

implications for model selection. For each parametrization and each hidden size $D$, we select the learning rate based on a grid search. In "transferred grid search" we do a grid search using the smallest model (hidden size 128) and transfer the best validating learning rate to the hidden size $D$ model, whereas in "grid search" we perform the grid search on the hidden size $D$ model. Relative to the test accuracy of the best performing model across learning rate and parametrization, we see that (a) $\mu$P outperforms SP, though the gap decreases with hidden size, and (b) the transfer strategy works well for $\mu$P but poorly for SP once the hidden size exceeds 256.

## 3.5 RELATED WORK

Variational inference in the context of Bayesian deep learning has seen rapid development in recent years [30–33, 44, 45]. Using a Wasserstein regularizer [37] in the context of generalized VI [36] is arguably most related to our work, given our theoretical results. Structure in the variational parameters has always played an important role for computational reasons [38, 46, 47] and often only a few layers are treated probabilistically [39], with some methods only considering the last layer, effectively treating the neural network as a feature extractor [48, 49]. The Laplace approximation if applied in the last-layer also falls under this category, which has the advantage that it can be applied post-hoc [13, 24–26, 50–54]. Deep ensembles repeat the standard training process using multiple random initializations [29, 55] and have been linked to Bayesian methods [56, 57] with certain caveats [58, 59]. While we use SGD only to optimize the variational parameters and arguably average over samples by using momentum, SGD has also been used widely to directly approximate samples from a target distribution [27, 56, 60, 61], a popular example being stochastic weight averaging (SWA) [27, 28]. Our theoretical analysis extends recent developments on the implicit bias of overparameterized linear models [4, 5, 7] to the probabilistic setting. For classification, works have focused on convergence rates [6], SGD [7], SGD with momentum [8], and the multiclass setting [10]. Results on the implicit bias of neural network training [22] often assume large widths [9, 62–65] allowing similar arguments as for linear models. The former is exemplified by the neural tangent parametrization, under which neural networks behave like kernel methods in the infinite width limit [66]. Yang et al. [41, 43, 64, 65] developed an alternative parameterization that still admits feature learning in the infinite width limit, which we extended to the case of variational networks.

## 4 THEORETICAL ANALYSIS

Consider an overparameterized linear model with a Gaussian prior, trained via the expected loss using (stochastic) gradient descent. We show that, in both regression (Theorem 1) and binary classification (Theorem 2), our approach can be understood as generalized variational inference with a 2-Wasserstein regularizer, which penalizes deviation from the prior among models with zero training error. Theorems 1 and 2 recover analogous results for non-probabilistic models [4, 5, 21].

### 4.1 LINEAR REGRESSION

**Theorem 1** (Implicit Bias in Regression)
*Let $f_{\boldsymbol{w}}(\boldsymbol{x}) = \boldsymbol{x}^{\mathsf{T}}\boldsymbol{w}$ be an overparametrized linear model with $P > N$. Define a Gaussian prior $p(\boldsymbol{w}) = \mathcal{N}\big(\boldsymbol{w}; \boldsymbol{\mu}_0, \boldsymbol{S}_0 \boldsymbol{S}_0^{\mathsf{T}}\big)$ and likelihood $p(\boldsymbol{y} \mid \boldsymbol{w}) = \mathcal{N}\big(\boldsymbol{y}; f_{\boldsymbol{w}}(\boldsymbol{X}), \sigma^2 \boldsymbol{I}\big)$ and assume a variational family $q_{\boldsymbol{\theta}}(\boldsymbol{w}) = \mathcal{N}\big(\boldsymbol{w}; \boldsymbol{\mu}, \boldsymbol{S}\boldsymbol{S}^{\mathsf{T}}\big)$ with $\boldsymbol{\theta} = (\boldsymbol{\mu}, \boldsymbol{S})$ such that $\boldsymbol{\mu} \in \mathbb{R}^P$ and $\boldsymbol{S} \in \mathbb{R}^{P \times R}$ where $R \leq P$. If the learning rate sequence $(\eta_t)_t$ is chosen such that the limit point $\boldsymbol{\theta}_\star^{\mathrm{GD}} = \lim_{t \to \infty} \boldsymbol{\theta}_t^{\mathrm{GD}}$ identified by gradient descent, initialized at $\boldsymbol{\theta}_0 = (\boldsymbol{\mu}_0, \boldsymbol{S}_0)$, is a (global) minimizer of the expected log-likelihood $\bar{\ell}(\boldsymbol{\theta})$, then*

$$\boldsymbol{\theta}_\star^{\mathrm{GD}} \in \underset{\substack{\boldsymbol{\theta} = (\boldsymbol{\mu}, \boldsymbol{S}) \\ s.t.\ \boldsymbol{\theta} \in \arg\min \bar{\ell}(\boldsymbol{\theta})}}{\arg\min} \mathrm{W}_2^2(q_{\boldsymbol{\theta}}, p). \tag{7}$$

*Further, this also holds in the case of stochastic gradient descent and when using momentum.*

*Proof.* See Section S1.1.1. □

Theorem 1 states that, among those variational parameters which minimize the expected loss, SGD (with momentum) converges to the unique variational distribution which is closest in 2-Wasserstein

distance to the prior. This characterization of the implicit regularization of SGD as generalized variational inference differs from a standard ELBO objective (2) in VI via the choice of regularizer. Since the variational parameters minimize the expected loss in Equation (7), all samples from the predictive distribution interpolate the training data (see Figure 1(b), right panel), the same way a standard neural network would. In contrast, when training with a KL regularizer, the uncertainty does not collapse at the training data (see Figure 1(b), left panel). In fact, a KL regularizer would diverge to infinity for a Gaussian with vanishing variance. Now, for test points that are increasingly out-of-distribution, i.e. less aligned with the span of the training data, the variational predictive matches the prior predictive more closely. Interestingly, $q_{\boldsymbol{\theta}^{\mathrm{GD}}_\star}$ is equal to the distribution over weights of an ensemble of linear models initialized from the prior and trained independently (see Section S1.1.3). Next, we prove a similar result for binary classification.

## 4.2 BINARY CLASSIFICATION OF LINEARLY SEPARABLE DATA

Consider a binary classification problem with labels $y_n \in \{-1, 1\}$, a linear model $f_{\boldsymbol{w}}(\boldsymbol{x}) = \boldsymbol{x}^\mathsf{T}\boldsymbol{w}$, and a variational distribution $q_{\boldsymbol{\theta}}(\boldsymbol{w})$ with variational parameters $\boldsymbol{\theta}$. The expected empirical loss is $\bar{\ell}(\boldsymbol{\theta}) = \sum_{n \in [N]} \mathbb{E}_{q_{\boldsymbol{\theta}}(\boldsymbol{w})}\big(\ell(y_n \boldsymbol{x}_n^\mathsf{T}\boldsymbol{w})\big)$. We assume without loss of generality[2] that all labels are positive, i.e. $y_n = 1$ for all $n$, and that the dataset is linearly separable.

**Assumption 1** The dataset is *linearly separable*: $\exists \boldsymbol{w} \in \mathbb{R}^P$ such that $\forall n : \boldsymbol{w}^\mathsf{T}\boldsymbol{x}_n > 0$.

For an overparametrized linear model, if $\boldsymbol{X} \in \mathbb{R}^{N \times P}$ has full row rank the dataset is guaranteed to be linearly separable.[3] Define the solution to the hard margin SVM, the $L_2$ *max margin vector* as

$$\hat{\boldsymbol{w}} = \operatorname*{arg\,min}_{\boldsymbol{w} \in \mathbb{R}^P} \|\boldsymbol{w}\|_2^2 \quad \text{s.t.} \quad \boldsymbol{w}^\mathsf{T}\boldsymbol{x}_n \geq 1, \tag{8}$$

and the set of *support vectors* $\mathcal{S} = \operatorname*{arg\,min}_{n \in [N]} \boldsymbol{x}_n^\mathsf{T}\hat{\boldsymbol{w}}$ indexing the data points on the margin. We make the following additional assumption which is satisfied with high probability under mild assumptions on the training data distribution and degree of overparametrization [67, 68].

**Assumption 2** The SVM support vectors span the dataset: $\operatorname{span}(\{\boldsymbol{x}_n\}_{n \in [N]}) = \operatorname{span}(\{\boldsymbol{x}_n\}_{n \in \mathcal{S}})$.

We can now characterize the implicit bias in the case of binary classification.

**Theorem 2** (Implicit Bias in Binary Classification)
*Let $f_{\boldsymbol{w}}(\boldsymbol{x}) = \boldsymbol{x}^\mathsf{T}\boldsymbol{w}$ be an (overparametrized) linear model and define a Gaussian prior $p(\boldsymbol{w}) = \mathcal{N}\big(\boldsymbol{w}; \boldsymbol{\mu}_0, \boldsymbol{S}_0\boldsymbol{S}_0^\mathsf{T}\big)$. Assume a variational distribution $q_{\boldsymbol{\theta}}(\boldsymbol{w}) = \mathcal{N}\big(\boldsymbol{w}; \boldsymbol{\mu}, \boldsymbol{S}\boldsymbol{S}^\mathsf{T}\big)$ over the weights $\boldsymbol{w} \in \mathbb{R}^P$ with variational parameters $\boldsymbol{\theta} = (\boldsymbol{\mu}, \boldsymbol{S})$ such that $\boldsymbol{S} \in \mathbb{R}^{P \times R}$ and $R \leq P$. Assume we are using the exponential loss $\ell(u) = \exp(-u)$ and optimize the expected empirical loss $\bar{\ell}(\boldsymbol{\theta})$ via gradient descent initialized at the prior, i.e. $\boldsymbol{\theta}_0 = (\boldsymbol{\mu}_0, \boldsymbol{S}_0)$, with a sufficiently small learning rate $\eta$. Then for almost any dataset which is linearly separable (Assumption 1) and for which the support vectors span the data (Assumption 2), the rescaled gradient descent iterates (rGD)*

$$\boldsymbol{\theta}_t^{\mathrm{rGD}} = (\boldsymbol{\mu}_t^{\mathrm{rGD}}, \boldsymbol{S}_t^{\mathrm{rGD}}) = \left(\tfrac{1}{\log(t)}\boldsymbol{\mu}_t^{\mathrm{GD}} + \boldsymbol{P}_{\mathrm{null}(\boldsymbol{X})}\boldsymbol{\mu}_0, \boldsymbol{S}_t^{\mathrm{GD}}\right) \tag{9}$$

*converge to a limit point $\boldsymbol{\theta}_\star^{\mathrm{rGD}} = \lim_{t \to \infty} \boldsymbol{\theta}_t^{\mathrm{rGD}}$ for which it holds that*

$$\boldsymbol{\theta}_\star^{\mathrm{rGD}} \in \operatorname*{arg\,min}_{\substack{\boldsymbol{\theta} = (\boldsymbol{\mu}, \boldsymbol{S}) \\ \text{s.t. } \boldsymbol{\theta} \in \Theta_\star}} \mathrm{W}_2^2(q_{\boldsymbol{\theta}}, p), \tag{10}$$

*where the feasible set $\Theta_\star = \{(\boldsymbol{\mu}, \boldsymbol{S}) \mid \boldsymbol{P}_{\mathrm{range}(\boldsymbol{X}^\mathsf{T})}\boldsymbol{\mu} = \hat{\boldsymbol{w}} \quad \text{and} \quad \forall n : \mathrm{Var}_{q_{\boldsymbol{\theta}}}(f_{\boldsymbol{w}}(\boldsymbol{x}_n)) = 0\}$ consists of mean parameters which, if projected onto the training data, are equivalent to the $L_2$ max margin vector and covariance parameters such that there is no uncertainty at training data.*

*Proof.* See Section S1.2. □

Theorem 2 states that the mean parameters $\boldsymbol{\mu}_t$ converge to the $L_2$ max-margin vector $\hat{\boldsymbol{w}}$ in the span of the training data, i.e. the data manifold, and there uncertainty collapses to zero. This is analogous

---

[2] This is not a restriction since we can always absorb the sign into the inputs, such that $\boldsymbol{x}'_n := y_n \boldsymbol{x}_n$.
[3] We can always choose $\boldsymbol{w} = \boldsymbol{X}^\mathsf{T}(\boldsymbol{X}\boldsymbol{X}^\mathsf{T})^{-1}\boldsymbol{1}$, i.e. the weights linearly interpolating $\boldsymbol{y} = \boldsymbol{1} = (1, \ldots, 1)^\mathsf{T}$.

to the regression case, where zero training loss enforces interpolation of the training data. In the null space of the training data, i.e. off of the data manifold, the model falls back on the prior as enforced by the 2-Wasserstein distance. The assumption of an exponential loss is standard in the literature and we expect this to extend to (binary) cross-entropy in the same way it does in results for standard neural networks [4, 6–8, 10]. Similarly, we conjecture that Theorem 2 can be extended to SGD with momentum [cf. 7, 8]. While Theorem 2 is similar to Theorem 1, there are some subtle differences. First, the feasible set for the minimization problem in Equation (10) is not the set of minima of the expected loss. This is because the exponential function does not have an optimum in contrast to a quadratic function. However, the sequence of variational parameters identified by gradient descent still satisfies $\lim_{t\to\infty} \bar{\ell}(\boldsymbol{\theta}_t) = 0$. Second, without transformation of the mean parameters, the exponential loss results in the mean parameters being unbounded. This necessitates the transformation in Equation (9) as we explain in detail in Section S1.3.

## 5 EXPERIMENTS

We benchmark the *generalization* and *robustness* of our approach, Implicit Bias VI (IBVI), against standard neural networks and several baselines for uncertainty quantification, namely Temperature Scaling (TS) [69], Laplace approximation (LA-GS) & (LA-ML) [24–26], Weight-Space VI (WSVI) [30, 31], SWA-Gaussian (SWAG) [28], and Deep Ensembles (DE) [29], on a set of standard benchmark datasets for image classification and robustness to input corruptions. We use convolutional architectures (LeNet5 [70] or ResNet34 [71]), which, for all datasets but MNIST, are initialized with pretrained weights except for the input and output layer. All models are trained with SGD with momentum $\gamma = 0.9$ and a batch size of $N_b = 128$ for 200 epochs in single precision on an NVIDIA GH200 GPU. Results shown are averaged across five random seeds. A detailed description of the datasets, metrics, models and training can be found in Section S3. An implementation of our method can be found at:

https://github.com/inferno-ml/inferno

**In-Distribution Generalization and Uncertainty Quantification** In order to assess the in-distribution generalization, we measure the test error, negative log-likelihood (NLL), and calibration error (ECE) on MNIST, CIFAR10, CIFAR100 and TinyImageNet. As Figure 4 shows for CIFAR100, and Figure S10 for all datasets, the test error for post-hoc methods (TS, LA-GS, LA-ML) is unchanged. As expected, SWAG and IBVI perform similarly with only Ensembles providing an increase in accuracy, but at substantial memory overhead compared to most other approaches. Similarity of IBVI to Ensembles is perhaps expected in light of their equivalence for linear models (see Proposition S1). In-distribution uncertainty quantification measured in terms of NLL is improved substantially by TS, DE, and IBVI, with only LA and WSVI showing occasional worsening of NLL compared to the base model. The full results in Figure S10 show that TS, DE, and IBVI consistently are also the best calibrated. As described in Section 3.3, for IBVI we train with a single sample only and a probabilistic input and output layer with low-rank covariance, reducing the memory overhead compared to a standard neural network to as little as $\approx 10\%$ with similar training time (see Figure 4). See Section S3.3.2 for the full experimental results including different parametrizations (SP vs $\mu$P).

**Robustness to Input Corruptions** We evaluate the robustness of the different models on MNISTC [72], CIFAR10C, CIFAR100C, and TinyImageNetC [73]. These are corrupted versions of the original datasets, where the images are modified via a set of 15 corruptions, such as impulse noise, blur, pixelation, etc. We selected the maximum severity for each corruption and averaged the performance across all. As expected, the performance of all models drops compared to the in-distribution performance measured on the standard test sets as Figure 5 shows. Besides DE which consistently show lower test error, also IBVI shows improved accuracy on corrupted data compared to all other approaches. When using the maximal update parametrization, SWAG shows good accuracy on the two larger datasets (see Figure S12). TS, DE, and IBVI perform consistently well in terms of uncertainty quantification (both for NLL and ECE) across all datasets, with LA-ML being somewhat competitive in terms of NLL. However, compared to the in-distribution setting IBVI has better uncertainty quantification than Ensembles across all datasets.

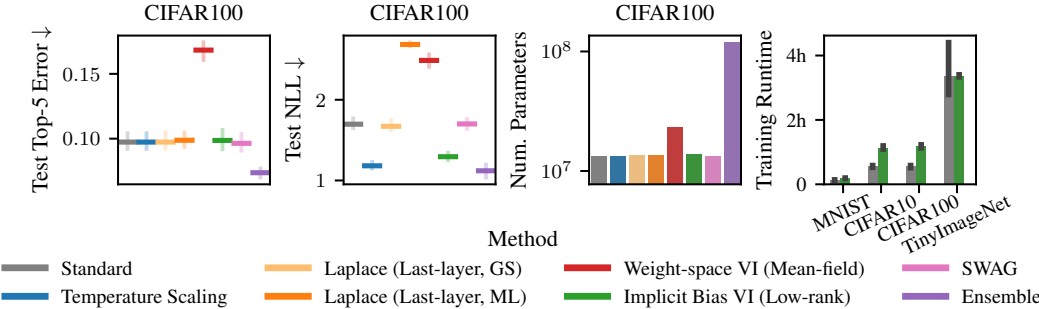

Figure 4: *In-distribution generalization and uncertainty quantification.* Implicit Bias VI (IBVI) has similar test error to other Bayesian deep learning approaches and achieves competitive uncertainty quantification on in-distribution data. While ensembles have improved accuracy, they come at an additional memory overhead. Training a probabilistic model via IBVI has only a minor computational overhead during training, both in time and memory, over standard deep learning.

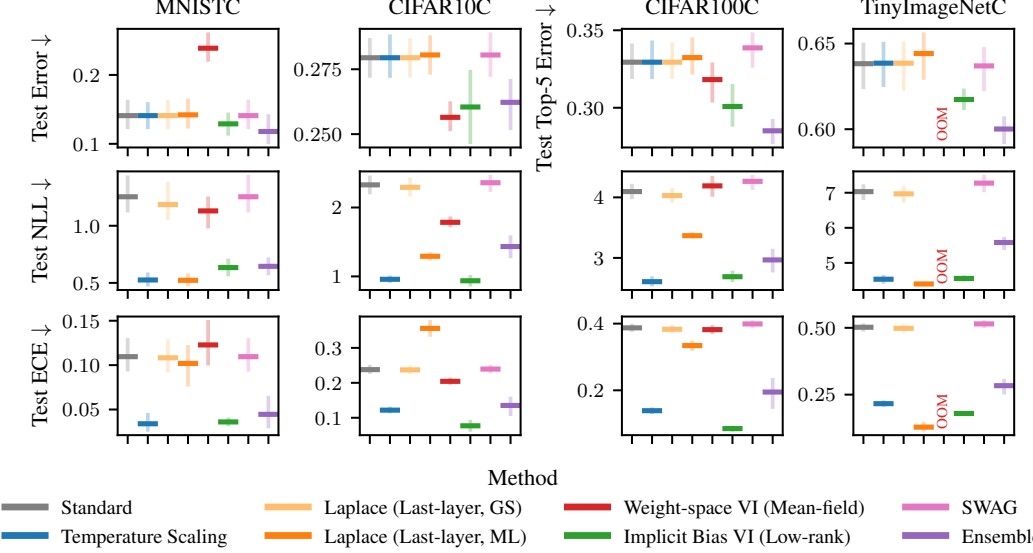

Figure 5: *Generalization on robustness benchmark problems.* When comparing different methods for Bayesian deep learning with regards to robustness to 15 different input corruptions, our approach, Implicit Bias VI, consistently has competitive uncertainty quantification across different datasets and metrics without sacrificing accuracy compared to a non-probabilistic network.

**Limitations** Compared to standard neural networks, when training via Implicit Bias VI, we observed that often lower learning rates were necessary due to the additional stochasticity in the objective (see also Section 3.3). While this does not have a significant impact on generalization, it sometimes requires slightly more epochs to achieve similar in-distribution performance to standard neural networks. Effectively, early in training it takes a bit more time for IBVI to become sufficiently certain about those features which are critical for in-distribution performance. This also means that folk knowledge on learning rate settings for specific architectures may not immediately transfer. In the experiments we train models with probabilistic in- and output layers with our approach, but we have so far not explored other covariance structures or where in the network probabilistic layers are most beneficial. While there is theoretical evidence that even just a single probabilistic hidden layer may be sufficient [39], we believe there is potential for improvement. Beyond the prior induced by the choice of parametrization, we did not experiment with more informative or learned priors, which could potentially give significant performance improvements on certain tasks [15].

## 6 Conclusion

In this paper, we demonstrated how to improve the robustness of deep neural networks while quantifying predictive uncertainty by exploiting the implicit regularization of (stochastic) gradient descent. We rigorously characterized this implicit bias for an overparametrized linear model and showed that our approach is equivalent to generalized variational inference with a 2-Wasserstein regularizer at reduced computational cost. We demonstrated the importance of parameterization and how it impacts the inductive bias via the initialization — thus conferring desirable properties such as learning rate transfer. Lastly, we empirically demonstrated competitive performance with state-of-the-art methods for Bayesian deep learning on a set of in- and out-of-distribution benchmarks with minimal computational overhead over standard deep learning. In principle, our approach is not restricted to Gaussian variational families and should seamlessly extend to location-scale families, which could further improve performance. Finally, it would be interesting to explore connections between Implicit Bias VI and Bayesian deep learning in function-space [e.g., 37, 54, 74–76].

### Acknowledgments

JW, BC, JM and JPC are supported by the Gatsby Charitable Foundation (GAT3708), the Simons Foundation (542963), the NSF AI Institute for Artificial and Natural Intelligence (ARNI: NSF DBI 2229929) and the Kavli Foundation. This work used the DeltaAI system at the National Center for Supercomputing Applications through allocations CIS250340 and CIS250292 from the Advanced Cyberinfrastructure Coordination Ecosystem: Services & Support (ACCESS) program, which is supported by U.S. National Science Foundation grants #2138259, #2138286, #2138307, #2137603, and #2138296. The authors would like to thank Hanna Dettki for valuable input, which significantly improved this paper.

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

SUPPLEMENTARY MATERIAL

This supplementary material contains additional results and proofs for all theoretical statements. References referring to sections, equations or theorem-type environments within this document are prefixed with 'S', while references to, or results from, the main paper are stated as is.

## S1 THEORETICAL RESULTS

**Lemma S1**
*Let $q(\boldsymbol{w}) = \mathcal{N}(\boldsymbol{w}; \boldsymbol{\mu}, \boldsymbol{\Sigma})$, $p(\boldsymbol{w}) = \mathcal{N}(\boldsymbol{w}; \boldsymbol{\mu}_0, \boldsymbol{\Sigma}_0)$ such that $\boldsymbol{\mu}, \boldsymbol{\mu}_0 \in \mathbb{R}^P$, $\boldsymbol{\Sigma}, \boldsymbol{\Sigma}_0 \in \mathbb{R}^{P \times P}$ positive semi-definite and let $\boldsymbol{V}_A \in \mathbb{R}^{P \times N}$, $\boldsymbol{V}_B \in \mathbb{R}^{P \times (P-N)}$ be matrices with pairwise orthonormal columns that together define an orthonormal basis of $\mathbb{R}^P$, i.e. for $\boldsymbol{V} = \begin{bmatrix} \boldsymbol{V}_A & \boldsymbol{V}_B \end{bmatrix}$ it holds that $\boldsymbol{V}\boldsymbol{V}^\mathsf{T} = \boldsymbol{V}^\mathsf{T}\boldsymbol{V} = \boldsymbol{I}$ and $\mathrm{span}(\boldsymbol{V}) = \mathbb{R}^P$. Assume further that*

$$\boldsymbol{V}_A^\mathsf{T} \boldsymbol{\Sigma} \boldsymbol{V}_A = \boldsymbol{0}, \tag{S11}$$

*then the squared 2-Wasserstein distance is given by*

$$\mathrm{W}_2^2(q, p) = \left\| \boldsymbol{V}_A^\mathsf{T} \boldsymbol{\mu} - \boldsymbol{V}_A^\mathsf{T} \boldsymbol{\mu}_0 \right\|_2^2 + \mathrm{W}_2^2\big(\mathcal{N}\big(\boldsymbol{V}_B^\mathsf{T}\boldsymbol{\mu}, \boldsymbol{V}_B^\mathsf{T}\boldsymbol{\Sigma}\boldsymbol{V}_B\big), \mathcal{N}\big(\boldsymbol{V}_B^\mathsf{T}\boldsymbol{\mu}_0, \boldsymbol{V}_B^\mathsf{T}\boldsymbol{\Sigma}_0\boldsymbol{V}_B\big)\big) + C, \tag{S12}$$

*where the constant $C$ is independent of $(\boldsymbol{\mu}, \boldsymbol{\Sigma})$.*

*Proof.* Consider the matrix

$$\boldsymbol{V}^\mathsf{T} \boldsymbol{\Sigma} \boldsymbol{V} = \begin{bmatrix} \boldsymbol{0}_{N \times N} & \boldsymbol{V}_A^\mathsf{T}\boldsymbol{\Sigma}\boldsymbol{V}_B \\ \boldsymbol{V}_B^\mathsf{T}\boldsymbol{\Sigma}\boldsymbol{V}_A & \boldsymbol{V}_B^\mathsf{T}\boldsymbol{\Sigma}\boldsymbol{V}_B \end{bmatrix}.$$

Since $\boldsymbol{V}^\mathsf{T}\boldsymbol{\Sigma}\boldsymbol{V}$ is symmetric positive semi-definite, its off-diagonal block $\boldsymbol{V}_A^\mathsf{T}\boldsymbol{\Sigma}\boldsymbol{V}_B$ satisfies

$$(\boldsymbol{I} - \boldsymbol{0}\boldsymbol{0}^\dagger)\boldsymbol{V}_A^\mathsf{T}\boldsymbol{\Sigma}\boldsymbol{V}_B = \boldsymbol{0} \iff \boldsymbol{V}_A^\mathsf{T}\boldsymbol{\Sigma}\boldsymbol{V}_B = \boldsymbol{0}$$

by Boyd and Vandenberghe [A5.5, 77]. Therefore, we have

$$\boldsymbol{V}^\mathsf{T}\boldsymbol{\Sigma}\boldsymbol{V} = \begin{bmatrix} \mathbf{0}_{N\times N} & \boldsymbol{V}_A^\mathsf{T}\boldsymbol{\Sigma}\boldsymbol{V}_B \\ \boldsymbol{V}_B^\mathsf{T}\boldsymbol{\Sigma}\boldsymbol{V}_A & \boldsymbol{V}_B^\mathsf{T}\boldsymbol{\Sigma}\boldsymbol{V}_B \end{bmatrix} = \begin{bmatrix} \mathbf{0}_{N\times N} & \mathbf{0}_{N\times(P-N)} \\ \mathbf{0}_{(P-N)\times N} & \boldsymbol{V}_B^\mathsf{T}\boldsymbol{\Sigma}\boldsymbol{V}_B \end{bmatrix}. \tag{S13}$$

The squared 2-Wasserstein distance between $q(\boldsymbol{w})$ and $p(\boldsymbol{w})$ is given by

$$\mathrm{W}_2^2(q,p) = \|\boldsymbol{\mu} - \boldsymbol{\mu}_0\|_2^2 + \mathrm{tr}(\boldsymbol{\Sigma} - 2(\boldsymbol{\Sigma}^{\frac{1}{2}}\boldsymbol{\Sigma}_0\boldsymbol{\Sigma}^{\frac{1}{2}})^{\frac{1}{2}} + \boldsymbol{\Sigma}_0).$$

For the squared norm term it holds by unitary invariance of $\|\cdot\|_2$ that

$$\|\boldsymbol{\mu} - \boldsymbol{\mu}_0\|_2^2 = \|\boldsymbol{V}^\mathsf{T}(\boldsymbol{\mu} - \boldsymbol{\mu}_0)\|_2^2 = \left\| \begin{bmatrix} \boldsymbol{V}_A^\mathsf{T}(\boldsymbol{\mu} - \boldsymbol{\mu}_0) \\ \boldsymbol{V}_B^\mathsf{T}(\boldsymbol{\mu} - \boldsymbol{\mu}_0) \end{bmatrix} \right\|_2^2 = \|\boldsymbol{V}_A^\mathsf{T}\boldsymbol{\mu} - \boldsymbol{V}_A^\mathsf{T}\boldsymbol{\mu}_0\|_2^2 + \|\boldsymbol{V}_B^\mathsf{T}\boldsymbol{\mu} - \boldsymbol{V}_B^\mathsf{T}\boldsymbol{\mu}_0\|_2^2.$$

Now for the trace term we have that

$$\begin{aligned}
&\mathrm{tr}(\boldsymbol{V}\boldsymbol{V}^\mathsf{T}(\boldsymbol{\Sigma} - 2(\boldsymbol{\Sigma}^{\frac{1}{2}}\boldsymbol{\Sigma}_0\boldsymbol{\Sigma}^{\frac{1}{2}})^{\frac{1}{2}} + \boldsymbol{\Sigma}_0)) \\
&= \mathrm{tr}(\boldsymbol{V}^\mathsf{T}\boldsymbol{\Sigma}\boldsymbol{V}) - 2\,\mathrm{tr}(\boldsymbol{V}^\mathsf{T}(\boldsymbol{\Sigma}^{\frac{1}{2}}\boldsymbol{\Sigma}_0\boldsymbol{\Sigma}^{\frac{1}{2}})^{\frac{1}{2}}\boldsymbol{V}) + \mathrm{tr}(\boldsymbol{V}^\mathsf{T}\boldsymbol{\Sigma}_0\boldsymbol{V}) \\
&= \mathrm{tr}(\boldsymbol{V}_A^\mathsf{T}\boldsymbol{\Sigma}\boldsymbol{V}_A) + \mathrm{tr}(\boldsymbol{V}_B^\mathsf{T}\boldsymbol{\Sigma}\boldsymbol{V}_B) + \mathrm{tr}(\boldsymbol{V}_A^\mathsf{T}\boldsymbol{\Sigma}_0\boldsymbol{V}_A) + \mathrm{tr}(\boldsymbol{V}_B^\mathsf{T}\boldsymbol{\Sigma}_0\boldsymbol{V}_B) - 2\,\mathrm{tr}(\boldsymbol{V}^\mathsf{T}(\boldsymbol{\Sigma}^{\frac{1}{2}}\boldsymbol{\Sigma}_0\boldsymbol{\Sigma}^{\frac{1}{2}})^{\frac{1}{2}}\boldsymbol{V}) \\
&\overset{\pm c}{=} \mathrm{tr}(\boldsymbol{V}_B^\mathsf{T}\boldsymbol{\Sigma}\boldsymbol{V}_B) + \mathrm{tr}(\boldsymbol{V}_B^\mathsf{T}\boldsymbol{\Sigma}_0\boldsymbol{V}_B) - 2\,\mathrm{tr}(\boldsymbol{V}^\mathsf{T}(\boldsymbol{\Sigma}^{\frac{1}{2}}\boldsymbol{\Sigma}_0\boldsymbol{\Sigma}^{\frac{1}{2}})^{\frac{1}{2}}\boldsymbol{V})
\end{aligned} \tag{S14}$$

where we used Eq. (S11) and $\overset{\pm c}{=}$ denotes equality up to constants independent of $(\boldsymbol{\mu}, \boldsymbol{\Sigma})$.

Now by Eq. (S13), we have that $\boldsymbol{\Sigma} = \boldsymbol{V}_B\boldsymbol{M}\boldsymbol{V}_B^\mathsf{T}$ for $\boldsymbol{M} = \boldsymbol{V}_B^\mathsf{T}\boldsymbol{\Sigma}\boldsymbol{V}_B$ and its unique principal square root is given by $\boldsymbol{\Sigma}^{\frac{1}{2}} = \boldsymbol{V}_B\boldsymbol{M}^{\frac{1}{2}}\boldsymbol{V}_B^\mathsf{T}$ since

$$(\boldsymbol{V}_B\boldsymbol{M}^{\frac{1}{2}}\boldsymbol{V}_B^\mathsf{T})(\boldsymbol{V}_B\boldsymbol{M}^{\frac{1}{2}}\boldsymbol{V}_B^\mathsf{T}) = \boldsymbol{V}_B\boldsymbol{M}^{\frac{1}{2}}\boldsymbol{I}_{(P-N)\times(P-N)}\boldsymbol{M}^{\frac{1}{2}}\boldsymbol{V}_B^\mathsf{T} = \boldsymbol{\Sigma}.$$

It also holds that the unique principal square root

$$(\boldsymbol{\Sigma}^{\frac{1}{2}}\boldsymbol{\Sigma}_0\boldsymbol{\Sigma}^{\frac{1}{2}})^{\frac{1}{2}} = \boldsymbol{V}_B(\boldsymbol{M}^{\frac{1}{2}}\boldsymbol{V}_B^\mathsf{T}\boldsymbol{\Sigma}_0\boldsymbol{V}_B\boldsymbol{M}^{\frac{1}{2}})^{\frac{1}{2}}\boldsymbol{V}_B^\mathsf{T}$$

since direct calculation gives

$$\begin{aligned}
(\boldsymbol{V}_B(\boldsymbol{M}^{\frac{1}{2}}\boldsymbol{V}_B^\mathsf{T}\boldsymbol{\Sigma}_0\boldsymbol{V}_B\boldsymbol{M}^{\frac{1}{2}})^{\frac{1}{2}}\boldsymbol{V}_B^\mathsf{T})(\boldsymbol{V}_B(\boldsymbol{M}^{\frac{1}{2}}\boldsymbol{V}_B^\mathsf{T}\boldsymbol{\Sigma}_0\boldsymbol{V}_B\boldsymbol{M}^{\frac{1}{2}})^{\frac{1}{2}}\boldsymbol{V}_B^\mathsf{T}) \\
= \boldsymbol{V}_B\boldsymbol{M}^{\frac{1}{2}}\boldsymbol{V}_B^\mathsf{T}\boldsymbol{\Sigma}_0\boldsymbol{V}_B\boldsymbol{M}^{\frac{1}{2}}\boldsymbol{V}_B^\mathsf{T} = \boldsymbol{\Sigma}^{\frac{1}{2}}\boldsymbol{\Sigma}_0\boldsymbol{\Sigma}^{\frac{1}{2}}.
\end{aligned}$$

Therefore we have that

$$\mathrm{tr}(\boldsymbol{V}^\mathsf{T}(\boldsymbol{\Sigma}^{\frac{1}{2}}\boldsymbol{\Sigma}_0\boldsymbol{\Sigma}^{\frac{1}{2}})^{\frac{1}{2}}\boldsymbol{V}) = \mathrm{tr}(\boldsymbol{V}^\mathsf{T}\boldsymbol{V}_B(\boldsymbol{M}^{\frac{1}{2}}\boldsymbol{V}_B^\mathsf{T}\boldsymbol{\Sigma}_0\boldsymbol{V}_B\boldsymbol{M}^{\frac{1}{2}})^{\frac{1}{2}}\boldsymbol{V}_B^\mathsf{T}\boldsymbol{V}) = \mathrm{tr}((\boldsymbol{M}^{\frac{1}{2}}\boldsymbol{V}_B^\mathsf{T}\boldsymbol{\Sigma}_0\boldsymbol{V}_B\boldsymbol{M}^{\frac{1}{2}})^{\frac{1}{2}}).$$

Putting it all together we obtain

$$\begin{aligned}
\mathrm{W}_2^2(q,p) &\overset{\pm c}{=} \|\boldsymbol{V}_A^\mathsf{T}\boldsymbol{\mu} - \boldsymbol{V}_A^\mathsf{T}\boldsymbol{\mu}_0\|_2^2 + \|\boldsymbol{V}_B^\mathsf{T}\boldsymbol{\mu} - \boldsymbol{V}_B^\mathsf{T}\boldsymbol{\mu}_0\|_2^2 + \mathrm{tr}(\boldsymbol{V}_B^\mathsf{T}\boldsymbol{\Sigma}\boldsymbol{V}_B) + \mathrm{tr}(\boldsymbol{V}_B^\mathsf{T}\boldsymbol{\Sigma}_0\boldsymbol{V}_B) - 2\,\mathrm{tr}(\boldsymbol{V}^\mathsf{T}(\boldsymbol{\Sigma}^{\frac{1}{2}}\boldsymbol{\Sigma}_0\boldsymbol{\Sigma}^{\frac{1}{2}})^{\frac{1}{2}}\boldsymbol{V}) \\
&= \|\boldsymbol{V}_A^\mathsf{T}\boldsymbol{\mu} - \boldsymbol{V}_A^\mathsf{T}\boldsymbol{\mu}_0\|_2^2 + \|\boldsymbol{V}_B^\mathsf{T}\boldsymbol{\mu} - \boldsymbol{V}_B^\mathsf{T}\boldsymbol{\mu}_0\|_2^2 + \mathrm{tr}(\boldsymbol{V}_B^\mathsf{T}\boldsymbol{\Sigma}\boldsymbol{V}_B) + \mathrm{tr}(\boldsymbol{V}_B^\mathsf{T}\boldsymbol{\Sigma}_0\boldsymbol{V}_B) - 2\,\mathrm{tr}((\boldsymbol{M}^{\frac{1}{2}}\boldsymbol{V}_B^\mathsf{T}\boldsymbol{\Sigma}_0\boldsymbol{V}_B\boldsymbol{M}^{\frac{1}{2}})^{\frac{1}{2}}) \\
&= \|\boldsymbol{V}_A^\mathsf{T}\boldsymbol{\mu} - \boldsymbol{V}_A^\mathsf{T}\boldsymbol{\mu}_0\|_2^2 + \mathrm{W}_2^2(\mathcal{N}(\boldsymbol{V}_B^\mathsf{T}\boldsymbol{\mu}, \boldsymbol{V}_B^\mathsf{T}\boldsymbol{\Sigma}\boldsymbol{V}_B), \mathcal{N}(\boldsymbol{V}_B^\mathsf{T}\boldsymbol{\mu}_0, \boldsymbol{V}_B^\mathsf{T}\boldsymbol{\Sigma}_0\boldsymbol{V}_B))
\end{aligned}$$

which completes the proof. $\qquad\square$

## S1.1 Overparametrized Linear Regression

### S1.1.1 Characterization of Implicit Bias (Proof of Theorem 1)

**Theorem 1** (Implicit Bias in Regression)
*Let $f_{\boldsymbol{w}}(\boldsymbol{x}) = \boldsymbol{x}^\mathsf{T}\boldsymbol{w}$ be an overparametrized linear model with $P > N$. Define a Gaussian prior $p(\boldsymbol{w}) = \mathcal{N}(\boldsymbol{w}; \boldsymbol{\mu}_0, \boldsymbol{S}_0\boldsymbol{S}_0^\mathsf{T})$ and likelihood $p(\boldsymbol{y} \mid \boldsymbol{w}) = \mathcal{N}(\boldsymbol{y}; f_{\boldsymbol{w}}(\boldsymbol{X}), \sigma^2\boldsymbol{I})$ and assume a variational family $q_{\boldsymbol{\theta}}(\boldsymbol{w}) = \mathcal{N}(\boldsymbol{w}; \boldsymbol{\mu}, \boldsymbol{S}\boldsymbol{S}^\mathsf{T})$ with $\boldsymbol{\theta} = (\boldsymbol{\mu}, \boldsymbol{S})$ such that $\boldsymbol{\mu} \in \mathbb{R}^P$ and $\boldsymbol{S} \in \mathbb{R}^{P\times R}$ where $R \le P$. If the learning rate sequence $(\eta_t)_t$ is chosen such that the limit point $\boldsymbol{\theta}_\star^{\mathrm{GD}} = \lim_{t\to\infty}\boldsymbol{\theta}_t^{\mathrm{GD}}$ identified by gradient descent, initialized at $\boldsymbol{\theta}_0 = (\boldsymbol{\mu}_0, \boldsymbol{S}_0)$, is a (global) minimizer of the expected log-likelihood $\bar{\ell}(\boldsymbol{\theta})$, then*

$$\boldsymbol{\theta}_\star^{\mathrm{GD}} \in \operatorname*{arg\,min}_{\substack{\boldsymbol{\theta}=(\boldsymbol{\mu},\boldsymbol{S}) \\ s.t.\ \boldsymbol{\theta}\in\arg\min\bar{\ell}(\boldsymbol{\theta})}} \mathrm{W}_2^2(q_{\boldsymbol{\theta}}, p). \tag{7}$$

*Further, this also holds in the case of stochastic gradient descent and when using momentum.*

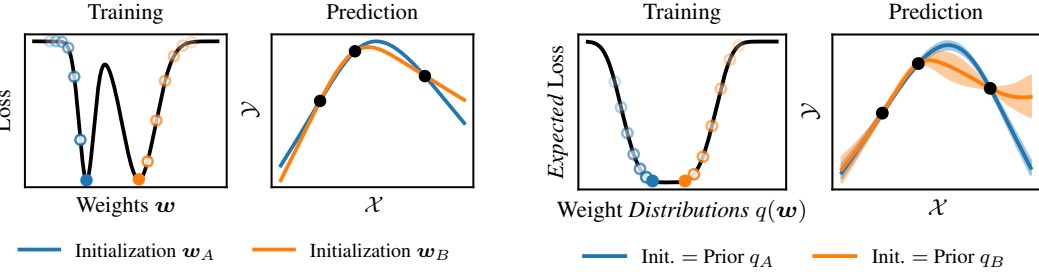

(a) NN trained with *no explicit regularization.*  (b) BNN trained with *no explicit regularization.*

Figure S1: *Implicit regularization in standard neural networks versus in probabilistic networks.* Left panels: A neural network trained without explicit regularization can converge to different global minima of the loss. Optimization of the weights will implicitly regularize towards one or the other. Right panels: Analogously, there are multiple distributions over neural networks that are global minima of the *expected* loss. Optimization of the *distribution* over the weights will implicitly regularize towards one or the other. Our approach uses this implicit regularization instead of an explicit regularization to a prior.

*Proof.* Let $\boldsymbol{\theta}_\star = (\boldsymbol{\mu}_\star, \boldsymbol{S}_\star)$ be a minimizer of $\bar{\ell}(\boldsymbol{\theta})$. By assumption it holds that the expected negative log-likelihood is equal to the following non-negative loss function up to an additive constant:

$$\bar{\ell}(\boldsymbol{\theta}) = \mathbb{E}_{q_{\boldsymbol{\theta}}(\boldsymbol{w})}(\ell(\boldsymbol{y}, f_{\boldsymbol{w}}(\boldsymbol{X}))) = \mathbb{E}_{q_{\boldsymbol{\theta}}(\boldsymbol{w})}(-\log p(\boldsymbol{y} \mid \boldsymbol{w}))$$

$$\overset{+c}{=} \frac{1}{2\sigma^2} \mathbb{E}_{q_{\boldsymbol{\theta}}(\boldsymbol{w})}(\|\boldsymbol{y} - \boldsymbol{X}\boldsymbol{w}\|_2^2)$$

$$= \frac{1}{2\sigma^2}(\|\boldsymbol{y} - \boldsymbol{X}\boldsymbol{\mu}\|_2^2 + \mathrm{tr}(\boldsymbol{X}\boldsymbol{\Sigma}\boldsymbol{X}^\mathsf{T})) \geq 0,$$

where $\boldsymbol{\Sigma} = \boldsymbol{S}\boldsymbol{S}^\mathsf{T}$ and non-negativity follows from $\boldsymbol{\Sigma}$ being symmetric positive semi-definite. Therefore any (global) minimizer $\boldsymbol{\theta}_\star = (\boldsymbol{\mu}_\star, \boldsymbol{\Sigma}_\star)$ necessarily satisfies

$$\|\boldsymbol{y} - \boldsymbol{X}\boldsymbol{\mu}_\star\|_2^2 = 0, \tag{S15}$$

$$\mathrm{tr}(\boldsymbol{X}\boldsymbol{\Sigma}_\star\boldsymbol{X}^\mathsf{T}) = 0. \tag{S16}$$

Let $\boldsymbol{V} = [\boldsymbol{V}_{\mathrm{range}} \quad \boldsymbol{V}_{\mathrm{null}}] \in \mathbb{R}^{P \times P}$ be the orthonormal matrix of right singular vectors of $\boldsymbol{X} = \boldsymbol{U}\boldsymbol{\Lambda}\boldsymbol{V}^\mathsf{T}$, where $\boldsymbol{V}_{\mathrm{range}} \in \mathbb{R}^{P \times N}$ and $\boldsymbol{V}_{\mathrm{null}} \in \mathbb{R}^{P \times (P-N)}$. Since $\boldsymbol{X} \in \mathbb{R}^{N \times P}$ and we are in the overparametrized regime, i.e. $P > N$, the optimal mean parameter decomposes into the least-squares solution and a null space contribution

$$\boldsymbol{\mu}_\star = \boldsymbol{V}_{\mathrm{range}}\boldsymbol{u}_\star + \boldsymbol{V}_{\mathrm{null}}\boldsymbol{z} = \boldsymbol{X}^\dagger\boldsymbol{y} + \boldsymbol{V}_{\mathrm{null}}\boldsymbol{z}. \tag{S17}$$

Furthermore, it holds for positive semi-definite $\boldsymbol{\Sigma} \in \mathbb{R}^{P \times P}$ that

$$0 \leq \mathrm{tr}(\boldsymbol{X}\boldsymbol{\Sigma}\boldsymbol{X}^\mathsf{T}) = \mathrm{tr}(\boldsymbol{U}\boldsymbol{\Lambda}\boldsymbol{V}^\mathsf{T}\boldsymbol{\Sigma}\boldsymbol{V}\boldsymbol{\Lambda}\boldsymbol{U}^\mathsf{T}) = \mathrm{tr}(\boldsymbol{\Lambda}\boldsymbol{V}^\mathsf{T}\boldsymbol{\Sigma}\boldsymbol{V}\boldsymbol{\Lambda})$$

$$= \mathrm{tr}([\boldsymbol{\Lambda}_{N \times N} \quad \boldsymbol{0}] \begin{bmatrix} \boldsymbol{V}_{\mathrm{range}}^\mathsf{T}\boldsymbol{\Sigma}\boldsymbol{V}_{\mathrm{range}} & * \\ * & * \end{bmatrix} \begin{bmatrix} \boldsymbol{\Lambda}_{N \times N} \\ \boldsymbol{0} \end{bmatrix})$$

$$= \mathrm{tr}(\boldsymbol{\Lambda}_{N \times N}\boldsymbol{V}_{\mathrm{range}}^\mathsf{T}\boldsymbol{\Sigma}\boldsymbol{V}_{\mathrm{range}}\boldsymbol{\Lambda}_{N \times N})$$

$$= \sum_{i=1}^{N} \lambda_i^2 [\boldsymbol{V}_{\mathrm{range}}^\mathsf{T}\boldsymbol{\Sigma}\boldsymbol{V}_{\mathrm{range}}]_{ii}$$

where $\lambda_i^2 > 0$ are the squared singular values of $\boldsymbol{X}$, which are strictly positive since $\mathrm{rank}(\boldsymbol{X}) = N$. Therefore using Equation (S16) any global minimizer necessarily satisfies $[\boldsymbol{V}_{\mathrm{range}}^\mathsf{T}\boldsymbol{\Sigma}_\star\boldsymbol{V}_{\mathrm{range}}]_{ii} = 0$ for $i \in \{1, \ldots, N\}$. Now since $\boldsymbol{V}_{\mathrm{range}}^\mathsf{T}\boldsymbol{\Sigma}_\star\boldsymbol{V}_{\mathrm{range}}$ is symmetric positive semi-definite and its diagonal is zero, so is its trace and therefore the sum of its non-negative eigenvalues is necessarily zero. Thus all eigenvalues are zero and therefore

$$\boldsymbol{V}_{\mathrm{range}}^\mathsf{T}\boldsymbol{\Sigma}\boldsymbol{V}_{\mathrm{range}} = \boldsymbol{0}. \tag{S18}$$

Now by Lemma S1 we have that the squared 2-Wasserstein distance between $q_{\boldsymbol{\theta}_\star}(\boldsymbol{w}) = \mathcal{N}(\boldsymbol{w}; \boldsymbol{\mu}_\star, \boldsymbol{\Sigma}_\star)$ and the initialization $p(\boldsymbol{w}) = \mathcal{N}(\boldsymbol{w}; \boldsymbol{\mu}_0, \boldsymbol{\Sigma}_0)$ is given up to a constant independent of $(\boldsymbol{\mu}_\star, \boldsymbol{\Sigma}_\star)$ by

$$
\begin{aligned}
\mathrm{W}_2(q_{\boldsymbol{\theta}_\star}, p) &\stackrel{+c}{=} \left\| \boldsymbol{V}_{\text{range}}^\mathsf{T} \boldsymbol{\mu}_\star - \boldsymbol{V}_{\text{range}}^\mathsf{T} \boldsymbol{\mu}_0 \right\|_2^2 + \mathrm{W}_2^2\big( \mathcal{N}(\boldsymbol{V}_{\text{null}}^\mathsf{T} \boldsymbol{\mu}_\star, \boldsymbol{V}_{\text{null}}^\mathsf{T} \boldsymbol{\Sigma}_\star \boldsymbol{V}_{\text{null}}), \mathcal{N}(\boldsymbol{V}_{\text{null}}^\mathsf{T} \boldsymbol{\mu}_0, \boldsymbol{V}_{\text{null}}^\mathsf{T} \boldsymbol{\Sigma}_0 \boldsymbol{V}_{\text{null}}) \big) \\
&= \left\| \boldsymbol{X}^\dagger \boldsymbol{y} - \boldsymbol{V}_{\text{range}}^\mathsf{T} \boldsymbol{\mu}_0 \right\|_2^2 + \mathrm{W}_2^2\big( \mathcal{N}(\boldsymbol{V}_{\text{null}}^\mathsf{T} \boldsymbol{\mu}_\star, \boldsymbol{V}_{\text{null}}^\mathsf{T} \boldsymbol{\Sigma}_\star \boldsymbol{V}_{\text{null}}), \mathcal{N}(\boldsymbol{V}_{\text{null}}^\mathsf{T} \boldsymbol{\mu}_0, \boldsymbol{V}_{\text{null}}^\mathsf{T} \boldsymbol{\Sigma}_0 \boldsymbol{V}_{\text{null}}) \big) \\
&\stackrel{+c}{=} \mathrm{W}_2^2\big( \mathcal{N}(\boldsymbol{V}_{\text{null}}^\mathsf{T} \boldsymbol{\mu}_\star, \boldsymbol{V}_{\text{null}}^\mathsf{T} \boldsymbol{\Sigma}_\star \boldsymbol{V}_{\text{null}}), \mathcal{N}(\boldsymbol{V}_{\text{null}}^\mathsf{T} \boldsymbol{\mu}_0, \boldsymbol{V}_{\text{null}}^\mathsf{T} \boldsymbol{\Sigma}_0 \boldsymbol{V}_{\text{null}}) \big).
\end{aligned}
$$

Therefore among variational distributions $q_{\boldsymbol{\theta}_\star}$ with parameters $\boldsymbol{\theta}_\star$ that minimize the expected loss $\bar{\ell}(\boldsymbol{\theta})$, any such $\boldsymbol{\theta}_\star$ that minimizes the squared 2-Wasserstein distance to the prior satisfies

$$
(\underbrace{\boldsymbol{V}_{\text{null}}^\mathsf{T} \boldsymbol{\mu}_\star}_{=:\boldsymbol{z}}, \underbrace{\boldsymbol{V}_{\text{null}}^\mathsf{T} \boldsymbol{\Sigma}_\star \boldsymbol{V}_{\text{null}}}_{=:\boldsymbol{M}}) = (\boldsymbol{V}_{\text{null}}^\mathsf{T} \boldsymbol{\mu}_0, \boldsymbol{V}_{\text{null}}^\mathsf{T} \boldsymbol{\Sigma}_0 \boldsymbol{V}_{\text{null}}). \tag{S19}
$$

**(Stochastic) Gradient Descent**   It remains to show that (stochastic) gradient descent identifies a minimum of the expected loss $\bar{\ell}(\boldsymbol{\theta})$, such that the above holds. By assumption we have for the loss on a batch $\boldsymbol{X}_b$ of data that

$$
\begin{aligned}
\bar{\ell}(\boldsymbol{\theta}) &= \mathbb{E}_{q_{\boldsymbol{\theta}}(\boldsymbol{w})}(\ell(\boldsymbol{y}_b, f_{\boldsymbol{w}}(\boldsymbol{X}_b))) = \mathbb{E}_{q_{\boldsymbol{\theta}}(\boldsymbol{w})}(-\log p(\boldsymbol{y}_b \mid \boldsymbol{w})) \\
&\stackrel{+c}{=} \frac{1}{2\sigma^2}\big( \|\boldsymbol{y}_b - \boldsymbol{X}_b \boldsymbol{\mu}\|_2^2 + \mathrm{tr}(\boldsymbol{X}_b \boldsymbol{\Sigma} \boldsymbol{X}_b^\mathsf{T}) \big).
\end{aligned}
$$

Therefore, at convergence of (stochastic) gradient descent the variational parameters $\boldsymbol{\theta}_\infty = (\boldsymbol{\mu}_\infty, \boldsymbol{S}_\infty)$ are given by

$$
\boldsymbol{\mu}_\infty = \boldsymbol{\mu}_0 - \sum_{t=1}^\infty \eta_t \nabla_{\boldsymbol{\mu}} \bar{\ell}_b(\boldsymbol{\theta}_{t-1}) = \boldsymbol{\mu}_0 + \sum_{t=1}^\infty \frac{\eta_t}{\sigma^2} \boldsymbol{X}_b^\mathsf{T} (\boldsymbol{y}_b - \boldsymbol{X}_b \boldsymbol{\mu}_{t-1})
$$

as well as

$$
\boldsymbol{S}_\infty = \boldsymbol{S}_0 - \sum_{t=1}^\infty \eta_t \nabla_{\boldsymbol{S}} \bar{\ell}_b(\boldsymbol{\theta}_{t-1}) = \boldsymbol{S}_0 - \sum_{t=1}^\infty \frac{\eta_t}{\sigma^2} \boldsymbol{X}_b^\mathsf{T} \boldsymbol{X}_b \boldsymbol{S}_{t-1}
$$

and therefore

$$
\boldsymbol{z}_\infty = \boldsymbol{V}_{\text{null}}^\mathsf{T} \boldsymbol{\mu}_\infty = \boldsymbol{V}_{\text{null}}^\mathsf{T} \boldsymbol{\mu}_0 + \sum_{t=1}^\infty \frac{\eta_t}{\sigma^2} \boldsymbol{V}_{\text{null}}^\mathsf{T} \underbrace{\boldsymbol{X}_b^\mathsf{T}(\boldsymbol{y}_b - \boldsymbol{X}_b \boldsymbol{\mu}_{t-1})}_{\in \text{range}(\boldsymbol{X}_b^\mathsf{T})} = \boldsymbol{V}_{\text{null}}^\mathsf{T} \boldsymbol{\mu}_0
$$

$$
\boldsymbol{V}_{\text{null}}^\mathsf{T} \boldsymbol{S}_\infty = \boldsymbol{V}_{\text{null}}^\mathsf{T} \boldsymbol{S}_0 - \sum_{t=1}^\infty \frac{\eta_t}{\sigma^2} \boldsymbol{V}_{\text{null}}^\mathsf{T} \underbrace{\boldsymbol{X}_b^\mathsf{T} \boldsymbol{X}_b \boldsymbol{S}_{t-1}}_{\text{columns} \in \text{range}(\boldsymbol{X}_b^\mathsf{T})} = \boldsymbol{V}_{\text{null}}^\mathsf{T} \boldsymbol{S}_0
$$

where we used continuity of linear maps between finite-dimensional spaces. It follows that

$$
\boldsymbol{M}_\infty = \boldsymbol{V}_{\text{null}}^\mathsf{T} \boldsymbol{\Sigma}_\infty \boldsymbol{V}_{\text{null}} = \boldsymbol{V}_{\text{null}}^\mathsf{T} \boldsymbol{S}_\infty \boldsymbol{S}_\infty^\mathsf{T} \boldsymbol{V}_{\text{null}} = \boldsymbol{V}_{\text{null}}^\mathsf{T} \boldsymbol{S}_0 \boldsymbol{S}_0^\mathsf{T} \boldsymbol{V}_{\text{null}} = \boldsymbol{V}_{\text{null}}^\mathsf{T} \boldsymbol{\Sigma}_0 \boldsymbol{V}_{\text{null}}.
$$

Therefore any limit point of (stochastic) gradient descent that minimizes the expected log-likelihood also minimizes the 2-Wasserstein distance to the prior, since $\boldsymbol{\theta}_\infty$ satisfies Equation (S19).

**Momentum**   In case we are using (stochastic) gradient descent with momentum, the updates are given by

$$
\begin{aligned}
\boldsymbol{\mu}_{t+1} &= \boldsymbol{\mu}_t + \gamma_t \Delta \boldsymbol{\mu}_t - \eta_t \nabla_{\boldsymbol{\mu}} \bar{\ell}_b(\boldsymbol{\theta}_t + \alpha_t \Delta \boldsymbol{\theta}_t) \\
\boldsymbol{S}_{t+1} &= \boldsymbol{S}_t + \gamma_t \Delta \boldsymbol{S}_t - \eta_t \nabla_{\boldsymbol{S}} \bar{\ell}_b(\boldsymbol{\theta}_t + \alpha_t \Delta \boldsymbol{\theta}_t)
\end{aligned} \tag{S20}
$$

where

$$
\Delta \boldsymbol{\theta}_t = \begin{pmatrix} \Delta \boldsymbol{\mu}_t \\ \Delta \boldsymbol{S}_t \end{pmatrix} = \boldsymbol{\theta}_t - \boldsymbol{\theta}_{t-1}, \qquad \Delta \boldsymbol{\theta}_0 = \boldsymbol{0}.
$$

for parameters $\gamma_t, \alpha_t \geq 0$, which includes Nesterov's acceleration ($\gamma_t = \alpha_t$) [78] and heavy ball momentum ($\alpha_t = 0$) [79].

To prove that the updates of the variational parameters are always orthogonal to the null space of $\boldsymbol{X}_b$, we proceed by induction. The base case is trivial since $\Delta\boldsymbol{\theta}_0 = \boldsymbol{0}$. Assume now that $\boldsymbol{V}_{\text{null}}^{\mathsf{T}}\Delta\boldsymbol{\mu}_t = \boldsymbol{0}$ and $\boldsymbol{V}_{\text{null}}^{\mathsf{T}}\Delta\boldsymbol{S}_t = \boldsymbol{0}$, then by Equation (S20), we have

$$\boldsymbol{V}_{\text{null}}^{\mathsf{T}}\Delta\boldsymbol{\mu}_{t+1} = \boldsymbol{V}_{\text{null}}^{\mathsf{T}}(\boldsymbol{\mu}_{t+1} - \boldsymbol{\mu}_t) = \gamma_t \boldsymbol{V}_{\text{null}}^{\mathsf{T}}\Delta\boldsymbol{\mu}_t - \eta_t \boldsymbol{V}_{\text{null}}^{\mathsf{T}}\nabla_{\boldsymbol{\mu}}\bar{\ell}_b(\boldsymbol{\theta}_t + \alpha_t\Delta\boldsymbol{\theta}_t) = \boldsymbol{0}$$

$$\boldsymbol{V}_{\text{null}}^{\mathsf{T}}\Delta\boldsymbol{S}_{t+1} = \boldsymbol{V}_{\text{null}}^{\mathsf{T}}(\boldsymbol{S}_{t+1} - \boldsymbol{S}_t) = \gamma_t \boldsymbol{V}_{\text{null}}^{\mathsf{T}}\Delta\boldsymbol{S}_t - \eta_t \boldsymbol{V}_{\text{null}}^{\mathsf{T}}\nabla_{\boldsymbol{S}}\bar{\ell}_b(\boldsymbol{\theta}_t + \alpha_t\Delta\boldsymbol{\theta}_t) = \boldsymbol{0}$$

where we used the induction hypothesis and the fact that the gradients are orthogonal to the null space as shown earlier.

Therefore by the same argument as above we have that $\boldsymbol{\theta}_\infty$ computed via (stochastic) gradient descent with momentum satisfies Equation (S19), which directly implies Theorem 1. $\qquad\square$

### S1.1.2 Non-Asymptotic Error Analysis

**Theorem S3** (Non-Asymptotic Error of Gradient Flow)
*Let $f_{\boldsymbol{w}}(\boldsymbol{x}) = \boldsymbol{x}^{\mathsf{T}}\boldsymbol{w}$ be a linear model. Define a prior $p(\boldsymbol{w}) = \mathcal{N}(\boldsymbol{w}; \boldsymbol{\mu}_0, \boldsymbol{S}_0\boldsymbol{S}_0^{\mathsf{T}})$ and assume noise-free observations $y(\cdot) = f_{\boldsymbol{w}}(\cdot)$ for $\boldsymbol{w} \sim p(\boldsymbol{w})$. Further, define a variational distribution $q_{\boldsymbol{\theta}}(\boldsymbol{w}) = \mathcal{N}(\boldsymbol{w}; \boldsymbol{\mu}, \boldsymbol{S}\boldsymbol{S}^{\mathsf{T}})$ with $\boldsymbol{\theta} = (\boldsymbol{\mu}, \boldsymbol{S})$ such that $\boldsymbol{\mu} \in \mathbb{R}^P$ and $\boldsymbol{S} \in \mathbb{R}^{P\times R}$ where $R \leq P$. Let $\boldsymbol{\theta}(t) = (\boldsymbol{\mu}(t), \boldsymbol{S}(t))$ be the variational parameters at time $t \geq 0$ given by the gradient flow of the expected loss*

$$\dot{\boldsymbol{\theta}}(t) = -\nabla_{\boldsymbol{\theta}}\bar{\ell}(\boldsymbol{\theta}(t)) \tag{S21}$$

*initialized at $\boldsymbol{\theta}(0) = (\boldsymbol{\mu}_0, \boldsymbol{S}_0)$. Then the expected squared error of the mean prediction*

$$\mathbb{E}_{\binom{\boldsymbol{y}}{y_{\text{test}}}}\left(\left(y_{\text{test}} - f_{\boldsymbol{\mu}(t)}(\boldsymbol{x}_{\text{test}})\right)^2\right) = \text{Var}_{\boldsymbol{w}\sim q_{\boldsymbol{\theta}(t)}}(f_{\boldsymbol{w}}(\boldsymbol{x}_{\text{test}})) \tag{S22}$$

*at any test point $\boldsymbol{x}_{\text{test}} \in \mathbb{R}^P$. In other words, assuming the training and test data are drawn from the prior predictive, the predictive error of $f_{\boldsymbol{\mu}(t)}(\cdot)$ at any time $t \geq 0$ is* exactly *quantified by the predictive uncertainty of the variational distribution, not only at initialization and in the limit $t \to \infty$.*

*Proof.* The dynamics of the variational parameters as defined by the gradient flow in Equation (S21) are given by

$$\dot{\boldsymbol{\mu}}(t) = -\nabla_{\boldsymbol{\mu}}\bar{\ell}(\boldsymbol{\mu}(t)) = \boldsymbol{X}^{\mathsf{T}}(\boldsymbol{y} - \boldsymbol{X}\boldsymbol{\mu}(t)) = -\boldsymbol{X}^{\mathsf{T}}\boldsymbol{X}(\boldsymbol{\mu}(t) - \boldsymbol{w}) = \frac{d}{dt}(\boldsymbol{\mu}(t) - \boldsymbol{w}),$$

$$\dot{\boldsymbol{S}}(t) = -\nabla_{\boldsymbol{S}}\bar{\ell}(\boldsymbol{S}(t)) = -\boldsymbol{X}^{\mathsf{T}}\boldsymbol{X}\boldsymbol{S}(t).$$

Since these dynamics are matrix differential equations, the mean and covariance parameters as a function of time are given by

$$\boldsymbol{\mu}(t) = \boldsymbol{w} + e^{-\boldsymbol{X}^{\mathsf{T}}\boldsymbol{X}t}(\boldsymbol{\mu}_0 - \boldsymbol{w}), \tag{S23}$$

$$\boldsymbol{S}(t) = e^{-\boldsymbol{X}^{\mathsf{T}}\boldsymbol{X}t}\boldsymbol{S}_0. \tag{S24}$$

Thus the expected predictive error at time step $t \geq 0$ is given by

$$\mathbb{E}_{\binom{\boldsymbol{y}}{y_{\text{test}}}}\left(\left\|y_{\text{test}} - f_{\boldsymbol{\mu}(t)}(\boldsymbol{x}_{\text{test}})\right\|_2^2\right) = \mathbb{E}_{\binom{\boldsymbol{X}}{\boldsymbol{x}_{\text{test}}}\boldsymbol{w}}\left(\left\|y_{\text{test}} - \boldsymbol{x}_{\text{test}}^{\mathsf{T}}\boldsymbol{\mu}(t)\right\|_2^2\right)$$

$$= \mathbb{E}_{\boldsymbol{w}}\left(\left\|\boldsymbol{x}_{\text{test}}^{\mathsf{T}}\boldsymbol{w} - \boldsymbol{x}_{\text{test}}^{\mathsf{T}}\left(\boldsymbol{w} + e^{-\boldsymbol{X}^{\mathsf{T}}\boldsymbol{X}t}(\boldsymbol{\mu}_0 - \boldsymbol{w})\right)\right\|_2^2\right)$$

$$= \mathbb{E}_{\boldsymbol{w}}\left(\left\|\boldsymbol{x}_{\text{test}}^{\mathsf{T}}e^{-\boldsymbol{X}^{\mathsf{T}}\boldsymbol{X}t}(\boldsymbol{\mu}_0 - \boldsymbol{w})\right\|_2^2\right)$$

where we used Equation (S23). We have since $\mathbb{E}(\boldsymbol{w}) = \boldsymbol{\mu}_0$, that the above

$$= \text{tr}\left(\text{Cov}(\boldsymbol{w} - \boldsymbol{\mu}_0)e^{-\boldsymbol{X}^{\mathsf{T}}\boldsymbol{X}t}\boldsymbol{x}_{\text{test}}\boldsymbol{x}_{\text{test}}^{\mathsf{T}}e^{-\boldsymbol{X}^{\mathsf{T}}\boldsymbol{X}t}\right)$$

$$= \text{tr}\left(\boldsymbol{x}_{\text{test}}^{\mathsf{T}}e^{-\boldsymbol{X}^{\mathsf{T}}\boldsymbol{X}t}\boldsymbol{S}_0\boldsymbol{S}_0^{\mathsf{T}}e^{-\boldsymbol{X}^{\mathsf{T}}\boldsymbol{X}t}\boldsymbol{x}_{\text{test}}\right)$$

$$= \mathrm{tr}\big(\boldsymbol{x}_{\text{test}}^{\mathsf{T}} \boldsymbol{S}_t \boldsymbol{S}_t^{\mathsf{T}} \boldsymbol{x}_{\text{test}}\big)$$
$$= \mathrm{Var}_{\boldsymbol{w} \sim q_{\boldsymbol{\theta}(t)}}(f_{\boldsymbol{w}}(\boldsymbol{x}_{\text{test}}))$$

where we used Equation (S24) in the second-to-last equality. This completes the proof. $\qquad\square$

**Theorem S4** (Non-Asymptotic Error of SGD)
*Let $f_{\boldsymbol{w}}(\boldsymbol{x}) = \boldsymbol{x}^{\mathsf{T}} \boldsymbol{w}$ be a linear model. Define a prior $p(\boldsymbol{w}) = \mathcal{N}\big(\boldsymbol{w}; \boldsymbol{\mu}_0, \boldsymbol{S}_0 \boldsymbol{S}_0^{\mathsf{T}}\big)$ and assume noise-free observations $y(\cdot) = f_{\boldsymbol{w}}(\cdot)$ for $\boldsymbol{w} \sim p(\boldsymbol{w})$. Further, define a variational distribution $q_{\boldsymbol{\theta}}(\boldsymbol{w}) = \mathcal{N}\big(\boldsymbol{w}; \boldsymbol{\mu}, \boldsymbol{S}\boldsymbol{S}^{\mathsf{T}}\big)$ with $\boldsymbol{\theta} = (\boldsymbol{\mu}, \boldsymbol{S})$ such that $\boldsymbol{\mu} \in \mathbb{R}^P$ and $\boldsymbol{S} \in \mathbb{R}^{P \times R}$ where $R \le P$. Assume the expected loss is given by $\bar{\ell}(\boldsymbol{\theta}) = \mathbb{E}_{q_{\boldsymbol{\theta}}(\boldsymbol{w})}\big(\frac{1}{2}\|\boldsymbol{y} - \boldsymbol{X}\boldsymbol{w}\|_2^2\big)$ and let $\boldsymbol{\theta}(t) = (\boldsymbol{\mu}(t), \boldsymbol{S}(t))$ be the variational parameters at step $t$ of (stochastic) gradient descent with learning rate sequence $(\eta_t)_t$, initialized at $\boldsymbol{\theta}(0) = (\boldsymbol{\mu}_0, \boldsymbol{S}_0)$. Then the expected squared error of the mean prediction*

$$\mathbb{E}_{\binom{\boldsymbol{y}}{y_{\text{test}}}}\Big(\big(y_{\text{test}} - f_{\boldsymbol{\mu}(t)}(\boldsymbol{x}_{\text{test}})\big)^2\Big) = \mathrm{Var}_{\boldsymbol{w} \sim q_{\boldsymbol{\theta}(t)}}(f_{\boldsymbol{w}}(\boldsymbol{x}_{\text{test}})) \tag{S25}$$

*at any test point $\boldsymbol{x}_{\text{test}} \in \mathbb{R}^P$. In other words, assuming the training and test data are drawn from the prior predictive, the predictive error of $f_{\boldsymbol{\mu}(t)}(\cdot)$ at any optimization step $t$ is exactly quantified by the predictive uncertainty of the variational distribution.*

*Further, if the learning rate $\eta_t \le \frac{1}{\lambda_{\max}(\boldsymbol{X}_t^{\mathsf{T}} \boldsymbol{X}_t)}$ for all steps $t$, then*

$$\mathrm{tr}(\mathrm{Cov}_{\boldsymbol{w} \sim q_{\boldsymbol{\theta}(t+1)}}(\boldsymbol{w})) \le \mathrm{tr}(\mathrm{Cov}_{\boldsymbol{w} \sim q_{\boldsymbol{\theta}(t)}}(\boldsymbol{w})), \tag{S26}$$

*i.e. uncertainty about the parameters decreases monotonically during optimization.*

*Proof.* The expected loss is given up to an additive constant by

$$\bar{\ell}(\boldsymbol{\theta}) = \mathbb{E}_{q_{\boldsymbol{\theta}}(\boldsymbol{w})}(\ell(\boldsymbol{y}, f_{\boldsymbol{w}}(\boldsymbol{X}))) \stackrel{+c}{=} \frac{1}{2}(\|\boldsymbol{y} - \boldsymbol{X}\boldsymbol{\mu}\|_2^2 + \mathrm{tr}(\boldsymbol{X}\boldsymbol{S}\boldsymbol{S}^{\mathsf{T}}\boldsymbol{X}^{\mathsf{T}})).$$

Now let $(\boldsymbol{X}_t, \boldsymbol{y}_t)$ be the minibatch at step $t \ge 1$. Then it holds that

$$f_{\boldsymbol{\mu}(t)}(\boldsymbol{x}_{\text{test}}) - y_{\text{test}} = \boldsymbol{x}_{\text{test}}^{\mathsf{T}}(\boldsymbol{\mu}(t) - \boldsymbol{w}). \tag{S27}$$

Further, the mean parameters identified by SGD are given by

$$\begin{aligned}
\boldsymbol{\mu}(t) - \boldsymbol{w} &= \boldsymbol{\mu}(t-1) - \boldsymbol{w} - \eta_t \nabla_{\boldsymbol{\mu}} \bar{\ell}(\boldsymbol{\theta}(t-1)) \\
&= \boldsymbol{\mu}(t-1) - \boldsymbol{w} - \eta_t \boldsymbol{X}_t^{\mathsf{T}}(\boldsymbol{X}_t \boldsymbol{\mu}(t-1) - \boldsymbol{y}_t) \\
&= \boldsymbol{\mu}(t-1) - \boldsymbol{w} - \eta_t \boldsymbol{X}_t^{\mathsf{T}} \boldsymbol{X}_t(\boldsymbol{\mu}(t-1) - \boldsymbol{w}) \\
&= (\boldsymbol{I} - \eta_t \boldsymbol{X}_t^{\mathsf{T}} \boldsymbol{X}_t)(\boldsymbol{\mu}(t-1) - \boldsymbol{w}) \\
&= \prod_{j=1}^{t}(\boldsymbol{I} - \eta_j \boldsymbol{X}_j^{\mathsf{T}} \boldsymbol{X}_j)(\boldsymbol{\mu}(0) - \boldsymbol{w}) \\
&= \boldsymbol{B}_t(\boldsymbol{\mu}_0 - \boldsymbol{w})
\end{aligned}$$

where we defined $\boldsymbol{B}_t = \prod_{j=1}^{t}(\boldsymbol{I} - \eta_j \boldsymbol{X}_j^{\mathsf{T}} \boldsymbol{X}_j)$. The covariance parameters are given by

$$\begin{aligned}
\boldsymbol{S}(t) &= \boldsymbol{S}(t-1) - \eta_t \nabla_{\boldsymbol{S}} \bar{\ell}(\boldsymbol{\theta}(t-1)) \\
&= \boldsymbol{S}(t-1) - \eta_t \boldsymbol{X}_t^{\mathsf{T}} \boldsymbol{X}_t \boldsymbol{S}(t-1) \\
&= (\boldsymbol{I} - \eta_t \boldsymbol{X}_t^{\mathsf{T}} \boldsymbol{X}_t)\boldsymbol{S}(t-1) \\
&= \prod_{j=1}^{t}(\boldsymbol{I} - \eta_j \boldsymbol{X}_j^{\mathsf{T}} \boldsymbol{X}_j)\boldsymbol{S}(0) \\
&= \boldsymbol{B}_t \boldsymbol{S}_0
\end{aligned}$$

Therefore the predictive error at step $t \in \{0, 1, \dots\}$ is given by

$$\mathbb{E}_{\binom{\boldsymbol{y}}{y_{\text{test}}}}\Big(\big\|y_{\text{test}} - f_{\boldsymbol{\mu}(t)}(\boldsymbol{x}_{\text{test}})\big\|_2^2\Big) = \mathbb{E}_{\binom{\boldsymbol{X}}{\boldsymbol{x}_{\text{test}}}\boldsymbol{w}}\Big(\big\|y_{\text{test}} - \boldsymbol{x}_{\text{test}}^{\mathsf{T}}\boldsymbol{\mu}(t)\big\|_2^2\Big)$$

$$= \mathbb{E}_{\boldsymbol{w}} \left( \left\| \boldsymbol{x}_{\text{test}}^{\mathsf{T}} (\boldsymbol{\mu}(t) - \boldsymbol{w}) \right\|_2^2 \right)$$

$$= \mathbb{E}_{\boldsymbol{w}} \left( \left\| \boldsymbol{x}_{\text{test}}^{\mathsf{T}} \boldsymbol{B}_t (\boldsymbol{\mu}_0 - \boldsymbol{w}) \right\|_2^2 \right).$$

We have since $\mathbb{E}(\boldsymbol{\mu}_0 - \boldsymbol{w}) = \boldsymbol{0}$, that the above

$$= \operatorname{tr}(\boldsymbol{B}_t^{\mathsf{T}} \boldsymbol{x}_{\text{test}} \boldsymbol{x}_{\text{test}}^{\mathsf{T}} \boldsymbol{B}_t \operatorname{Cov}(\boldsymbol{w} - \boldsymbol{\mu}_0))$$

$$= \operatorname{tr}(\boldsymbol{x}_{\text{test}}^{\mathsf{T}} \boldsymbol{B}_t \boldsymbol{S}_0 \boldsymbol{S}_0^{\mathsf{T}} \boldsymbol{B}_t^{\mathsf{T}} \boldsymbol{x}_{\text{test}})$$

$$= \operatorname{tr}(\boldsymbol{x}_{\text{test}}^{\mathsf{T}} \boldsymbol{S}(t) \boldsymbol{S}(t)^{\mathsf{T}} \boldsymbol{x}_{\text{test}})$$

$$= \operatorname{Var}_{\boldsymbol{w} \sim q_{\boldsymbol{\theta}(t)}}(f_{\boldsymbol{w}}(\boldsymbol{x}_{\text{test}})).$$

This proves Equation (S25).

To prove the second statement, we begin by showing that $\boldsymbol{I} - \eta_t \boldsymbol{X}_t^{\mathsf{T}} \boldsymbol{X}_t$ has a spectrum in the interval $[0, 1]$. We have by Weyl's theorem, since $\boldsymbol{I}$ and $\boldsymbol{C}_{t+1} := -\eta_{t+1} \boldsymbol{X}_{t+1}^{\mathsf{T}} \boldsymbol{X}_{t+1}$ are hermitian, that

$$\lambda_p(\boldsymbol{I}) + \lambda_{\min}(\boldsymbol{C}_{t+1}) \leq \lambda_p(\boldsymbol{I} + \boldsymbol{C}_{t+1}) \leq \lambda_p(\boldsymbol{I}) + \lambda_{\max}(\boldsymbol{C}_{t+1})$$

$$\iff 1 - \eta_{t+1} \lambda_{\max}(\boldsymbol{X}_{t+1}^{\mathsf{T}} \boldsymbol{X}_{t+1}) \leq \lambda_p(\boldsymbol{I} + \boldsymbol{C}_{t+1}) \leq 1 - \eta_{t+1} \lambda_{\min}(\boldsymbol{X}_{t+1}^{\mathsf{T}} \boldsymbol{X}_{t+1})$$

$$\implies 1 - \frac{\lambda_{\max}(\boldsymbol{X}_{t+1}^{\mathsf{T}} \boldsymbol{X}_{t+1})}{\lambda_{\max}(\boldsymbol{X}_{t+1}^{\mathsf{T}} \boldsymbol{X}_{t+1})} \leq \lambda_p(\boldsymbol{I} + \boldsymbol{C}_{t+1}) \leq 1$$

$$\iff 0 \leq \lambda_p(\boldsymbol{I} + \boldsymbol{C}_{t+1}) \leq 1$$

where we used the assumption on the learning rate that $\forall t : \eta_t \leq \frac{1}{\lambda_{\max}(\boldsymbol{X}_t^{\mathsf{T}} \boldsymbol{X}_t)}$. Now by von Neumann's trace inequality, it holds that

$$\operatorname{tr}(\operatorname{Cov}_{\boldsymbol{w} \sim q_{\boldsymbol{\theta}(t+1)}}(\boldsymbol{w})) = \operatorname{tr}((\boldsymbol{I} - \eta_{t+1} \boldsymbol{X}_{t+1}^{\mathsf{T}} \boldsymbol{X}_{t+1}) \boldsymbol{S}_t \boldsymbol{S}_t^{\mathsf{T}} (\boldsymbol{I} - \eta_{t+1} \boldsymbol{X}_{t+1}^{\mathsf{T}} \boldsymbol{X}_{t+1})^{\mathsf{T}})$$

$$= \operatorname{tr}(\boldsymbol{S}_t \boldsymbol{S}_t^{\mathsf{T}} (\boldsymbol{I} - \eta_{t+1} \boldsymbol{X}_{t+1}^{\mathsf{T}} \boldsymbol{X}_{t+1})(\boldsymbol{I} - \eta_{t+1} \boldsymbol{X}_{t+1}^{\mathsf{T}} \boldsymbol{X}_{t+1}))$$

$$\leq \sum_{p=1}^{P} \lambda_p(\boldsymbol{S}_t \boldsymbol{S}_t^{\mathsf{T}}) \lambda_p((\boldsymbol{I} - \eta_{t+1} \boldsymbol{X}_{t+1}^{\mathsf{T}} \boldsymbol{X}_{t+1})^2)$$

$$= \sum_{p=1}^{P} \lambda_p(\boldsymbol{S}_t \boldsymbol{S}_t^{\mathsf{T}}) \lambda_p((\boldsymbol{I} - \eta_{t+1} \boldsymbol{X}_{t+1}^{\mathsf{T}} \boldsymbol{X}_{t+1}))^2$$

$$\leq \sum_{p=1}^{P} \lambda_p(\boldsymbol{S}_t \boldsymbol{S}_t^{\mathsf{T}})$$

$$= \operatorname{tr}(\operatorname{Cov}_{\boldsymbol{w} \sim q_{\boldsymbol{\theta}(t)}}(\boldsymbol{w})).$$

$\square$

### S1.1.3 CONNECTION TO ENSEMBLES

**Proposition S1** (Connection to Ensembles)
*Consider an ensemble of overparametrized linear models $f_{\boldsymbol{w}}(\boldsymbol{x}) = \boldsymbol{x}^{\mathsf{T}} \boldsymbol{w}$ initialized with weights drawn from the prior $\boldsymbol{w}_0^{(i)} \sim \mathcal{N}(\boldsymbol{w}; \boldsymbol{\mu}_0, \boldsymbol{S}_0 \boldsymbol{S}_0^{\mathsf{T}})$. Assume each model is trained independently to convergence via (S)GD such that $\boldsymbol{w}_\star^{(i)} = \arg\min_{\boldsymbol{w}} \ell(\boldsymbol{y}, f_{\boldsymbol{w}}(\boldsymbol{X}))$. Then the distribution over the weights of the trained ensemble $q_{\text{Ens}}(\boldsymbol{w})$ is equal to the variational approximation $q_{\boldsymbol{\theta}_\star}(\boldsymbol{w})$ learned via (S)GD initialized at the prior hyperparameters $\boldsymbol{\theta}_0 = (\boldsymbol{\mu}_0, \boldsymbol{S}_0)$, i.e.*

$$q_{\text{Ens}}(\boldsymbol{w}) = q_{\boldsymbol{\theta}_\star^{\text{GD}}}(\boldsymbol{w}). \tag{S28}$$

*Proof.* The parameters $\boldsymbol{w}_\infty^{(i)}$ of the (independently) trained ensemble members identified via (stochastic) gradient descent are given by

$$\boldsymbol{w}_\infty^{(i)} = \arg\min_{\boldsymbol{w} \in F} \| \boldsymbol{w} - \boldsymbol{w}_0^{(i)} \|_2$$

where $F = \{w \in \mathbb{R}^P \mid f_w(X) = Xw = y\}$ is the set of interpolating solutions [5, Sec. 2.1]. Since we can write $F$ equivalently via the minimum norm solution and an arbitrary null space contribution, s.t. $F = \{w = X^\dagger y + w_{\text{null}} \mid w_{\text{null}} \in \text{null}(X)\}$ we have

$$= X^\dagger y + \underset{w_{\text{null}} \in \text{null}(X)}{\arg\min} \|w_{\text{null}} - (w_0^{(i)} - X^\dagger y)\|_2$$

$$= X^\dagger y + \text{proj}_{\text{null}(X)} \left( w_0^{(i)} - \underbrace{X^\dagger y}_{\in \text{range}(X^\mathsf{T})} \right)$$

where we used the characterization of an orthogonal projection onto a linear subspace as the (unique) closest point in the subspace. Finally, we use that the minimum norm solution is in the range space of the data and rewrite the projection in matrix form, s.t.

$$= X^\dagger y + P_{\text{null}} w_0^{(i)}.$$

Therefore the distribution over the parameters $w_\infty^{(i)}$ of the ensemble members computed via (S)GD with initial parameters $w_0 \sim \mathcal{N}(w; \mu_0, S_0 S_0^\mathsf{T})$ is given by

$$q_{\text{Ens}}(w) = \mathcal{N}\left( w; \underbrace{X^\dagger y + P_{\text{null}} \mu_0}_{=\mu_{\text{Ens}}}, \underbrace{P_{\text{null}} S_0 S_0^\mathsf{T} P_{\text{null}}^\mathsf{T}}_{=S_{\text{Ens}}} \right).$$

Now the expected negative log-likelihood of the distribution over the parameters of the trained ensemble members $q_{\text{Ens}}(w)$ with hyperparameters $\theta_{\text{Ens}} = (\mu_{\text{Ens}}, S_{\text{Ens}})$ is

$$\bar{\ell}(\theta_{\text{Ens}}) \overset{+c}{=} \frac{1}{2\sigma^2} \left( \|y - X\mu_{\text{Ens}}\|_2^2 + \text{tr}(X S_{\text{Ens}} S_{\text{Ens}}^\mathsf{T} X^\mathsf{T}) \right) = 0$$

and therefore $\theta_{\text{Ens}}$ is a minimizer of the expected log-likelihood. Further it holds that

$$z = V_{\text{null}}^\mathsf{T} (P_{\text{null}} \mu_0) = V_{\text{null}}^\mathsf{T} \mu_0$$

$$M = V_{\text{null}}^\mathsf{T} (P_{\text{null}} S_0)(P_{\text{null}} S_0)^\mathsf{T} V_{\text{null}} = V_{\text{null}}^\mathsf{T} S_0 S_0^\mathsf{T} V_{\text{null}} = V_{\text{null}}^\mathsf{T} \Sigma_0 V_{\text{null}}$$

and thus by Equation (S19), the distribution of the trained ensemble parameters minimizes the 2-Wasserstein distance to the prior distribution, i.e.

$$q_{\text{Ens}} = \underset{q(w)=\mathcal{N}(w;\mu,\Sigma)}{\arg\min} W_2^2(q(w), \mathcal{N}(w; \mu_0, \Sigma_0)).$$

Combining this with the characterization of the variational posterior in Theorem 1 proves the claim. $\square$

## S1.2 BINARY CLASSIFICATION OF LINEARLY SEPARABLE DATA

In this subsection we provide proofs of claims from Section 4.2. We begin with presenting some preliminary results from Soudry et al. [4] which will be used throughout the proof. Next, we will analyze the gradient flow of the expected loss. We extend the results for the gradient flow to gradient descent and derive the characterization of the implicit bias, completing the proof of Theorem 2.

**Theorem 2** (Implicit Bias in Binary Classification)
*Let $f_w(x) = x^\mathsf{T} w$ be an (overparametrized) linear model and define a Gaussian prior $p(w) = \mathcal{N}(w; \mu_0, S_0 S_0^\mathsf{T})$. Assume a variational distribution $q_\theta(w) = \mathcal{N}(w; \mu, SS^\mathsf{T})$ over the weights $w \in \mathbb{R}^P$ with variational parameters $\theta = (\mu, S)$ such that $S \in \mathbb{R}^{P \times R}$ and $R \le P$. Assume we are using the exponential loss $\ell(u) = \exp(-u)$ and optimize the expected empirical loss $\bar{\ell}(\theta)$ via gradient descent initialized at the prior, i.e. $\theta_0 = (\mu_0, S_0)$, with a sufficiently small learning rate $\eta$. Then for almost any dataset which is linearly separable (Assumption 1) and for which the support vectors span the data (Assumption 2), the rescaled gradient descent iterates (rGD)*

$$\theta_t^{\text{rGD}} = (\mu_t^{\text{rGD}}, S_t^{\text{rGD}}) = \left( \frac{1}{\log(t)} \mu_t^{\text{GD}} + P_{\text{null}(X)} \mu_0, S_t^{\text{GD}} \right) \tag{9}$$

*converge to a limit point $\theta_\star^{\text{rGD}} = \lim_{t\to\infty} \theta_t^{\text{rGD}}$ for which it holds that*

$$\theta_\star^{\text{rGD}} \in \underset{\substack{\theta=(\mu,S) \\ s.t. \, \theta \in \Theta_\star}}{\arg\min} W_2^2(q_\theta, p), \tag{10}$$

*where the feasible set $\Theta_\star = \{(\mu, S) \mid P_{\text{range}(X^\mathsf{T})} \mu = \hat{w} \text{ and } \forall n : \text{Var}_{q_\theta}(f_w(x_n)) = 0\}$ consists of mean parameters which, if projected onto the training data, are equivalent to the $L_2$ max margin vector and covariance parameters such that there is no uncertainty at training data.*

### S1.2.1 PRELIMINARIES

Recall that the expected loss is given by

$$\bar{\ell}(\boldsymbol{\theta}) = \sum_{n=1}^{N} \mathbb{E}_{q_{\boldsymbol{\theta}}(\boldsymbol{w})}\big(\ell(y_n \boldsymbol{x}_n^{\mathsf{T}} \boldsymbol{w})\big), \tag{S29}$$

and specifically, for the exponential loss, we have

$$\bar{\ell}(\boldsymbol{\theta}) = \bar{\ell}(\boldsymbol{\mu}, \boldsymbol{S}) = \sum_{n=1}^{N} \exp\left(-\boldsymbol{x}_n^{\mathsf{T}} \boldsymbol{\mu} + \tfrac{1}{2} \boldsymbol{x}_n^{\mathsf{T}} \boldsymbol{S} \boldsymbol{S}^{\mathsf{T}} \boldsymbol{x}_n\right). \tag{S30}$$

Throughout these proofs, for any mean parameter iterate $\boldsymbol{\mu}_t$, we define the residual as

$$\boldsymbol{r}_t = \boldsymbol{\mu}_t - \hat{\boldsymbol{w}} \log t - \tilde{\boldsymbol{w}} \tag{S31}$$

where $\hat{\boldsymbol{w}}$ is the solution to the hard margin SVM, and $\tilde{\boldsymbol{w}}$ is the vector which satisfies

$$\forall n \in \mathcal{S} : \eta \exp\left(-\boldsymbol{x}_n^{\mathsf{T}} \tilde{\boldsymbol{w}}\right) = \alpha_n, \tag{S32}$$

where weights $\alpha_n$ are defined through the KKT conditions on the hard margin SVM problem, i.e.

$$\hat{\boldsymbol{w}} = \sum_{n \in \mathcal{S}} \alpha_n \boldsymbol{x}_n. \tag{S33}$$

In Lemma 12 (Appendix B) of Soudry et al. [4], it is shown that, for almost any dataset, there are no more than $P$ support vectors and $\alpha_n \neq 0, \forall n \in \mathcal{S}$. Furthermore, we denote the minimum margin to a non-support vector as:

$$\kappa = \min_{n \notin \mathcal{S}} \boldsymbol{x}_n^{\mathsf{T}} \hat{\boldsymbol{w}} > 1. \tag{S34}$$

Finally, we define $\boldsymbol{P}_{\mathcal{S}} \in \mathbb{R}^{P \times P}$ as the orthogonal projection matrix to the subspace spanned by the support vectors, and $\bar{\boldsymbol{P}}_{\mathcal{S}} = \boldsymbol{I} - \boldsymbol{P}_{\mathcal{S}}$ as the complementary projection.

### S1.2.2 GRADIENT FLOW FOR THE EXPECTED LOSS

Similar as in Soudry et al. [4], we begin by studying the gradient flow dynamics, i.e. taking the continuous time limit of gradient descent:

$$\dot{\boldsymbol{\theta}}_t = -\nabla \bar{\ell}(\boldsymbol{\theta}_t), \tag{S35}$$

which can be written componentwise as:

$$\dot{\boldsymbol{\mu}}_t = -\nabla_{\boldsymbol{\mu}} \bar{\ell}(\boldsymbol{\mu}_t, \boldsymbol{S}_t) = \sum_{n=1}^{N} \exp\left(-\boldsymbol{\mu}_t^{\mathsf{T}} \boldsymbol{x}_n + \frac{1}{2} \boldsymbol{x}_n^{\mathsf{T}} \boldsymbol{S}_t \boldsymbol{S}_t^{\mathsf{T}} \boldsymbol{x}_n\right) \boldsymbol{x}_n \tag{S36}$$

$$\dot{\boldsymbol{S}}_t = -\nabla_{\boldsymbol{S}} \bar{\ell}(\boldsymbol{\mu}_t, \boldsymbol{S}_t) = -\sum_{n=1}^{N} \exp\left(-\boldsymbol{\mu}_t^{\mathsf{T}} \boldsymbol{x}_n + \frac{1}{2} \boldsymbol{x}_n^{\mathsf{T}} \boldsymbol{S}_t \boldsymbol{S}_t^{\mathsf{T}} \boldsymbol{x}_n\right) \boldsymbol{x}_n \boldsymbol{x}_n^{\mathsf{T}} \boldsymbol{S}_t. \tag{S37}$$

We begin by showing that the total uncertainty, as measured by the Frobenius norm of the covariance factor, is bounded during the gradient flow dynamics. To that end, we derive the following dynamics:

$$\frac{d}{dt} \frac{1}{2} \|\boldsymbol{S}_t\|_F^2 = \operatorname{tr}(\boldsymbol{S}_t^{\mathsf{T}} \dot{\boldsymbol{S}}_t) = -\sum_{n=1}^{N} \exp\left(-\boldsymbol{\mu}_t^{\mathsf{T}} \boldsymbol{x}_n + \frac{1}{2} \boldsymbol{x}_n^{\mathsf{T}} \boldsymbol{S}_t \boldsymbol{S}_t^{\mathsf{T}} \boldsymbol{x}_n\right) \|\boldsymbol{x}_n^{\mathsf{T}} \boldsymbol{S}_t\|^2 \leq 0, \tag{S38}$$

and therefore

$$\|\boldsymbol{S}_t\|_F^2 \leq \|\boldsymbol{S}_0\|_F^2. \tag{S39}$$

Finally, by Cauchy-Schwarz inequality, we have that

$$\|\boldsymbol{S}_t \boldsymbol{S}_t^{\mathsf{T}}\|_F \leq \|\boldsymbol{S}_t\|_F^2 \leq \|\boldsymbol{S}_0\|_F^2. \tag{S40}$$

We continue by studying the convergence behavior of the mean parameter $\boldsymbol{\mu}_t$.

**Mean parameter** Our goal is to show that $\|r_t\|$ is bounded. Equation (S31) implies that

$$\dot{r}_t = \dot{\mu}_t - \frac{1}{t}\hat{w} = -\nabla_\mu \bar{\ell}(\mu_t, S_t) - \frac{1}{t}\hat{w}. \tag{S41}$$

This in turn implies that

$$
\begin{aligned}
\frac{1}{2}\frac{d}{dt}\|r_t\|^2 &= \dot{r}_t^\mathsf{T} r_t \\
&= \sum_{n=1}^{N} \exp\left(-\mu_t^\mathsf{T} x_n + \frac{1}{2} x_n^\mathsf{T} S_t S_t^\mathsf{T} x_n\right) x_n^\mathsf{T} r_t - \frac{1}{t}\hat{w}^\mathsf{T} r_t \\
&= \sum_{n\in\mathcal{S}} \exp\left(-\log(t)\hat{w}^\mathsf{T} x_n - \tilde{w}^\mathsf{T} x_n + \frac{1}{2} x_n^\mathsf{T} S_t S_t^\mathsf{T} x_n - x_n^\mathsf{T} r_t\right) x_n^\mathsf{T} r_t - \frac{1}{t}\hat{w}^\mathsf{T} r_t \\
&\quad + \sum_{n\notin\mathcal{S}} \exp\left(-\log(t)\hat{w}^\mathsf{T} x_n - \tilde{w}^\mathsf{T} x_n + \frac{1}{2} x_n^\mathsf{T} S_t S_t^\mathsf{T} x_n - x_n^\mathsf{T} r_t\right) x_n^\mathsf{T} r_t \\
&= \left[\frac{1}{t}\sum_{n\in\mathcal{S}} \exp\left(-\tilde{w}^\mathsf{T} x_n\right)\left(\exp\left(-x_n^\mathsf{T} r_t + \frac{1}{2} x_n^\mathsf{T} S_t S_t^\mathsf{T} x_n\right) - 1\right) x_n^\mathsf{T} r_t\right] \\
&\quad + \left[\sum_{n\notin\mathcal{S}} \left(\frac{1}{t}\right)^{\hat{w}^\mathsf{T} x_n} \exp\left(-\tilde{w}^\mathsf{T} x_n + \frac{1}{2} x_n^\mathsf{T} S_t S_t^\mathsf{T} x_n\right) \exp\left(-x_n^\mathsf{T} r_t\right) x_n^\mathsf{T} r_t\right].
\end{aligned}
\tag{S42}
$$

where in last line we used the fact that $\hat{w}^\mathsf{T} x_n = 1$ for $n \in \mathcal{S}$ as in (S32), and that $\sum_{n\in\mathcal{S}} \exp(-x_n^\mathsf{T}\tilde{w})x_n = \hat{w}$ as in (S33). We begin by examining the first bracket, studying three possible cases for each of the summands. First, note that if $x_n^\mathsf{T} r_t \leq 0$, then since $\frac{1}{2} x_n^\mathsf{T} S_t S_t^\mathsf{T} x_n \geq 0$, we have that

$$\left(\exp\left(-x_n^\mathsf{T} r_t + \frac{1}{2} x_n^\mathsf{T} S_t S_t^\mathsf{T} x_n\right) - 1\right) x_n^\mathsf{T} r_t \leq 0. \tag{S43}$$

Next, by defining $B := \|S_0\|_F^2 \max_n \|x_n\|_2$, if $0 < x_n^\mathsf{T} r_t < \frac{B}{2}$, we have that

$$\left|\left(\exp\left(-x_n^\mathsf{T} r_t + \frac{1}{2} x_n^\mathsf{T} S_t S_t^\mathsf{T} x_n\right) - 1\right) x_n^\mathsf{T} r_t\right| < \left(\exp\left(\frac{B}{2}\right) - 1\right)\frac{B}{2}, \tag{S44}$$

and if $x_n^\mathsf{T} r_t \geq \frac{B}{2}$, we have that

$$\left(\exp\left(-x_n^\mathsf{T} r_t + \frac{1}{2} x_n^\mathsf{T} S_t S_t^\mathsf{T} x_n\right) - 1\right) x_n^\mathsf{T} r_t \leq 0. \tag{S45}$$

Finally, for arbitrary $\epsilon \geq \max\{B, 1\}$, if $|x_n^\mathsf{T} r_t| \geq \epsilon$, we have that

$$\left(\exp\left(-x_n^\mathsf{T} r_t + \frac{1}{2} x_n^\mathsf{T} S_t S_t^\mathsf{T} x_n\right) - 1\right) x_n^\mathsf{T} r_t \leq \left(\exp\left(-\frac{B}{2}\right) - 1\right)\epsilon < 0. \tag{S46}$$

Furthermore, let $\gamma_* = \min_{n\in\mathcal{S}} \tilde{w}^\mathsf{T} x_n$ and $\gamma^* = \max_{n\in\mathcal{S}} \tilde{w}^\mathsf{T} x_n$. Now, by taking $\epsilon \geq \max\{B, 1\}$ large enugh such that

$$\left|\exp(-\gamma^*)\left(\exp\left(-\frac{B}{2}\right) - 1\right)\epsilon\right| \geq |\mathcal{S}|\exp(-\gamma_*)\left(\exp\left(\frac{B}{2}\right) - 1\right)\frac{B}{2}, \tag{S47}$$

if there exists a support vector $n \in \mathcal{S}$ such that $|x_n^\mathsf{T} r_t| \geq \epsilon$, then

$$\frac{1}{t}\sum_{n\in\mathcal{S}} \exp\left(-\tilde{w}^\mathsf{T} x_n\right)\left(\exp\left(-x_n^\mathsf{T} r_t + \frac{1}{2} x_n^\mathsf{T} S_t S_t^\mathsf{T} x_n\right) - 1\right) x_n^\mathsf{T} r_t \leq 0. \tag{S48}$$

The idea of this is that if there exists a support vector such that $|x_n^\mathsf{T} r_t|$ is sufficiently big, then the first bracket in Eq. (S42) is negative.

On the other hand, for the second bracket in Eq. (S42), note that for $n \notin \mathcal{S}$, we have that $\boldsymbol{x}_n^\top \hat{\boldsymbol{w}} \geq \kappa$, and hence

$$
\sum_{n \notin \mathcal{S}} \left(\frac{1}{t}\right)^{\hat{\boldsymbol{w}}^\top \boldsymbol{x}_n} \exp\left(-\tilde{\boldsymbol{w}}^\top \boldsymbol{x}_n + \frac{1}{2} \boldsymbol{x}_n^\top \boldsymbol{S}_t \boldsymbol{S}_t^\top \boldsymbol{x}_n\right) \exp\left(-\boldsymbol{x}_n^\top \boldsymbol{r}_t\right) \boldsymbol{x}_n^\top \boldsymbol{r}_t
$$
$$
\leq \frac{1}{t^\kappa} \exp\left(\frac{1}{2}\|\boldsymbol{S}_0\|_F^2 \max_n \boldsymbol{x}_n^\top \boldsymbol{x}_n\right) \sum_{n \notin \mathcal{S}} \exp\left(-\tilde{\boldsymbol{w}}^\top \boldsymbol{x}_n\right) = \mathcal{O}\left(\frac{1}{t^\kappa}\right), \tag{S49}
$$

where in the last line we used that $ze^{-z} \leq 1, \forall z \in \mathbb{R}$ and fact that $\|\boldsymbol{S}_t \boldsymbol{S}_t^\top\|_F \leq \|\boldsymbol{S}_0\|_F^2 < \infty$.

We will now combine the results from above to show that the residual $\boldsymbol{r}_t$ is bounded in the following way: if there exists a support vector $n \in \mathcal{S}$ such that $|\boldsymbol{x}_n^\top \boldsymbol{r}_t| \geq \epsilon$ for big enough $\epsilon > 0$, then $\frac{1}{2}\frac{d}{dt}\|\boldsymbol{r}_t\|^2 = \mathcal{O}(t^{-\kappa})$. If such a support vector does not exist at time $t$, we will show that $\boldsymbol{r}_t$ is containted inside a compact set. To that end, if $\|\boldsymbol{P}_\mathcal{S} \boldsymbol{r}_t\| \geq \epsilon_1$, we have that

$$
\max_{n \in \mathcal{S}} \left|\boldsymbol{x}_n^\top \boldsymbol{r}_t\right|^2 \geq \frac{1}{|\mathcal{S}|} \sum_{n \in \mathcal{S}} \left|\boldsymbol{x}_n^\top \boldsymbol{P}_\mathcal{S} \boldsymbol{r}_t\right|^2 = \frac{1}{|\mathcal{S}|} \left\|\boldsymbol{X}_\mathcal{S}^\top \boldsymbol{P}_\mathcal{S} \boldsymbol{r}_t\right\|^2 \geq \frac{1}{|\mathcal{S}|} \sigma_{\min}^2(\boldsymbol{X}_\mathcal{S}) \epsilon_1^2, \tag{S50}
$$

where in the first inequality we used the fact that $\boldsymbol{P}_\mathcal{S}^\top \boldsymbol{x}_n = \boldsymbol{x}_n$ for $n \in \mathcal{S}$. Hence by choosing $\epsilon_1$ such that $\sigma_{\min}^2(\boldsymbol{X}_\mathcal{S}) \epsilon_1^2 / |\mathcal{S}| = \epsilon^2$, where the $\epsilon$ is chosen in Eq. (S47), we have that

$$
\|\boldsymbol{P}_\mathcal{S} \boldsymbol{r}_t\| \geq \epsilon_1 \Rightarrow \frac{1}{2}\frac{d}{dt}\|\boldsymbol{r}_t\|^2 = \mathcal{O}\left(t^{-\kappa}\right). \tag{S51}
$$

On the other hand, if $\|\boldsymbol{P}_\mathcal{S} \boldsymbol{r}_t\| \leq \epsilon_1$, recall that

$$
\boldsymbol{r}_t = (\boldsymbol{\mu}_t - \boldsymbol{\mu}_0) + \boldsymbol{\mu}_0 - \hat{\boldsymbol{w}} \log t - \tilde{\boldsymbol{w}}, \tag{S52}
$$

and since all updates to the mean parameter are in the space spanned by the support vectors (Assumption 2), we have that

$$
\bar{\boldsymbol{P}}_\mathcal{S} \boldsymbol{r}_t = \bar{\boldsymbol{P}}_\mathcal{S} \boldsymbol{\mu}_0 - \bar{\boldsymbol{P}}_\mathcal{S} \tilde{\boldsymbol{w}}. \tag{S53}
$$

We can now conclude that

$$
\|\boldsymbol{P}_\mathcal{S} \boldsymbol{r}_t\| \leq \epsilon_1 \Rightarrow \|\boldsymbol{r}_t\| \leq \|\boldsymbol{P}_\mathcal{S} \boldsymbol{r}_t\| + \|\bar{\boldsymbol{P}}_\mathcal{S} \boldsymbol{r}_t\| \leq \epsilon_1 + \|\bar{\boldsymbol{P}}_\mathcal{S} \boldsymbol{\mu}_0\| + \|\bar{\boldsymbol{P}}_\mathcal{S} \tilde{\boldsymbol{w}}\| < \infty. \tag{S54}
$$

Finally, combining the results from Eq. (S49) and Eq. (S54), recalling that $\kappa > 1$, we have that $\|\boldsymbol{r}_t\|$ is bounded for all $t > 0$. This completes the first part of the proof and shows that

$$
\boldsymbol{\mu}_t = \hat{\boldsymbol{w}} \log t + \tilde{\boldsymbol{w}} + \boldsymbol{r}_t = \hat{\boldsymbol{w}} \log t + \mathcal{O}(1), \tag{S55}
$$

and in particular

$$
\lim_{t \to \infty} \frac{\boldsymbol{\mu}_t}{\|\boldsymbol{\mu}_t\|} = \frac{\hat{\boldsymbol{w}}}{\|\hat{\boldsymbol{w}}\|}. \tag{S56}
$$

We proceed by showing that the limit covariance parameter vanishes in the span of the support vectors.

**Covariance parameter** We begin by substituting the definition of the residual $\boldsymbol{r}_t$ (S31) into the gradient flow dynamics for the covariance factor $\boldsymbol{S}_t$:

$$
\dot{\boldsymbol{S}}_t = -\nabla_{\boldsymbol{S}} \bar{\ell}(\boldsymbol{\mu}_t, \boldsymbol{S}_t)
$$
$$
= -\sum_{n=1}^N \exp\left(-\boldsymbol{\mu}_t^\top \boldsymbol{x}_n + \frac{1}{2}\boldsymbol{x}_n^\top \boldsymbol{S}_t \boldsymbol{S}_t^\top \boldsymbol{x}_n\right) \boldsymbol{x}_n \boldsymbol{x}_n^\top \boldsymbol{S}_t. \tag{S57}
$$

Next, we split the sum into contributions from support vectors and non–support vectors. For $n \in \mathcal{S}$, we use the property $\boldsymbol{x}_n^\top \hat{\boldsymbol{w}} = 1$; for $n \notin \mathcal{S}$, the margin is strictly larger than one, which introduces higher–order decay in $t$:

$$
\dot{\boldsymbol{S}}_t = -\sum_{n \in \mathcal{S}} \frac{1}{t} \exp\left(-\tilde{\boldsymbol{w}}^\top \boldsymbol{x}_n - \boldsymbol{r}_t^\top \boldsymbol{x}_n\right) \exp\left(\frac{1}{2}\boldsymbol{x}_n^\top \boldsymbol{S}_t \boldsymbol{S}_t^\top \boldsymbol{x}_n\right) \boldsymbol{x}_n \boldsymbol{x}_n^\top \boldsymbol{S}_t
$$
$$
- \sum_{n \notin \mathcal{S}} \left(\frac{1}{t}\right)^{\boldsymbol{x}_n^\top \hat{\boldsymbol{w}}} \exp\left(-\tilde{\boldsymbol{w}}^\top \boldsymbol{x}_n - \boldsymbol{r}_t^\top \boldsymbol{x}_n\right) \exp\left(\frac{1}{2}\boldsymbol{x}_n^\top \boldsymbol{S}_t \boldsymbol{S}_t^\top \boldsymbol{x}_n\right) \boldsymbol{x}_n \boldsymbol{x}_n^\top \boldsymbol{S}_t. \tag{S58}
$$

Since $\boldsymbol{r}_t$ is bounded (from the previous part of the proof), the exponential prefactor is uniformly bounded away from zero. We formalize this by defining

$$C := \min_{n \in [N]} \min_{t \geq 0} \exp\left(-\tilde{\boldsymbol{w}}^\mathsf{T}\boldsymbol{x}_n - \boldsymbol{r}_t^\mathsf{T}\boldsymbol{x}_n\right) > 0. \tag{S59}$$

We also let $\sigma_{\min}$ denote the smallest non–zero eigenvalue of the matrix $\sum_{n \in \mathcal{S}} \boldsymbol{x}_n \boldsymbol{x}_n^\mathsf{T}$. Finally, to measure the size of $\boldsymbol{S}_t$ restricted to the support–vector subspace, we define

$$\Delta_t := \mathrm{tr}\left(\boldsymbol{P}_\mathcal{S}\boldsymbol{S}_t\boldsymbol{S}_t^\mathsf{T}\boldsymbol{P}_\mathcal{S}\right).$$

We now compute the derivative of $\Delta_t$ over time. Differentiating and substituting the dynamics of $\boldsymbol{S}_t$ yields

$$\begin{aligned}
\tfrac{1}{2}\tfrac{d}{dt}\Delta_t &= \mathrm{tr}\left(\boldsymbol{P}_\mathcal{S}\dot{\boldsymbol{S}}_t\boldsymbol{S}_t^\mathsf{T}\boldsymbol{P}_\mathcal{S}\right) \\
&= -\frac{1}{t}\sum_{n \in \mathcal{S}} \exp\left(-\tilde{\boldsymbol{w}}^\mathsf{T}\boldsymbol{x}_n - \boldsymbol{r}_t^\mathsf{T}\boldsymbol{x}_n\right) \exp\left(\tfrac{1}{2}\boldsymbol{x}_n^\mathsf{T}\boldsymbol{S}_t\boldsymbol{S}_t^\mathsf{T}\boldsymbol{x}_n\right) \mathrm{tr}\left(\boldsymbol{P}_\mathcal{S}\boldsymbol{x}_n\boldsymbol{x}_n^\mathsf{T}\boldsymbol{S}_t\boldsymbol{S}_t^\mathsf{T}\boldsymbol{P}_\mathcal{S}\right) \\
&\quad + \mathcal{O}\left(\tfrac{1}{t^\kappa}\right).
\end{aligned} \tag{S60}$$

At this point we use two facts: 1. from (S59), the exponential prefactor is bounded below by $C > 0$, 2. from the definition of $\sigma_{\min}$, we can control the quadratic form $\sum_{n \in \mathcal{S}} \boldsymbol{x}_n \boldsymbol{x}_n^\mathsf{T}$. Applying both gives

$$\tfrac{1}{2}\tfrac{d}{dt}\Delta_t \leq -\frac{C\sigma_{\min}}{t}\Delta_t + \mathcal{O}\left(\tfrac{1}{t^\kappa}\right). \tag{S61}$$

Finally, by Grönwall's lemma, there exists a constant $K > 0$ and a starting time $t_0 > 0$ such that

$$\Delta_t \leq \Delta_{t_0}\left(\frac{t}{t_0}\right)^{-2C\sigma_{\min}} + \frac{K}{2C\sigma_{\min} + \kappa - 1}\, t^{-(\kappa-1)}, \quad \forall t \geq t_0. \tag{S62}$$

Since both $|\mathcal{S}|C\sigma_{\min} > 0$ and $\kappa > 1$, we conclude that $\Delta_t \to 0$ as $t \to \infty$. In words: the covariance factor vanishes when projected onto the span of the support vectors, i.e.

$$\forall n \in \mathcal{S} : \lim_{t \to \infty} \boldsymbol{x}_n^\mathsf{T}\boldsymbol{S}_t\boldsymbol{S}_t^\mathsf{T}\boldsymbol{x}_n = 0, \tag{S63}$$

as claimed. $\qquad \square$

### S1.2.3 Complete Proof of Theorem 2

We will now extend the results for the gradient flow to gradient descent and then use these results to characterize the implicit bias of gradient descent as generalized variational inference.

Throughout this proof, let

$$\boldsymbol{A}_t = \sum_{n=1}^N \exp\left(-\boldsymbol{\mu}_t^\mathsf{T}\boldsymbol{x}_n + \frac{1}{2}\boldsymbol{x}_n^\mathsf{T}\boldsymbol{S}_t\boldsymbol{S}_t^\mathsf{T}\boldsymbol{x}_n\right)\boldsymbol{x}_n\boldsymbol{x}_n^\mathsf{T} \tag{S64}$$

be a positive definite matrix at iteration $t$. We begin the section with a few lemmata which will be used throughout the proof.

**Lemma S2**
*Suppose that we start gradient descent from $(\boldsymbol{\mu}_0, \boldsymbol{S}_0)$. If $\eta < \lambda_{\max}(\boldsymbol{A}_0)^{-1}$, then for the gradient descent iterates*

$$\boldsymbol{S}_{t+1} = \boldsymbol{S}_t - \eta\nabla_{\boldsymbol{S}}\bar{\ell}(\boldsymbol{\mu}_t, \boldsymbol{S}_t), \tag{S65}$$

*we have that $\|\boldsymbol{S}_t\|_F \leq \|\boldsymbol{S}_0\|_F$ for all $t \geq 0$.*

*Proof.* First, note that the gradient descent update for the covariance factor is given by

$$\boldsymbol{S}_{t+1} = \boldsymbol{S}_t(\boldsymbol{I} - \eta\boldsymbol{A}_t), \tag{S66}$$

and hence we have that

$$\|\boldsymbol{S}_{t+1}\|_F = \|\boldsymbol{S}_t(\boldsymbol{I} - \eta\boldsymbol{A}_t)\|_F \leq \|\boldsymbol{S}_t\|_F\|(\boldsymbol{I} - \eta\boldsymbol{A}_t)\|_2. \tag{S67}$$

Now, since $\eta \leq \lambda_{\max}(\boldsymbol{A}_0)^{-1} \leq \lambda_{\max}(\boldsymbol{A}_t)^{-1}$ for all $t \geq 0$ and noting that $\boldsymbol{A}_t \succeq 0$, we have that

$$\|(\boldsymbol{I} - \eta\boldsymbol{A}_t)\|_2 \leq 1, \tag{S68}$$

and therefore

$$\|\boldsymbol{S}_{t+1}\|_F \leq \|\boldsymbol{S}_t\|_F. \tag{S69}$$

Finally, we can conclude that $\|\boldsymbol{S}_t\|_F \leq \|\boldsymbol{S}_0\|_F$ for all $t \geq 0$, as required.

$\square$

**Lemma S3**
*Suppose that we start gradient descent from $(\boldsymbol{\mu}_0, \boldsymbol{S}_0)$. If $\eta < \lambda_{\max}(\boldsymbol{A}_0)^{-1}$, then for the gradient descent iterates*

$$\boldsymbol{\mu}_{t+1} = \boldsymbol{\mu}_t - \eta\nabla_{\boldsymbol{\mu}}\bar{\ell}(\boldsymbol{\mu}_t, \boldsymbol{S}_t), \tag{S70}$$

*we have that $\sum_{u=0}^{\infty} \|\nabla_{\boldsymbol{\mu}}\bar{\ell}(\boldsymbol{\mu}_u, \boldsymbol{S}_u)\|^2 < \infty$. Consequently, we also have that $\lim_{t\to\infty} \|\nabla_{\boldsymbol{\mu}}\bar{\ell}(\boldsymbol{\mu}_t, \boldsymbol{S}_t)\|^2 = 0$.*

*Proof.* Note that our loss function is not globally smooth in $\boldsymbol{\mu}$. However, if we initialize at $(\boldsymbol{\mu}_0, \boldsymbol{S}_0)$, the gradient descent iterates with $\eta < \lambda_{\max}(\boldsymbol{A}_0)^{-1}$ maintain bounded local smoothness. The statement now follows directly from Lemma 10 in Soudry et al. [4]. $\square$

**Lemma S4**
*By choosing $\epsilon_1$ as in Eq. (S51), if $\|\boldsymbol{P}_{\mathcal{S}}\boldsymbol{r}_t\| \geq \epsilon_1$, we have that*

$$(\boldsymbol{r}_{t+1} - \boldsymbol{r}_t)^{\mathsf{T}}\boldsymbol{r}_t \leq \mathcal{O}\left(\frac{1}{t^{\kappa}}\right) + \mathcal{O}\left(\frac{1}{t^2}\right)\|\boldsymbol{r}_t\|. \tag{S71}$$

*If $\|\boldsymbol{P}_{\mathcal{S}}\boldsymbol{r}_t\| < \epsilon_1$, there exists a constant $C$ such that*

$$(\boldsymbol{r}_{t+1} - \boldsymbol{r}_t)^{\mathsf{T}}\boldsymbol{r}_t \leq C. \tag{S72}$$

*Proof.* We follow similar steps as in the gradient flow case. It holds that

$$\begin{aligned}
&(\boldsymbol{r}_{t+1} - \boldsymbol{r}_t)^{\mathsf{T}}\boldsymbol{r}_t \\
&= (-\eta\nabla_{\boldsymbol{\mu}}(\boldsymbol{\mu}_t, \boldsymbol{S}_t) - \hat{\boldsymbol{w}}(\log(t+1) - \log(t)))^{\mathsf{T}}\boldsymbol{r}_t \\
&= \eta\sum_{n=1}^{N}\exp\left(-\boldsymbol{\mu}_t^{\mathsf{T}}\boldsymbol{x}_n + \frac{1}{2}\boldsymbol{x}_n^{\mathsf{T}}\boldsymbol{S}_t\boldsymbol{S}_t^{\mathsf{T}}\boldsymbol{x}_n\right)\boldsymbol{x}_n^{\mathsf{T}}\boldsymbol{r}_t - \hat{\boldsymbol{w}}^{\mathsf{T}}\boldsymbol{r}_t\log(1 + t^{-1}) \\
&= \hat{\boldsymbol{w}}^{\mathsf{T}}\boldsymbol{r}_t(t^{-1} - \log(1 + t^{-1})) + \eta\sum_{n\notin\mathcal{S}}\exp\left(-\boldsymbol{\mu}_t^{\mathsf{T}}\boldsymbol{x}_n + \frac{1}{2}\boldsymbol{x}_n^{\mathsf{T}}\boldsymbol{S}_t\boldsymbol{S}_t^{\mathsf{T}}\boldsymbol{x}_n\right)\boldsymbol{x}_n^{\mathsf{T}}\boldsymbol{r}_t \\
&\quad + \eta\sum_{n\in\mathcal{S}}\left[-\frac{1}{t}\exp\left(-\tilde{\boldsymbol{w}}^{\mathsf{T}}\boldsymbol{x}_n\right) + \exp\left(-\boldsymbol{\mu}_t^{\mathsf{T}}\boldsymbol{x}_n + \frac{1}{2}\boldsymbol{x}_n^{\mathsf{T}}\boldsymbol{S}_t\boldsymbol{S}_t^{\mathsf{T}}\boldsymbol{x}_n\right)\right]\boldsymbol{x}_n^{\mathsf{T}}\boldsymbol{r}_t,
\end{aligned} \tag{S73}$$

where in the last equality we used Equation (S33) to expand $\hat{\boldsymbol{w}}^{\mathsf{T}}\boldsymbol{r}_t$. Furthermore, we can bound all four terms as follows, beginning with the first term:

$$\hat{\boldsymbol{w}}^{\mathsf{T}}\boldsymbol{r}_t(t^{-1} - \log(1 + t^{-1})) \leq \|\boldsymbol{r}_t\|\mathcal{O}\left(\frac{1}{t^2}\right), \tag{S74}$$

where we used that $\log(1 + t^{-1}) = t^{-1} + \mathcal{O}(t^{-2})$. For the second term, using the same argument as in Equation (S49), we derive that

$$\eta\sum_{n\notin\mathcal{S}}\exp\left(-\boldsymbol{\mu}_t^{\mathsf{T}}\boldsymbol{x}_n + \frac{1}{2}\boldsymbol{x}_n^{\mathsf{T}}\boldsymbol{S}_t\boldsymbol{S}_t^{\mathsf{T}}\boldsymbol{x}_n\right)\boldsymbol{x}_n^{\mathsf{T}}\boldsymbol{r}_t \leq \mathcal{O}\left(\frac{1}{t^{\kappa}}\right). \tag{S75}$$

For the third item, from Eq. (S48) and Eq. (S50), we have that $\|\boldsymbol{P}_{\mathcal{S}}\boldsymbol{r}_t\| \geq \epsilon_1$ implies that

$$\eta\sum_{n\in\mathcal{S}}\left[-\frac{1}{t}\exp\left(-\tilde{\boldsymbol{w}}^{\mathsf{T}}\boldsymbol{x}_n\right) + \exp\left(-\boldsymbol{\mu}_t^{\mathsf{T}}\boldsymbol{x}_n + \frac{1}{2}\boldsymbol{x}_n^{\mathsf{T}}\boldsymbol{S}_t\boldsymbol{S}_t^{\mathsf{T}}\boldsymbol{x}_n\right)\right]\boldsymbol{x}_n^{\mathsf{T}}\boldsymbol{r}_t \leq 0. \tag{S76}$$

The first result follows from combining the above three inequalities.

Next, if $\|\boldsymbol{P}_{\mathcal{S}} \boldsymbol{r}_t\| < \epsilon_1$, by defining $B := \|\boldsymbol{S}_0\|_F^2$, following the steps in Eq. (S44), we have that

$$\eta \sum_{n \notin \mathcal{S}} \exp\left(-\boldsymbol{\mu}_t^\mathsf{T} \boldsymbol{x}_n + \frac{1}{2} \boldsymbol{x}_n^\mathsf{T} \boldsymbol{S}_t \boldsymbol{S}_t^\mathsf{T} \boldsymbol{x}_n\right) \boldsymbol{x}_n^\mathsf{T} \boldsymbol{r}_t \leq \eta |\mathcal{S}| \left(\exp\left(\frac{B}{2}\right) - 1\right) \frac{B}{2}, \tag{S77}$$

and hence, combining this with Assumption 2 which implies that $\boldsymbol{r}_t$ is bounded as in Eq. (S54), one can find a constant $C$ such that

$$(\boldsymbol{r}_{t+1} - \boldsymbol{r}_t)^\mathsf{T} \boldsymbol{r}_t \leq C. \tag{S78}$$

$\square$

**Proof of Theorem 2**

*Proof.* As in the simple version of the proof, we begin by considering the convergence behavior of the mean parameter $\boldsymbol{\mu}_t$.

**Mean parameter**    Our goal is again to show that $\|\boldsymbol{r}_t\|$ is bounded. To that end, we will provide an upper bound to the following equation

$$\|\boldsymbol{r}_{t+1}\|^2 = \|\boldsymbol{r}_{t+1} - \boldsymbol{r}_t\|^2 + 2 (\boldsymbol{r}_{t+1} - \boldsymbol{r}_t)^\mathsf{T} \boldsymbol{r}_t + \|\boldsymbol{r}_t\|^2. \tag{S79}$$

First, consider the first term in the above equation:

$$\begin{aligned}
&\|\boldsymbol{r}_{t+1} - \boldsymbol{r}_t\|^2 \\
&= \|\boldsymbol{\mu}_{t+1} - \hat{\boldsymbol{w}} \log(t+1) - \tilde{\boldsymbol{w}} - \boldsymbol{\mu}_t + \hat{\boldsymbol{w}} \log(t) + \tilde{\boldsymbol{w}}\|^2 \\
&= \| - \eta \nabla_{\boldsymbol{\mu}} \bar{\ell}(\boldsymbol{\mu}_t, \boldsymbol{S}_t) - \hat{\boldsymbol{w}} \log(1 + t^{-1})]\|^2 \\
&\leq 2 \Big[\eta^2 \|\nabla_{\boldsymbol{\mu}} \bar{\ell}(\boldsymbol{\mu}_t, \boldsymbol{S}_t)\|^2 + \|\hat{\boldsymbol{w}}\|^2 \log^2(1 + t^{-1})\Big] \\
&\leq 2 \Big[\eta^2 \|\nabla_{\boldsymbol{\mu}} \bar{\ell}(\boldsymbol{\mu}_t, \boldsymbol{S}_t)\|^2 + \|\hat{\boldsymbol{w}}\|^2 t^{-2}\Big]
\end{aligned} \tag{S80}$$

where in the first inequality we used the standard inequality that $(x + y)^2 \leq 2(x^2 + y^2)$, and in the second inequality we used the fact that $\log(1 + x) \leq x$ for $x \geq 0$. Now, from Lemma S3 and the fact that $t^{-2}$ is summable, we conclude that there exists $C_1 < \infty$ such that

$$\sum_{t=1}^{\infty} \|\boldsymbol{r}_{t+1} - \boldsymbol{r}_t\|^2 \leq C_1 < \infty. \tag{S81}$$

Next, for the second term, recall that in Lemma S4 we showed that if $\|\boldsymbol{P}_{\mathcal{S}} \boldsymbol{r}_t\| \geq \epsilon_1$, then, for some constants $C_2, C_3 < \infty$, we have that, eventually

$$(\boldsymbol{r}_{t+1} - \boldsymbol{r}_t)^\mathsf{T} \boldsymbol{r}_t \leq C_2 \frac{1}{t^\kappa} + C_3 \frac{1}{t^2} \|\boldsymbol{r}_t\|, \tag{S82}$$

and that if $\|\boldsymbol{P}_{\mathcal{S}} \boldsymbol{r}_t\| < \epsilon_1$, then there exists a constant $C_4 < \infty$ such that

$$(\boldsymbol{r}_{t+1} - \boldsymbol{r}_t)^\mathsf{T} \boldsymbol{r}_t \leq C_4. \tag{S83}$$

We will show that when $\|\boldsymbol{P}_{\mathcal{S}} \boldsymbol{r}_t\| < \epsilon_1$, the residual $\boldsymbol{r}_t$ is contained in a compact set, and when $\|\boldsymbol{P}_{\mathcal{S}} \boldsymbol{r}_t\| \geq \epsilon_1$, the residual $\boldsymbol{r}_t$ can't escape to infinity. We now formally show this claim.

Let $S_1$ be the frst time such that $\|\boldsymbol{P}_{\mathcal{S}} \boldsymbol{r}_t\| \geq \epsilon_1$, if such a time does not exist, we are done since the support vectors span the data and hence $\|\boldsymbol{r}_t\|$ is bounded. Now, let $T_1$ be the first time after $S_1$ such that $\|\boldsymbol{P}_{\mathcal{S}} \boldsymbol{r}_t\| < \epsilon_1$, where we allow $T_1 = \infty$ if such a time does not exist. Continuing in this manner, we define the sequences $S_1 < T_1 < S_2 < T_2 < \ldots$, where we allow $T_i = \infty$ for some $i$.

We prooced by showing that $\|\boldsymbol{r}_t\|$ is uniformly bounded on each of the intervals $[S_i, T_i)$. To that end, note that for $t \in [S_i, T_i)$, we have that

$$\|\boldsymbol{r}_{t+1}\|^2 - \|\boldsymbol{r}_t\|^2 \leq 2C_2 \frac{1}{t^\kappa} + 2C_3 \frac{1}{t^2} \|\boldsymbol{r}_t\| + \|\boldsymbol{r}_{t+1} - \boldsymbol{r}_t\|^2, \tag{S84}$$

and hence, using the fact that $\kappa > 1$, by the discrete version of Grönwall's lemma, that

$$\max_{t \in [S_i, T_i)} (\|r_t\|^2 - \|r_{S_i}\|^2) \leq K, \tag{S85}$$

for some constant $K < \infty$ independent of $i$. Furthemore, we also know from Eq. (S83) that

$$\|r_{S_i}\| \leq \epsilon_1 + 2C_4 + \|r_{S_i} - r_{S_i-1}\|^2 \leq \epsilon_1 + 2C_4 + \max_{t \geq 0} \|r_{t+1} - r_t\|^2 < \infty, \tag{S86}$$

showing that the first jump outisde the $\epsilon_1$-ball is bounded. Combining the two results, we conclude that $\|r_t\|$ is uniformly bounded on each of the intervals $[S_i, T_i)$.

Finally, by noting that the support vectors span the data, we have that $\|r_t\|$ is uniformly bounded on each of the intervals $[T_i, S_{i+1})$. Combining the two results, we conclude that $\|r_t\|$ is uniformly bounded for all $t \geq 0$ and hence we have that

$$\lim_{t \to \infty} \frac{\mu_t}{\|\mu_t\|} = \frac{\hat{w}}{\|\hat{w}\|} \tag{S87}$$

and the following lemma.

**Lemma S5**
*For the mean parameter $\mu_t$, we have that*

$$\mu_t = \log(t)\hat{w} + \mathcal{O}(1). \tag{S88}$$

*Proof.* This follows immediately from the definition of the residual in Equation (S31):

$$\mu_t = \hat{w} \log t + r_t + \tilde{w}_t,$$

and the fact that $r_t$ and $\tilde{w}_t$ are bounded as we showed above. $\qquad\square$

We continue with the analysis of the covariance parameter over optimization iterations.

**Covariance parameter** As before, let $\Delta_t = \mathrm{tr}(P_S S_t S_t^\mathsf{T} P_S)$ be the trace of the projection of the covariance parameter on the space of support vectors in $S$. By following the ideas from the gradient flow case, we have the following dynamics:

$$\begin{aligned}
\Delta_{t+1} &= \mathrm{tr}(P_S (I - \eta A_t) S_t S_t^\mathsf{T} (I - \eta A_t)^\mathsf{T} P_S) \\
&= \mathrm{tr}(P_S S_t S_t^\mathsf{T} P_S) - 2\eta \, \mathrm{tr}(P_S S_t S_t^\mathsf{T} A_t P_S) + \eta^2 \, \mathrm{tr}(P_S A_t S_t S_t^\mathsf{T} A_t P_S) \\
&\leq \Delta_t - \frac{2\eta}{t} C\sigma_{\min} \mathrm{tr}(P_S S_t S_t^\mathsf{T} P_S) + \mathcal{O}\left(\frac{1}{t^\kappa}\right) + \mathcal{O}\left(\frac{1}{t^2}\right) \\
&= \Delta_t - \frac{2\eta}{t} C\sigma_{\min}\Delta_t + \mathcal{O}\left(\frac{1}{t^\kappa}\right) + \mathcal{O}\left(\frac{1}{t^2}\right),
\end{aligned} \tag{S89}$$

where we used the same arguments as in Equation (S60) to derive the last inequality, in addition to noting that $\lambda_{\max}(A_t^2) \leq \mathcal{O}\left(\frac{1}{t^2}\right)$ in order to bound the last term. Hence, we can write

$$\Delta_{t+1} - \Delta_t \leq -\frac{2\eta}{t} C\sigma_{\min}\Delta_t + \mathcal{O}\left(\frac{1}{t^\kappa}\right) + \mathcal{O}\left(\frac{1}{t^2}\right). \tag{S90}$$

Again, by the discrete version of Grönwall's lemma, we derive the equivalent result to Eq. (S62). Now, noting that $\sum_t \frac{1}{t}$ diverges, the fact that $\kappa > 1$ and $\eta C\sigma_{\min} > 0$, we conclude that $\Delta_t$ converges to zero. This implies that the covariance parameter converges to zero in the span of the support vectors, i.e.

$$\forall n \in S : \lim_{t \to \infty} x_n^\mathsf{T} S_t S_t^\mathsf{T} x_n = 0, \tag{S91}$$

as desired.

**Characterization as Generalized Variational Inference**  As a final step we need to show that the solution identified by gradient descent if appropriately transformed identifies the minimum 2-Wasserstein solution in the feasible set. Define the feasible set

$$\Theta_\star = \{(\boldsymbol{\mu}, \boldsymbol{S}) \mid \boldsymbol{P}_{\mathcal{S}} \boldsymbol{\mu} = \hat{\boldsymbol{w}} \quad \text{and} \quad \forall n \in \mathcal{S} : \text{Var}_{q_{\boldsymbol{\theta}}}(f_{\boldsymbol{w}}(\boldsymbol{x}_n)) = 0\} \tag{S92}$$

$$= \{(\boldsymbol{\mu}, \boldsymbol{S}) \mid \boldsymbol{P}_{\mathcal{S}} \boldsymbol{\mu} = \hat{\boldsymbol{w}} \quad \text{and} \quad \forall n \in \mathcal{S} : \boldsymbol{x}_n^\mathsf{T} \boldsymbol{S} \boldsymbol{S}^\mathsf{T} \boldsymbol{x}_n = 0\} \tag{S93}$$

and the variational parameters identified by rescaled gradient descent as

$$\boldsymbol{\theta}_\star^{\text{rGD}} = \lim_{t \to \infty} \boldsymbol{\theta}_t^{\text{rGD}} = \lim_{t \to \infty} \left( \frac{1}{\log(t)} \boldsymbol{\mu}_t + \boldsymbol{P}_{\text{null}(\boldsymbol{X})} \boldsymbol{\mu}_0, \boldsymbol{S}_t \right). \tag{S94}$$

It holds by Lemma S5 that

$$\boldsymbol{P}_{\mathcal{S}} \boldsymbol{\mu}_\star^{\text{rGD}} = \boldsymbol{P}_{\mathcal{S}} \left( \lim_{t \to \infty} \frac{1}{\log(t)} \boldsymbol{\mu}_t \right) + \boldsymbol{0} = \boldsymbol{P}_{\mathcal{S}} \hat{\boldsymbol{w}} = \hat{\boldsymbol{w}} \tag{S95}$$

and additionally by Equation (S91) we have for all $n \in \mathcal{S}$ that

$$\boldsymbol{x}_n^\mathsf{T} \boldsymbol{S}_\star^{\text{rGD}} (\boldsymbol{S}_\star^{\text{rGD}})^\mathsf{T} \boldsymbol{x}_n = \lim_{t \to \infty} \boldsymbol{x}_n^\mathsf{T} \boldsymbol{S}_t (\boldsymbol{S}_t)^\mathsf{T} \boldsymbol{x}_n = 0. \tag{S96}$$

Therefore, the limit point $\boldsymbol{\theta}_\star^{\text{rGD}}$ of rescaled gradient descent is in the feasible set. It remains to show that it is also a minimizer of the 2-Wasserstein distance to the prior / initialization. We will first show a more general result that does not require Assumption 2.

To that end define $\begin{pmatrix} \boldsymbol{V}_{\mathcal{S}} & \boldsymbol{V}_{\boldsymbol{X} \perp \mathcal{S}} & \boldsymbol{V}_{\text{null}(\boldsymbol{X})} \end{pmatrix} \in \mathbb{R}^{P \times P}$ where $\boldsymbol{V}_{\mathcal{S}} \in \mathbb{R}^{P \times P_{\mathcal{S}}}$ is an orthonormal basis of the span of the support vectors $\text{range}(\boldsymbol{X}_{\mathcal{S}}^\mathsf{T})$, $\boldsymbol{V}_{\boldsymbol{X} \perp \mathcal{S}} \in \mathbb{R}^{P \times (N - P_{\mathcal{S}})}$ an orthonormal basis of its orthogonal complement in $\text{range}(\boldsymbol{X}^\mathsf{T})$ and $\boldsymbol{V}_{\text{null}(\boldsymbol{X})} \in \mathbb{R}^{P \times (P - N)}$ the corresponding orthonormal basis of the null space $\text{null}(\boldsymbol{X})$ of the data. Let $\boldsymbol{V} = \begin{pmatrix} \boldsymbol{V}_{\mathcal{S}} & \boldsymbol{V}_{\text{null}(\boldsymbol{X})} \end{pmatrix} \in \mathbb{R}^{P \times (P - N + P_{\mathcal{S}})}$ and define the projected variational distribution and prior onto the span of the support vectors and the null space of the data as

$$q_{\boldsymbol{\theta}}^{\text{proj}}(\tilde{\boldsymbol{w}}) = \mathcal{N}\big(\tilde{\boldsymbol{w}}; \boldsymbol{P}_V \boldsymbol{\mu}, \boldsymbol{P}_V \boldsymbol{\Sigma} \boldsymbol{P}_V^\mathsf{T}\big) = \mathcal{N}\Big(\tilde{\boldsymbol{w}}; \tilde{\boldsymbol{\mu}}, \tilde{\boldsymbol{\Sigma}}\Big) \tag{S97}$$

$$p^{\text{proj}}(\tilde{\boldsymbol{w}}) = \mathcal{N}\big(\tilde{\boldsymbol{w}}; \boldsymbol{P}_V \boldsymbol{\mu}_0, \boldsymbol{P}_V \boldsymbol{\Sigma}_0 \boldsymbol{P}_V^\mathsf{T}\big) = \mathcal{N}\Big(\tilde{\boldsymbol{w}}; \tilde{\boldsymbol{\mu}}_0, \tilde{\boldsymbol{\Sigma}}_0\Big) \tag{S98}$$

where $\tilde{\boldsymbol{w}} \in \mathbb{R}^{P - N + P_{\mathcal{S}}}$. Now earlier we showed that the limit point of rescaled gradient descent is in the feasible set, defined in Equation (S94), and thus the same holds for the projected limit point of rescaled gradient descent, i.e.

$$(\tilde{\boldsymbol{\mu}}_\star^{\text{rGD}}, \tilde{\boldsymbol{S}}_\star^{\text{rGD}}) \in \Theta_\star \tag{S99}$$

in particular

$$\boldsymbol{P}_{\mathcal{S}} \tilde{\boldsymbol{\mu}}_\star^{\text{rGD}} = \boldsymbol{P}_{\mathcal{S}} \boldsymbol{\mu}_\star^{\text{rGD}} = \hat{\boldsymbol{w}}, \tag{S100}$$

$$\forall n \in \mathcal{S}: \quad \boldsymbol{x}_n^\mathsf{T} \tilde{\boldsymbol{S}}_\star^{\text{rGD}} (\tilde{\boldsymbol{S}}_\star^{\text{rGD}})^\mathsf{T} \boldsymbol{x}_n = \boldsymbol{x}_n^\mathsf{T} \boldsymbol{S}_\star^{\text{rGD}} (\boldsymbol{S}_\star^{\text{rGD}})^\mathsf{T} \boldsymbol{x}_n = 0. \tag{S101}$$

Therefore, we have for all $n \in \mathcal{S}$ that

$$0 = \boldsymbol{x}_n^\mathsf{T} \tilde{\boldsymbol{S}}_\star^{\text{rGD}} (\tilde{\boldsymbol{S}}_\star^{\text{rGD}})^\mathsf{T} \boldsymbol{x}_n = \|(\tilde{\boldsymbol{S}}_\star^{\text{rGD}})^\mathsf{T} \boldsymbol{x}_n\|_2^2 \iff (\tilde{\boldsymbol{S}}_\star^{\text{rGD}})^\mathsf{T} \boldsymbol{x}_n = \boldsymbol{0} \tag{S102}$$

$$\iff (\tilde{\boldsymbol{S}}_\star^{\text{rGD}})^\mathsf{T} \boldsymbol{V}_{\mathcal{S}} = \boldsymbol{0} \tag{S103}$$

and thus $\boldsymbol{V}_{\mathcal{S}}^\mathsf{T} \tilde{\boldsymbol{S}}_\star^{\text{rGD}} (\tilde{\boldsymbol{S}}_\star^{\text{rGD}})^\mathsf{T} \boldsymbol{V}_{\mathcal{S}} = \boldsymbol{0}$. Therefore by Lemma S1 it holds for the squared 2-Wasserstein distance between the projected limit point of rescaled gradient descent and the projected prior that

$$\text{W}_2^2\Big(q_{\boldsymbol{\theta}_\star}^{\text{proj}}, p^{\text{proj}}\Big) \overset{\pm c}{=} \left\|\boldsymbol{V}_{\mathcal{S}}^\mathsf{T} \tilde{\boldsymbol{\mu}} - \boldsymbol{V}_{\mathcal{S}}^\mathsf{T} \tilde{\boldsymbol{\mu}}_0\right\|_2^2 + \text{W}_2^2\Big(\mathcal{N}\Big(\boldsymbol{V}_{\text{null}}^\mathsf{T} \tilde{\boldsymbol{\mu}}, \boldsymbol{V}_{\text{null}}^\mathsf{T} \tilde{\boldsymbol{\Sigma}} \boldsymbol{V}_{\text{null}}\Big), \mathcal{N}\Big(\boldsymbol{V}_{\text{null}}^\mathsf{T} \tilde{\boldsymbol{\mu}}_0, \boldsymbol{V}_{\text{null}}^\mathsf{T} \tilde{\boldsymbol{\Sigma}}_0 \boldsymbol{V}_{\text{null}}\Big)\Big)$$

$$= \left\|\begin{pmatrix} \boldsymbol{V}_{\mathcal{S}}^\mathsf{T} \tilde{\boldsymbol{\mu}} - \boldsymbol{V}_{\mathcal{S}}^\mathsf{T} \tilde{\boldsymbol{\mu}}_0 \\ \boldsymbol{0} \end{pmatrix}\right\|_2^2 + \text{W}_2^2\Big(\mathcal{N}\Big(\boldsymbol{V}_{\text{null}}^\mathsf{T} \tilde{\boldsymbol{\mu}}, \boldsymbol{V}_{\text{null}}^\mathsf{T} \tilde{\boldsymbol{\Sigma}} \boldsymbol{V}_{\text{null}}\Big), \mathcal{N}\Big(\boldsymbol{V}_{\text{null}}^\mathsf{T} \tilde{\boldsymbol{\mu}}_0, \boldsymbol{V}_{\text{null}}^\mathsf{T} \tilde{\boldsymbol{\Sigma}}_0 \boldsymbol{V}_{\text{null}}\Big)\Big)$$

$$= \left\|\boldsymbol{V} \begin{pmatrix} \boldsymbol{V}_{\mathcal{S}}^\mathsf{T} \tilde{\boldsymbol{\mu}} - \boldsymbol{V}_{\mathcal{S}}^\mathsf{T} \tilde{\boldsymbol{\mu}}_0 \\ \boldsymbol{0} \end{pmatrix}\right\|_2^2 + \text{W}_2^2\Big(\mathcal{N}\Big(\boldsymbol{V}_{\text{null}}^\mathsf{T} \tilde{\boldsymbol{\mu}}, \boldsymbol{V}_{\text{null}}^\mathsf{T} \tilde{\boldsymbol{\Sigma}} \boldsymbol{V}_{\text{null}}\Big), \mathcal{N}\Big(\boldsymbol{V}_{\text{null}}^\mathsf{T} \tilde{\boldsymbol{\mu}}_0, \boldsymbol{V}_{\text{null}}^\mathsf{T} \tilde{\boldsymbol{\Sigma}}_0 \boldsymbol{V}_{\text{null}}\Big)\Big)$$

$$= \|\boldsymbol{P}_{\mathcal{S}}\tilde{\boldsymbol{\mu}} - \boldsymbol{P}_{\mathcal{S}}\tilde{\boldsymbol{\mu}}_0\|_2^2 + \mathrm{W}_2^2\Big(\mathcal{N}\big(\boldsymbol{V}_{\mathrm{null}}^{\mathsf{T}}\tilde{\boldsymbol{\mu}}, \boldsymbol{V}_{\mathrm{null}}^{\mathsf{T}}\tilde{\boldsymbol{\Sigma}}\boldsymbol{V}_{\mathrm{null}}\big), \mathcal{N}\big(\boldsymbol{V}_{\mathrm{null}}^{\mathsf{T}}\tilde{\boldsymbol{\mu}}_0, \boldsymbol{V}_{\mathrm{null}}^{\mathsf{T}}\tilde{\boldsymbol{\Sigma}}_0\boldsymbol{V}_{\mathrm{null}}\big)\Big)$$

$$= \|\hat{\boldsymbol{w}} - \boldsymbol{P}_{\mathcal{S}}\tilde{\boldsymbol{\mu}}_0\|_2^2 + \mathrm{W}_2^2\Big(\mathcal{N}\big(\boldsymbol{V}_{\mathrm{null}}^{\mathsf{T}}\tilde{\boldsymbol{\mu}}, \boldsymbol{V}_{\mathrm{null}}^{\mathsf{T}}\tilde{\boldsymbol{\Sigma}}\boldsymbol{V}_{\mathrm{null}}\big), \mathcal{N}\big(\boldsymbol{V}_{\mathrm{null}}^{\mathsf{T}}\tilde{\boldsymbol{\mu}}_0, \boldsymbol{V}_{\mathrm{null}}^{\mathsf{T}}\tilde{\boldsymbol{\Sigma}}_0\boldsymbol{V}_{\mathrm{null}}\big)\Big)$$

$$\overset{\pm c}{=} \mathrm{W}_2^2\Big(\mathcal{N}\big(\boldsymbol{V}_{\mathrm{null}}^{\mathsf{T}}\tilde{\boldsymbol{\mu}}, \boldsymbol{V}_{\mathrm{null}}^{\mathsf{T}}\tilde{\boldsymbol{\Sigma}}\boldsymbol{V}_{\mathrm{null}}\big), \mathcal{N}\big(\boldsymbol{V}_{\mathrm{null}}^{\mathsf{T}}\tilde{\boldsymbol{\mu}}_0, \boldsymbol{V}_{\mathrm{null}}^{\mathsf{T}}\tilde{\boldsymbol{\Sigma}}_0\boldsymbol{V}_{\mathrm{null}}\big)\Big)$$

where we used that $\boldsymbol{P}_{\mathcal{S}}\tilde{\boldsymbol{\mu}} = \hat{\boldsymbol{w}}$ for any $(\tilde{\boldsymbol{\mu}}, \tilde{\boldsymbol{S}})$ in the feasible set $\Theta_\star$. Therefore it suffices to show that the projected solution $\tilde{\boldsymbol{\theta}}_\star^{\mathrm{rGD}}$ minimizes

$$\mathrm{W}_2^2\Big(\mathcal{N}\big(\boldsymbol{V}_{\mathrm{null}}^{\mathsf{T}}\tilde{\boldsymbol{\mu}}, \boldsymbol{V}_{\mathrm{null}}^{\mathsf{T}}\tilde{\boldsymbol{\Sigma}}\boldsymbol{V}_{\mathrm{null}}\big), \mathcal{N}\big(\boldsymbol{V}_{\mathrm{null}}^{\mathsf{T}}\tilde{\boldsymbol{\mu}}_0, \boldsymbol{V}_{\mathrm{null}}^{\mathsf{T}}\tilde{\boldsymbol{\Sigma}}_0\boldsymbol{V}_{\mathrm{null}}\big)\Big) \geq 0. \tag{S104}$$

We have using the definition of the iterates in Equation (9) that

$$\boldsymbol{V}_{\mathrm{null}}^{\mathsf{T}}\tilde{\boldsymbol{\mu}}_\star^{\mathrm{rGD}} = \boldsymbol{V}_{\mathrm{null}}^{\mathsf{T}}\boldsymbol{P}_{\boldsymbol{V}}\left(\lim_{t\to\infty}\frac{1}{\log(t)}\boldsymbol{\mu}_t + \boldsymbol{P}_{\mathrm{null}(\boldsymbol{X})}\boldsymbol{\mu}_0\right) \tag{S105}$$

$$= \boldsymbol{V}_{\mathrm{null}}^{\mathsf{T}}(\hat{\boldsymbol{w}} + \boldsymbol{P}_{\mathrm{null}(\boldsymbol{X})}\boldsymbol{\mu}_0) = \boldsymbol{V}_{\mathrm{null}}^{\mathsf{T}}\boldsymbol{\mu}_0 \tag{S106}$$

where we used $\hat{\boldsymbol{w}} \in \mathrm{range}(\boldsymbol{X}_{\mathcal{S}}^{\mathsf{T}})$. Further, it holds for the gradient of the expected loss (S30) with respect to the covariance factor parameters that

$$\boldsymbol{V}_{\mathrm{null}}^{\mathsf{T}}\tilde{\boldsymbol{S}}_\star^{\mathrm{rGD}} = \boldsymbol{V}_{\mathrm{null}}^{\mathsf{T}}\boldsymbol{P}_{\boldsymbol{V}}\boldsymbol{S}_\star^{\mathrm{rGD}} = \boldsymbol{V}_{\mathrm{null}}^{\mathsf{T}}\boldsymbol{S}_\star^{\mathrm{rGD}} = \boldsymbol{V}_{\mathrm{null}}^{\mathsf{T}}\bigg(\boldsymbol{S}_0 - \underbrace{\sum_{t=1}^{\infty}\eta_t\nabla_{\boldsymbol{S}}\bar{\ell}(\boldsymbol{\mu}_t, \boldsymbol{S}_t)}_{\in\mathrm{range}(\boldsymbol{X}^{\mathsf{T}})}\bigg) \tag{S107}$$

$$= \boldsymbol{V}_{\mathrm{null}}^{\mathsf{T}}\boldsymbol{S}_0 = \boldsymbol{V}_{\mathrm{null}}^{\mathsf{T}}\boldsymbol{P}_{\boldsymbol{V}}\boldsymbol{S}_0 = \boldsymbol{V}_{\mathrm{null}}^{\mathsf{T}}\tilde{\boldsymbol{S}}_0. \tag{S108}$$

Therefore we have that

$$\mathrm{W}_2^2\Big(\mathcal{N}\big(\boldsymbol{V}_{\mathrm{null}}^{\mathsf{T}}\tilde{\boldsymbol{\mu}}_\star^{\mathrm{rGD}}, \boldsymbol{V}_{\mathrm{null}}^{\mathsf{T}}\tilde{\boldsymbol{\Sigma}}_\star^{\mathrm{rGD}}\boldsymbol{V}_{\mathrm{null}}\big), \mathcal{N}\big(\boldsymbol{V}_{\mathrm{null}}^{\mathsf{T}}\tilde{\boldsymbol{\mu}}_0, \boldsymbol{V}_{\mathrm{null}}^{\mathsf{T}}\tilde{\boldsymbol{\Sigma}}_0\boldsymbol{V}_{\mathrm{null}}\big)\Big) = 0 \tag{S109}$$

and thus the projected variational parameters $\tilde{\boldsymbol{\theta}}_\star^{\mathrm{rGD}}$ are both feasible (S99) and minimize the squared 2-Wasserstein distance to the projected initialization / prior (S104). This completes the proof for the generalized version of Theorem 2 without Assumption 2, which we state here for convenience.

**Lemma S6**
*Given the assumptions of Theorem 2, except for Assumption 2 meaning the support vectors $\boldsymbol{X}_{\mathcal{S}}$ do not necessarily span the data, it holds for the limit point of rescaled gradient descent that*

$$\boldsymbol{\theta}_\star^{\mathrm{rGD}} \in \underset{\substack{\boldsymbol{\theta}=(\boldsymbol{\mu}, \boldsymbol{S}) \\ s.t.\ \boldsymbol{\theta}\in\Theta_\star}}{\arg\min}\,\mathrm{W}_2^2\Big(q_{\boldsymbol{\theta}}^{\mathrm{proj}}, p^{\mathrm{proj}}\Big). \tag{S110}$$

*If in addition Assumption 2 holds, i.e. the support vectors span the training data $\boldsymbol{X}$, such that*

$$\mathrm{span}(\{\boldsymbol{x}_n\}_{n\in[N]}) = \mathrm{span}(\{\boldsymbol{x}_n\}_{n\in\mathcal{S}}), \tag{S111}$$

*then the orthogonal complement of the support vectors in $\mathrm{range}(\boldsymbol{X}^{\mathsf{T}})$ has dimension $N - P_{\mathcal{S}} = 0$ and thus the projection $\boldsymbol{P}_{\boldsymbol{V}} = \boldsymbol{I}_{P\times P}$ is the identity and therefore*

$$q_{\boldsymbol{\theta}}^{\mathrm{proj}} = q_{\boldsymbol{\theta}} \qquad \text{and} \qquad p^{\mathrm{proj}} = p. \tag{S112}$$

This completes the proof of Theorem 2.

$\square$

## S1.3 NLL Overfitting and the Need for (Temperature) Scaling

In Theorem 2, we assume we rescale the mean parameters. This is because the exponential loss can be made arbitrarily small for a mean vector that is aligned with the $L_2$ max-margin vector simply by increasing its magnitude. In fact, the sequence of mean parameters identified by gradient descent diverges to infinity at a logarithmic rate $\boldsymbol{\mu}_t^{\mathrm{GD}} \approx \log(t)\hat{\boldsymbol{w}}$ as we show[4] in Lemma S5 and illustrate in Figure S2 (right panel).

---

[4]This has been observed previously in the deterministic case (see Theorem 3 of Soudry et al. [4]) and thus naturally also appears in our probabilistic extension.

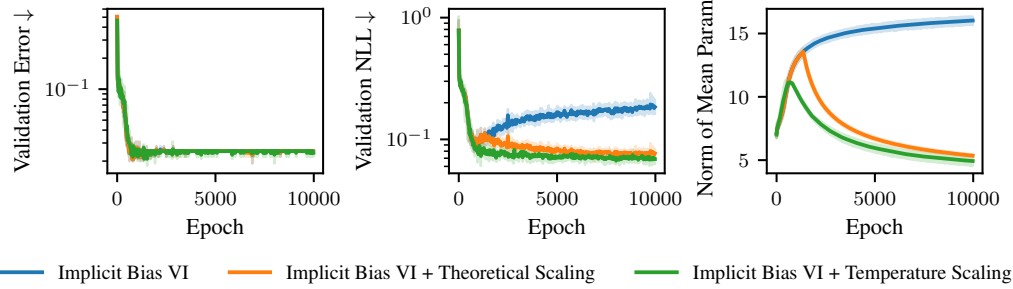

Figure S2: *NLL overfitting in classification due to implicit bias of the mean parameters.* As shown here for a two-hidden layer neural network on synthetic data, when training with vanilla SGD the mean parameters diverge to infinity $\|\boldsymbol{\mu}_t\|_2 \approx \mathcal{O}(\log(t))$ (right) and thus the classifier will eventually overfit in terms of negative log-likelihood (left and middle). Rescaling the GD iterates as in Theorem 2 or using temperature scaling [69] avoids overfitting.

This bias of the mean parameters towards the max-margin solution does not impact the train loss or validation error, but leads to overfitting in terms of validation NLL (see Figure S2) as long as there is at least one misclassified datapoint $\boldsymbol{x}$, since then the (average) validation NLL is given by

$$
\begin{aligned}
\bar{\ell}(\boldsymbol{\theta}_t^{\text{GD}}) &= \mathbb{E}_{q_{\boldsymbol{\theta}_t^{\text{GD}}}(\boldsymbol{w})}\big(\exp(-y\boldsymbol{x}^\mathsf{T}\boldsymbol{w})\big) = \exp(\boldsymbol{x}^\mathsf{T}\boldsymbol{\mu}_t^{\text{GD}} + \tfrac{1}{2}\boldsymbol{x}^\mathsf{T}\boldsymbol{S}_t^{\text{GD}}(\boldsymbol{S}_t^{\text{GD}})^\mathsf{T}\boldsymbol{x}) \\
&\approx \exp(\log(t)\boldsymbol{x}^\mathsf{T}\hat{\boldsymbol{w}} + \tfrac{1}{2}\boldsymbol{x}^\mathsf{T}\boldsymbol{S}_t^{\text{GD}}(\boldsymbol{S}_t^{\text{GD}})^\mathsf{T}\boldsymbol{x}) \to \infty \quad \text{as} \quad t \to \infty.
\end{aligned}
\tag{S113}
$$

However, by rescaling the mean parameters as we do in Theorem 2, this can be prevented as Figure S2 (middle panel) illustrates for a two-hidden layer neural network on synthetic data. Such overfitting in terms of NLL has been studied extensively empirically with the perhaps most common remedy being Temperature Scaling (TS) [69]. As we show empirically in Figure S2, instead of using the theoretical rescaling, using temperature scaling performs very well, especially in the non-asymptotic regime, which is why we also adopt it for our experiments in Section 5.

The aforementioned divergence of the mean parameters to infinity also explains the need for the projection of the prior mean parameters in Equation (9), since any bias from the initialization vanishes in the limit of infinite training. At first glance the additional projection seems computationally prohibitive for anything but a zero mean prior, but close inspection of the implicit bias of the covariance parameters $\boldsymbol{S}$ in Theorem 2 shows that at convergence

$$
\forall n : \text{Var}_{q_\theta}(f_{\boldsymbol{w}}(\boldsymbol{x}_n)) = \boldsymbol{x}_n^\mathsf{T}\boldsymbol{S}\boldsymbol{S}^\mathsf{T}\boldsymbol{x}_n = 0 \implies \text{range}(\boldsymbol{S}) \subset \text{null}(\boldsymbol{X})
\tag{S114}
$$

Meaning we can approximate a basis of the null space of the training data by computing a QR decomposition of the covariance factor in $\mathcal{O}(PR^2)$ once at the end of training. For $R = P$ the inclusion becomes an equality and the projection can be computed exactly.

# S2  PARAMETRIZATION, FEATURE LEARNING AND HYPERPARAMETER TRANSFER

**Notation**  For this section we need a more detailed neural network notation. Denote an $L$-hidden layer, width-$D$ feedforward neural network by $f(\boldsymbol{x}) \in \mathbb{R}_{\text{out}}^D$, with inputs $\boldsymbol{x} \in \mathbb{R}^{D_{\text{in}}}$, weights $\boldsymbol{W}^{(l)}$, pre-activations $\boldsymbol{h}^{(l)}(\boldsymbol{x}) \in \mathbb{R}^{D^{(l)}}$, and post-activations (or "features") $\boldsymbol{g}^{(l)}(\boldsymbol{x}) \in \mathbb{R}^{D^{(l)}}$. That is, $\boldsymbol{h}^{(1)}(\boldsymbol{x}) = \boldsymbol{W}^{(1)}\boldsymbol{x}$ and, for $l \in 1, \dots, L-1$,

$$
\boldsymbol{g}^{(l)}(\boldsymbol{x}) = \phi\left(\boldsymbol{h}^{(l)}(\boldsymbol{x})\right), \; \boldsymbol{h}^{(l+1)}(\boldsymbol{x}) = \boldsymbol{W}^{(l+1)}\boldsymbol{g}^{(l)}(\boldsymbol{x}),
$$

and the network output is given by $f(\boldsymbol{x}) = \boldsymbol{W}^{(L+1)}\boldsymbol{g}^{(L)}(\boldsymbol{x})$, where $\phi(\bullet)$ is an activation function.

For convenience, we may abuse notation and write $\boldsymbol{h}^{(0)}(\boldsymbol{x}) = \boldsymbol{x}$ and $\boldsymbol{h}^{(L+1)}(\boldsymbol{x}) = f(\boldsymbol{x})$. Throughout we use $\bullet^{(l)}$ to indicate the layer, subscript $\bullet_t$ to indicate the training time (i.e., epoch), $\Delta\bullet_t = \bullet_t - \bullet_0$ to indicate the change since initialization, and $[\bullet]_i$, $[\bullet]_{ij}$ to indicate the component within a vector or matrix.

## S2.1 Definitions of Stability and Feature Learning

The following definitions extend those of Yang and Hu [43] to the variational setting.

**Definition S1** (*bc* scaling)
In layer $l$, the variational parameters are initialized as

$$[\boldsymbol{\mu}_0^{(l)}]_i \sim \mathcal{N}\left(0, D^{-2b^{(l)}}\right), \quad [\boldsymbol{S}_0^{(l)}]_{ij} \sim \mathcal{N}\left(0, D^{-2\tilde{b}^{(l)}}\right)$$

and the learning rates for the mean and covariance parameters, respectively, are set to

$$\eta^{(l)} = \eta D^{-c^{(l)}}, \tilde{\eta}^{(l)} = \eta D^{-\tilde{c}^{(l)}}.$$

The hyperparameter $\eta$ represents a global learning rate that can be tuned, as for example in the hyperparameter transfer experiment from Section 3.4.

For the next two definitions, let $m_r(X) = \mathbb{E}_{\boldsymbol{z}}((X - \mathbb{E}_{\boldsymbol{z}}(X))^r)$ denote the $r$th central moment moment of a random variable $X$ with respect to $\boldsymbol{z}$, which represents all reparameterization noise in the random variable $X$. All Landau notation in Section S2 refers to asymptotic behavior in width $D$ in probability over reparameterization noise $\boldsymbol{z}$. We say that a vector sequence $\{\boldsymbol{v}_D\}_{D=1}^\infty$, where each $\boldsymbol{v}_D \in \mathbb{R}^D$, is $\mathcal{O}(D^{-a})$ if the scalar sequence $\{\sqrt{\frac{1}{D}\|\boldsymbol{v}_D\|^2}\}_{D=1}^\infty = \{\text{RMSE}(\boldsymbol{v}_D)\}_{D=1}^\infty$ is $\mathcal{O}(D^{-a})$.

**Definition S2** (Stability of Moment $r$)
A neural network is *stable in moment* $r$, if all of the following hold for all $\boldsymbol{x}$ and $l \in \{1, \ldots, L\}$.

1. At initialization ($t = 0$):

    (a) The pre- and post-activations are $\Theta(1)$:
    $$m_r(\boldsymbol{h}_0^{(l)}(\boldsymbol{x})), m_r(\boldsymbol{g}_0^{(l)}(\boldsymbol{x})) = \Theta(1).$$

    (b) The function is $\mathcal{O}(1)$:
    $$m_r(f_0(\boldsymbol{x})) = \mathcal{O}(1).$$

2. At any point during training $t > 0$:

    (a) The change from initialization in the pre- and post-activations are $\mathcal{O}(1)$:
    $$\Delta m_r(\boldsymbol{h}_t^{(l)}(\boldsymbol{x})), \Delta m_r(\boldsymbol{g}_t^{(l)}(\boldsymbol{x})) = \mathcal{O}(1).$$

    (b) The function is $\mathcal{O}(1)$:
    $$m_r(f_t(\boldsymbol{x})) = \mathcal{O}(1).$$

**Definition S3** (Feature Learning of Moment $r$)
*Feature learning* occurs in moment $r$ in layer $l$ if, for any $t > 0$, the change from initialization is $\Omega(1)$:

$$\Delta m_r\left(\boldsymbol{g}_t^{(l)}(\boldsymbol{x})\right) = \Omega(1).$$

As we will see later, Figure S5 and Figure S6 investigate feature learning for the first two moments.

## S2.2 Initialization Scaling for a Linear Network

In this section we illustrate how the initialization scaling $\{(b^{(l)}, \tilde{b}^{(l)})\}$ can be chosen for stability. For simplicity, we consider a linear feedforward network of width $D$ evaluated on a single input $\boldsymbol{x} \in \mathbb{R}_{\text{in}}^D$. We assume a Gaussian variational family that factorizes across layers. This implies the hidden units evolve as $\boldsymbol{h}_t^{(l+1)} = \boldsymbol{W}_t^{(l+1)}\boldsymbol{h}_t^{(l)}$ and the weights are linked to the variational parameters by $\text{vec}(\boldsymbol{W}_t^{(l)}) = \boldsymbol{\mu}_t^{(l)} + \boldsymbol{S}_t^{(l)}\boldsymbol{z}$.

Therefore, the mean and variance of the $i$th component hidden units in layer $l \in \{1, \ldots, L+1\}$, where $i \in 1 \ldots, D^{(l)}$, are given by

$$\mathbb{E}_z\left([\boldsymbol{h}_t^{(l)}]_i\right) = [\boldsymbol{\mu}_t^{(l)}]_I^\mathsf{T} \mathbb{E}_z\left(\boldsymbol{h}_t^{(l-1)}\right)$$

$$\mathrm{Var}_z\left([\boldsymbol{h}_t^{(l)}]_i\right) = [\boldsymbol{\mu}_t^{(l)}]_I^\mathsf{T} \boldsymbol{C}_t^{(l-1)} [\boldsymbol{\mu}^{(l)}]_I + \mathrm{tr}([\boldsymbol{S}_t^{(l)}]_{I,:}^\mathsf{T} \boldsymbol{A}_t^{(l-1)} [\boldsymbol{S}_t^{(l)}]_{I,:}),$$

where $I = \{iD^{(l-1)}, \ldots, (i+1)D^{(l-1)}\}$ and the second moment of and covariance of layer-$l$ hidden units are denoted by

$$\boldsymbol{A}_t^{(l)} = \mathbb{E}_z\left(\boldsymbol{h}_t^{(l)} \boldsymbol{h}_t^{((l))\mathsf{T}}\right)$$

$$\boldsymbol{C}_t^{(l)} = \boldsymbol{A}_t^{(l)} - \mathbb{E}_z\left(\boldsymbol{h}_t^{(l)}\right) \mathbb{E}_z\left(\boldsymbol{h}_t^{(l)}\right)^\mathsf{T}.$$

**Mean**   We start with the mean of the hidden units, which conveniently depends only on the mean variational parameters and the previous layer hidden units.

$$\mathbb{E}_z\left([\boldsymbol{h}_0^{(l)}]_i\right) = \sum_{j=1}^{D^{(l-1)}} [\boldsymbol{\mu}_0^{(l)}]_{I_j} \mathbb{E}_z\left([\boldsymbol{h}_0^{(l-1)}]_j\right)$$

$$= \mathcal{O}\left(\sqrt{D^{(l-1)}} \cdot D^{-b^{(l)}} \cdot 1\right)$$

$$= \begin{cases} \mathcal{O}\left(D^{-b^{(1)}}\right) & l = 1 \\ \mathcal{O}\left(D^{-(b^{(l)}-\frac{1}{2})}\right) & l \in \{2, \ldots, L+1\}. \end{cases}$$

Therefore, we require $b^{(1)} \geq 0$ and $b^{(l)} \geq \frac{1}{2}$ for $l \in \{2, \ldots, L+1\}$.

**Variance**   Next we examine the variance of hidden units. Consider the first term, which represents the contribution of the mean parameters.

$$[\boldsymbol{\mu}_0^{(l)}]_I^\mathsf{T} \boldsymbol{C}_0^{(l-1)} [\boldsymbol{\mu}^{(l)}]_I = \sum_{j=1}^{D^{(l-1)}} [\boldsymbol{\mu}_0^{(l)}]_{I_j}^2 [\boldsymbol{C}_0^{(l-1)}]_{j,j} + \sum_{j \neq j'}^{D^{(l-1)}} [\boldsymbol{\mu}_0^{(l)}]_{I_j} [\boldsymbol{C}_0^{(l-1)}]_{j,j'} [\boldsymbol{\mu}_0^{(l)}]_{I_{j'}}$$

$$= \mathcal{O}\left(D^{(l-1)} \cdot D^{-2b^{(l)}} \cdot 1\right) + \mathcal{O}\left(\sqrt{D^{(l-1)}(D^{(l-1)}-1)} \cdot D^{-b^{(l)}} \cdot 1 \cdot D^{-b^{(l)}}\right)$$

$$= \mathcal{O}\left(D^{(l-1)} \cdot D^{-2b^{(l)}}\right)$$

$$= \begin{cases} \mathcal{O}\left(D^{-2b^{(1)}}\right) & l = 1 \\ \mathcal{O}\left(D^{-(2b^{(l)}-1)}\right) & l \in l \in \{2, \ldots, L+1\}. \end{cases}$$

Therefore, we require $b^{(1)} \geq 0$ and $b^{(l)} \geq \frac{1}{2}$ for $l \in \{2, \ldots, L+1\}$. Notice these are the same requirements as above for the mean of the hidden units. We summarize the scaling for the mean parameters as

$$b^{(l)} \geq \begin{cases} 0 & l = 1 \\ \frac{1}{2} & l \in \{2, \ldots, L+1\}. \end{cases} \tag{S115}$$

Now consider the second term in the variance of the hidden units. Assume the rank scales with the input and output dimension of a layer as $R^{(l)} = (D^{(l-1)} D^{(l)})^{p^{(l)}}$, where $p^{(l)} \in [0, 1]$.

$$\mathrm{tr}([\boldsymbol{S}_0^{(l)}]_{I,:}^\mathsf{T} \boldsymbol{A}_0^{(l-1)} [\boldsymbol{S}_0^{(l)}]_{I,:}) = \sum_{r=1}^{R^{(l)}} [\boldsymbol{S}_0^{(l)}]_{I,r}^\mathsf{T} \boldsymbol{A}_0^{(l-1)} [\boldsymbol{S}_0^{(l)}]_{I,r}$$

$$= \sum_{r=1}^{R^{(l)}} \left( \sum_{j=1}^{D^{(l-1)}} [\boldsymbol{S}_0^{(l)}]_{I_j,r}^2 [\boldsymbol{A}_0^{(l-1)}]_{j,j} + \sum_{j \neq j'}^{D^{(l-1)}} [\boldsymbol{S}_0^{(l)}]_{I_j,r} [\boldsymbol{A}_0^{(l-1)}]_{j,j'} [\boldsymbol{S}_0^{(l)}]_{I_{j'},r} \right)$$

$$= \mathcal{O}\left(R^{(l)} D^{(l-1)} \cdot D^{-2\tilde{b}^{(l)}} \cdot 1\right) + \mathcal{O}\left(\sqrt{R^{(l)} D^{(l-1)}(D^{(l-1)}-1)} \cdot D^{-\tilde{b}^{(l)}} \cdot 1 \cdot D^{-\tilde{b}^{(l)}}\right)$$

$$= \mathcal{O}\left(R^{(l)} D^{(l-1)} D^{-2\tilde{b}^{(l)}}\right)$$

$$= \begin{cases} \mathcal{O}\left(D^{-(2\tilde{b}^{(1)}-p^{(1)})}\right) & l = 1 \\ \mathcal{O}\left(D^{-(2\tilde{b}^{(l)}-1-2p^{(l)})}\right) & l \in \{2, \ldots, L\} \\ \mathcal{O}\left(D^{-(2\tilde{b}^{(L+1)}-1-p^{(L+1)})}\right) & l = L + 1. \end{cases}$$

Therefore we require $\tilde{b}^{(0)} \geq \frac{p^{(1)}}{2}$, $\tilde{b}^{(l)} \geq \frac{1}{2} + p^{(l)}$ for $l \in \{2, \ldots, L\}$, and $\tilde{b}^{(L+1)} \geq \frac{1}{2} + \frac{p^{(L+1)}}{2}$. Notice we can write these conditions in terms of the mean scaling as

$$\boxed{\tilde{b}^{(l)} \geq b^{(l)} + \begin{cases} \frac{p^{(l)}}{2} & l = 1 \\ p^{(l)} & l \in \{2, \ldots, L\} \\ \frac{p^{(l)}}{2} & l = L + 1. \end{cases}} \tag{S116}$$

## S2.3 PROPOSED SCALING

The previous section derives the necessary conditions for stability at initialization. Recall from Section 3.4 that we propose scaling the contribution of the covariance parameters to the forward pass, i.e. the $\boldsymbol{Sz}$ term, by $R^{-1/2}$ since each element in the term is a sum over $R$ random variables, where $R$ is the rank of $\boldsymbol{S}$. In the more detailed notation of this section, the proposed scaling implies the forward pass in a linear layer is given by

$$[\boldsymbol{h}_t^{(l)}]_i = [\boldsymbol{W}_t]_{:,i} \boldsymbol{h}_t^{(l-1)} = \left([\boldsymbol{\mu}_t^{(l)}]_I + R^{-1/2}[\boldsymbol{S}_t^{(l)}]_I \boldsymbol{z}^{(l)}\right) \boldsymbol{h}_t^{(l-1)}. \tag{S117}$$

In practice, rather than scaling $[\boldsymbol{S}_t^{(l)}]_I \boldsymbol{z}^{(l)}$ by $R^{-1/2}$ in the forward pass, we apply Lemma J.1 from Yang et al. [41] to instead scale the initialization by $R^{-1/2}$ and, in SGD, the learning rate by $R^{-1}$. Scaling by the rank allows treating the mean and covariance parameters as if they were weights parameterized by $\mu$P in a non-probabilistic network, inheriting any scaling that has already been derived for that architecture.

From Table 3 of Yang et al. [41], we therefore scale the mean parameters as

$$b^{(l)} = \begin{cases} 0 & l = 1 \\ 1/2 & l \in \{2, \ldots, L\} \\ 1 & l = L + 1 \end{cases} \quad \text{and} \quad c^{(l)} = \begin{cases} -1 & l = 1 \\ 0 & l \in \{2, \ldots, L\} \\ 1 & l = L + 1. \end{cases} \tag{S118}$$

Assuming $R^{(l)} = (D^{(l-1)}D^{(l)})^{p^{(l)}}$ as before, where $p^{(l)} \in [0, 1]$, we the scale the covariance parameters as

$$\tilde{b}^{(l)} = b^{(l)} + \begin{cases} \frac{p^{(l)}}{2} & l = 1 \\ p^{(l)} & l \in \{2, \ldots, L\} \\ \frac{p^{(l)}}{2} & l = L + 1 \end{cases} \quad \text{and} \quad \tilde{c}^{(l)} = c^{(l)} + \begin{cases} p^{(l)} & l = 1 \\ 2p^{(l)} & l \in \{2, \ldots, L\} \\ p^{(l)} & l = L + 1. \end{cases} \tag{S119}$$

By comparing to Equations S115 and S116, we see the mean and covariance parameters in all but the output layer are initialized as large as possible while still maintaining stability. The output layer parameters scale to zero faster, since, as in $\mu$P for the weights of non-probabilistic networks, we set $b^{(L+1)}$ to 1 instead of $1/2$.

Note that in Section S2.2 we did not consider input and output dimensions that scaled with the width $D$ for simplicity. For our experiments, we take the exact $\mu$P initialization and learning rate scaling from Yang et al. [41] — which includes, for example, a $1/\texttt{fan\_in}$ scaling in the input layer — for the means and then make the rank adjustment for the covariance parameters as described above.

We investigate the proposed scaling in Figures S4 and S5. We train two-hidden-layer ($L = 2$) MLPs of hidden sizes 8, 16, 32, and 64 on a single observation $(x, y) = (1, 1)$ using a squared error loss. We use SGD with a learning rate of 0.05. For the variational networks, we assume a multivariate Gaussian variational family with a full rank covariance.

Figures S3 and S4 show the RMSE of the change in the hidden units from initialization, $\Delta \boldsymbol{g}_t^{(l)}(x) = \boldsymbol{g}_t^{(l)}(x) - \boldsymbol{g}_0^{(l)}(x)$, as a function of the hidden size. The RMSE of the hidden units *at* initialization,

$g_0^{(l)}$ is also shown in blue. Each panel corresponds to a layer of the network, so the first two panels correspond to features $g_t^{(1)}(x)$ and $g_t^{(2)}(x)$, respectively, while the third panel corresponds to the output of the network, $g_t^{(3)}(x) = f_t(x)$. The difference between the figures is the paramaterization. Figure S3 uses standard parameterization (SP) while Figure S4 uses maximal update parametrization ($\mu$P). We observe that (a) the features change more under $\mu$P than SP and (b) training is more stable across hidden sizes under $\mu$P than SP, especially for smaller networks.

Figures S5 and S6 show the analogous results for a variational network. The top row shows the change in the mean of the hidden units, while the bottom row shows the change in the standard deviation. As in the non-probabilistic case, we observe that (a) both the mean and standard deviation of the features change more under $\mu$P than SP and (b) training is more stable across hidden sizes under $\mu$P than SP, especially for smaller networks.

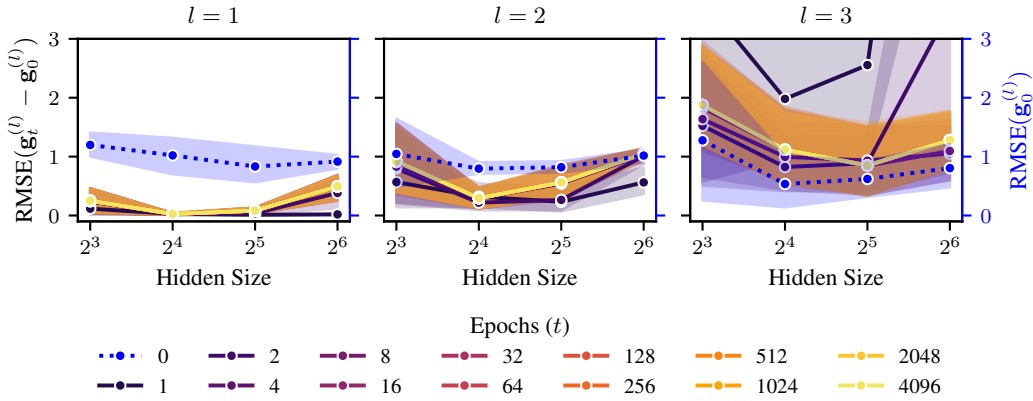

Figure S3: *MLP, Standard Parameterization.* RMSE of the change in the hidden units and, in blue, their initial values. Shaded region represents 95% confidence interval over 5 random initializations. The MLP is trained under SP.

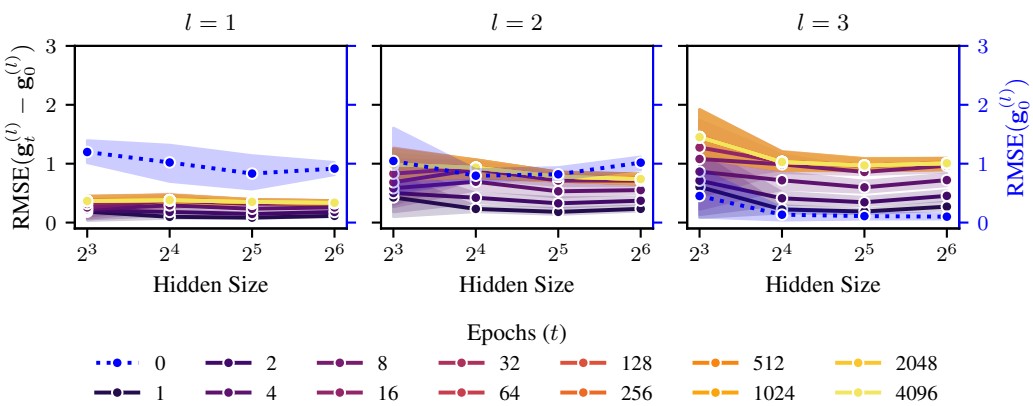

Figure S4: *MLP, Maximal Update Parameterization.* RMSE of the change in the hidden units and, in blue, their initial values. Shaded region represents 95% confidence interval over 5 random initializations. The MLP is trained under $\mu$P.

## S2.4 DETAILS ON HYPERPARAMETER TRANSFER EXPERIMENT

As discussed in Section 3.4 we train two-hidden-layer MLPs of width 128, 256, 512, 1024, and 2048 on CIFAR-10. For comparability to Figure 3 in Tensor Programs V [41] we use the same hyperpa-

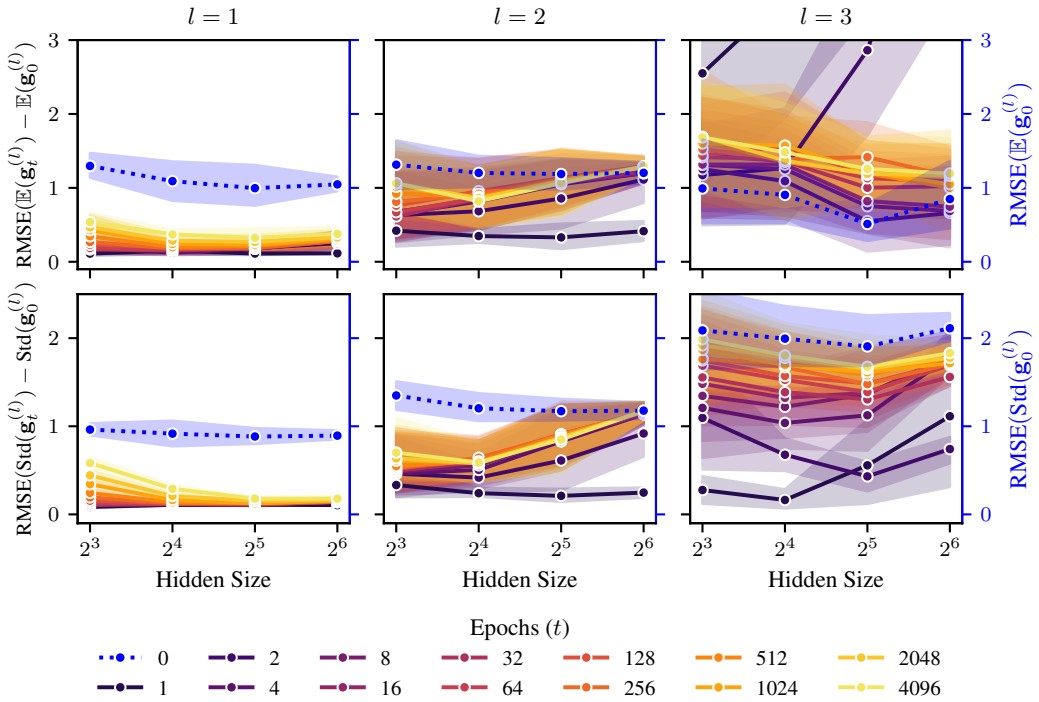

Figure S5: *Variational MLP, Standard Parameterization.* RMSE of the change in the hidden units and, in blue, their initial values. Shaded region represents 95% confidence interval over 5 random initializations. The variational MLP is trained under SP with a full rank covariance in each layer.

rameters but applied to the mean parameters.[5] For the input layer, we scale the mean parameters at initialization by a factor of 16 and in the forward pass by a factor of 1/16. For the output layer, we scale the mean parameters by 0.0 at initialization and by 32.0 in the forward pass. We use 20 epochs, batch size 64, and a grid of global learning rates ranging from $2^{-8}$ to $2^0$ with cosine annealing during training. For the grid search results shown in the right panel of Figure 3, we use validation NLL for model selection and then evaluate the relative test error compared to the best performing model for that width across parameterizations and learning rates.

## S3    EXPERIMENTS

This section outlines in more detail the experimental setup, including datasets (Section S3.1.1), metrics (Section S3.1.2), architectures, the training setup and method details (Section S3.3.1). It also contains additional experiments to the ones in the main paper (Sections S3.2, S3.3.2 and S3.3.3).

### S3.1    SETUP AND DETAILS

In all of our experiments we used the following datasets and metrics.

---

[5]Specifically, we used the hyperparameters as indicated here: https://github.com/microsoft/mup/blob/main/examples/MLP/demo.ipynb

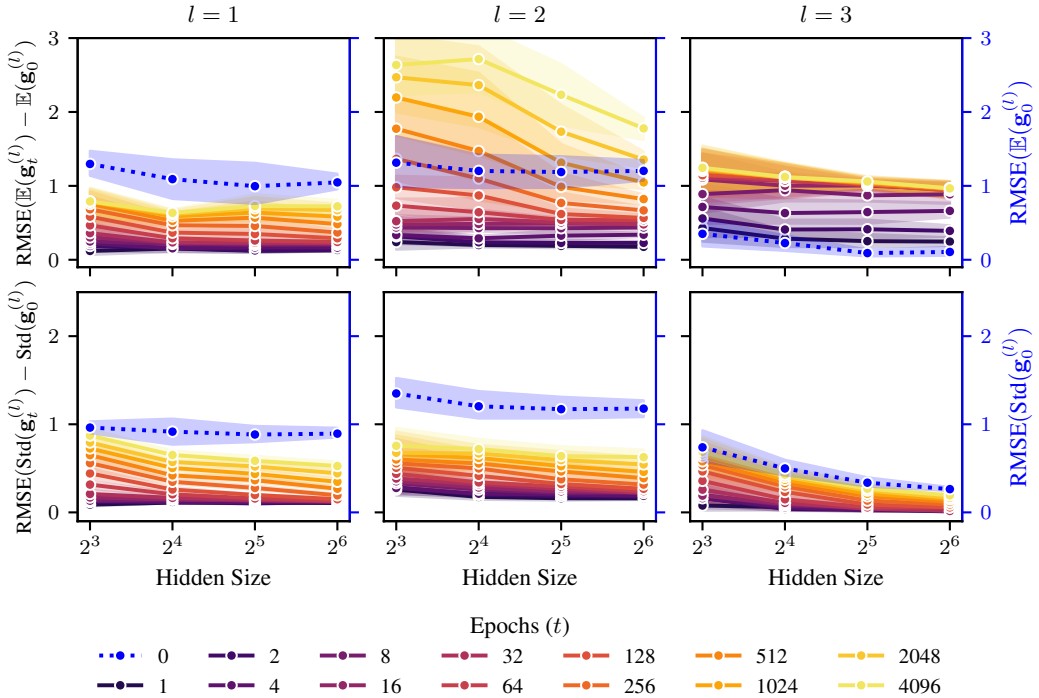

Figure S6: *Variational MLP, Maximal Update Parametrization.* RMSE of the change in the hidden units and, in blue, their initial values. Shaded region represents 95% confidence interval over 5 random initializations. The variational MLP is trained under $\mu$P with a full rank covariance in each layer.

### S3.1.1 DATASETS

Table S1: *Benchmark datasets used in our experiments.* All corrupted datasets are only intended for evaluation and thus only have test sets consisting of 15 different corruptions of the original test set.

| Dataset | $N$ | $N_{\text{test}}$ | $D_{\text{in}}$ | $C$ | Train / Validation Split |
|---|---|---|---|---|---|
| MNIST [70] | 60 000 | 10 000 | $28 \times 28$ | 10 | (0.9, 0.1) |
| CIFAR-10 [80] | 50 000 | 10 000 | $3 \times 32 \times 32$ | 10 | (0.9, 0.1) |
| CIFAR-100 [80] | 50 000 | 10 000 | $3 \times 32 \times 32$ | 100 | (0.9, 0.1) |
| TinyImageNet [81] | 100 000 | 10 000 | $3 \times 64 \times 64$ | 200 | (0.9, 0.1) |
| MNIST-C [72] | - | 150 000 | $28 \times 28$ | 10 | - |
| CIFAR-10-C [73] | - | 150 000 | $3 \times 32 \times 32$ | 10 | - |
| CIFAR-100-C [73] | - | 150 000 | $3 \times 32 \times 32$ | 100 | - |
| TinyImageNet-C [73] | - | 150 000 | $3 \times 64 \times 64$ | 200 | - |

### S3.1.2 METRICS

**Accuracy** The (top-k) accuracy is defined as

$$\text{Accuracy}_k(\boldsymbol{y}, \hat{\boldsymbol{y}}) = \frac{1}{N_{\text{test}}} \sum_{n=1}^{N_{\text{test}}} 1_{(y_n \in \hat{y}_n^{1:k})}. \tag{S120}$$

**Negative Log-Likelihood (NLL)** The (normalized) negative log likelihood for classification is given by

$$\text{NLL}(\boldsymbol{y}, \hat{\boldsymbol{y}}) = -\frac{1}{N_{\text{test}}} \sum_{n=1}^{N_{\text{test}}} \log \hat{\boldsymbol{p}}_{\hat{y}_n}, \tag{S121}$$

where $\hat{\boldsymbol{p}}_{\hat{y}_n}$ is the probability a model assigns to the predicted class $\hat{y}_n$.

**Expected Calibration Error (ECE)**   The expected calibration error measures how well a model is calibrated, i.e. how closely the predicted class probability matches the accuracy of the model. Assume the predicted probabilities of the model on the test set are binned into a given binning of the unit interval. Compute the accuracy $a_j$ and average predicted probability $\hat{p}_j$ of each bin, then the expected calibration error is given by

$$\text{ECE} = \sum_{j=1}^{J} b_j |a_j - \hat{p}_j|, \tag{S122}$$

where $b_j$ is the fraction of datapoints in bin $j \in \{1, \ldots, J\}$.

### S3.2   Time and Memory-Efficient Training

To keep the time and memory overhead low during training, we would like to draw as few samples of the parameters as possible to evaluate the training objective $\bar{\ell}(\boldsymbol{\theta})$. Drawing $M$ parameter samples for the loss increases the time and memory overhead of a forward and backward pass $M$ times (disregarding parallelism). Therefore it is paramount for efficiency to use as few parameter samples as possible, ideally $M = 1$.

When drawing fewer samples from the variational distribution, the variance in the training loss and gradients increases. In practice this means one has to potentially choose a smaller learning rate to still achieve good performance. This is analogous to the previously observed linear relationship $N_b \propto \eta$ between the optimal batch size $N_b$ and learning rate $\eta$ [e.g., 82–84]. Figure S7 shows this relationship between the number of parameter samples used for training and the learning rate on MNIST for a two-hidden layer MLP of width 128.

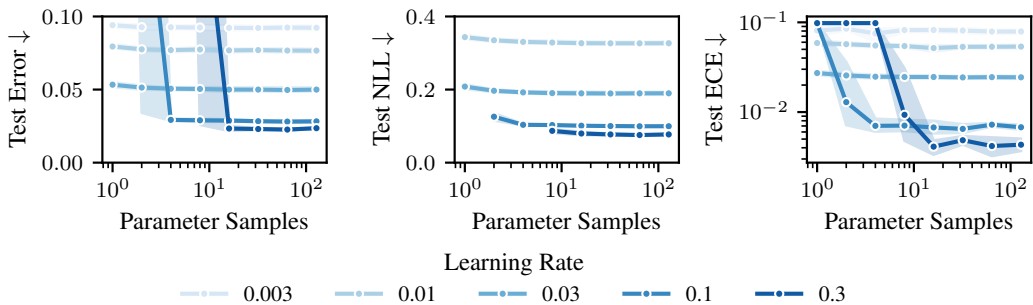

Figure S7: *Generalization versus number of parameter samples.* For a fixed number of epochs and batch size, fewer samples require a smaller learning rate. For a fixed learning rate, generalization performance quickly plateaus with more parameter samples.

As Figure S8 shows, when using momentum, generalization performance tends to increase, but only if either the number of samples is increased, or the learning rate is decreased accordingly. A similar relationship between noise in the objective and the use of momentum has previously been observed by Smith and Le [83], which propose and empirically verify a scaling law for the optimal batch size $N_b \propto \frac{\eta}{1-\gamma}$ as a function of the momentum parameter $\gamma > 0$.

### S3.3   In- and Out-of-distribution Generalization

This section recounts details of the methods we benchmark in Section 5, how they are trained and additional experimental results.

#### S3.3.1   Architectures, Training, and Methods

**Architectures**   We use convolutional architectures for all experiments in Section 5. For MNIST, we use a standard LeNet-5 [70] with ReLU activations. For CIFAR-10, CIFAR-100 and TinyIma-

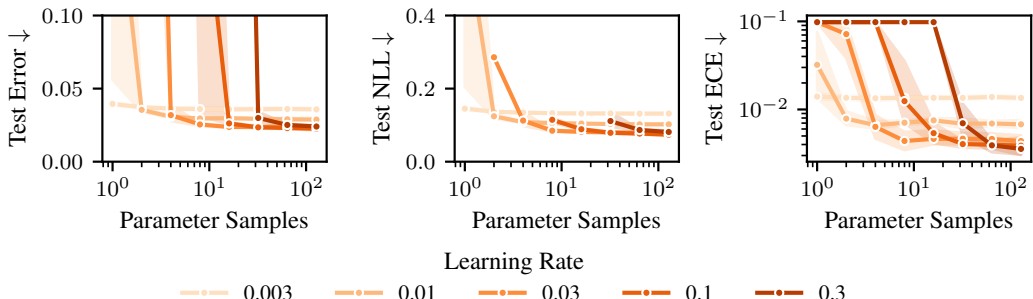

Figure S8: *Generalization versus number of parameter samples when using momentum.* Using momentum improves generalization performance, but when using fewer parameter samples, a smaller learning rate is necessary than for vanilla SGD as predicted by **??**.

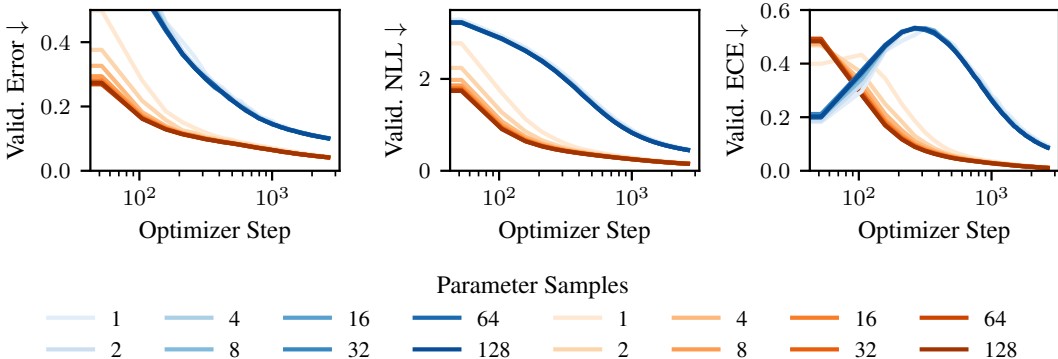

Figure S9: *Validation error during training for different numbers of parameter samples.* The difference in generalization error between different number of parameter samples vanishes with more optimization steps both for SGD (—) and when using momentum (—), *if* the learning rate is sufficiently small (in this example $\eta = 0.003$).

geNet we use a ResNet-34 [71] where the first layer is a 2D convolution with `kernel_size=3`, `stride=1` and `padding=1` to account for the image resolution of CIFAR and TinyImageNet and the normalization layers are `GroupNorm` layers. We use pretrained weights from ImageNet for all but the first and last layer of the ResNets from `torchvision` [85] and fully finetune all parameters during training.

**Training** We train all models using SGD with momentum ($\gamma = 0.9$) with batch size $N_b = 128$ and learning rate $\eta = 0.005$ for 200 epochs. We do not use a learning rate scheduler since we found that neither cosine annealing nor learning rate warm-up improved the results.

**Temperature Scaling [69]** For temperature scaling we optimize the scalar temperature parameter in the last layer on the validation set via the L-BFGS implementation in `torch` with an initial learning rate $\eta_{\text{TS}} = 0.1$, a maximum number of 100 iterations per optimization step and `history_size=100`.

**Laplace Approximation (Last-Layer, GS + ML) [26]** As recommended by Daxberger et al. [26] we use a post-hoc KFAC last-layer Laplace approximation with a GGN approximation to the Hessian. We tune the hyperparameters post-hoc using type-II maximum likelihood (ML). As an alternative we also do a grid search (GS) for the prior scale, which we found to be somewhat more robust in our experiments. Finally, we compute the predictive using an (extended) probit approximation. Our implementation of the Laplace approximation is a thin wrapper of `laplace` [26] and we use its default hyperparameters throughout.

**Weight-space VI (Mean-field) [30, 31]**   For variational inference, we used a mean-field variational family and trained via an ELBO objective with a weighting of the Kullback-Leibler regularization term to the prior. We chose a unit-variance Gaussian prior with mean that was set to the pretrained weights, except for the in- and output layer which had zero mean. We found that using a KL weight and more than a single sample (here $M = 8$) was necessary to achieve competitive performance. The KL weight was chosen to be inversely proportional to the number of parameters of the model, for which we observed better performance than a KL weight that was independent of the architecture. At test time we compute the predictive by averaging logits using 32 samples.

**Implicit Bias VI [ours]**   For all architectures in Section 5 we use a Gaussian in- and output layer with a low-rank covariance ($R = 10, 20$). We train with a single parameter sample $M = 1$ throughout and do temperature scaling at the end of training on the validation set with the same settings as when just performing temperature scaling. We do temperature scaling in classification due to the specific form of the implicit bias in classification as described in Section S1.3. Since IBVI trains by optimizing a minibatch approximation of the expected negative log-likelihood (an average over log-probabilities with respect to parameter samples), we also average log-probabilities at test-time to compute the predictive distribution over class probabilities. Although we did not see a significant difference between averaging log-probabilities, probabilities or logits. Like for WSVI we use 32 samples at test time.

**SWAG   [28]**   We used a slightly modified implementation of SWAG based on `torch-uncertainty` and the original implementation by Maddox et al. [28]. The beginning of the averaging cycle set to half the number of total epochs and a cycle length of one, i.e. SWAG updates happen every epoch. For all other hyperparameters we use the default settings.

**Deep Ensembles [29]**   We use five ensemble members initialized and trained independently. We compute the predictive by averaging the predicted probabilities of the ensemble members in line with standard practice [29]. We did not see a significant difference in performance between averaging logits or averaging class probabilities.

S3.3.2   IN-DISTRIBUTION GENERALIZATION AND UNCERTAINTY QUANTIFICATION

The full results from the in-distribution generalization experiment in Section 5 can be found in Figure S10. The same experiment but done in the Maximal Update parametrization is depicted in Figure S11. When finetuning a pretrained model, we found that on some datasets (CIFAR-100, TinyImageNet) $\mu$P resulted in somewhat lower performance, contrary to the results in Section 3.4, where we trained from scratch. This suggests that, when pretraining, there may be a modification to the parametrization that could improve generalization.

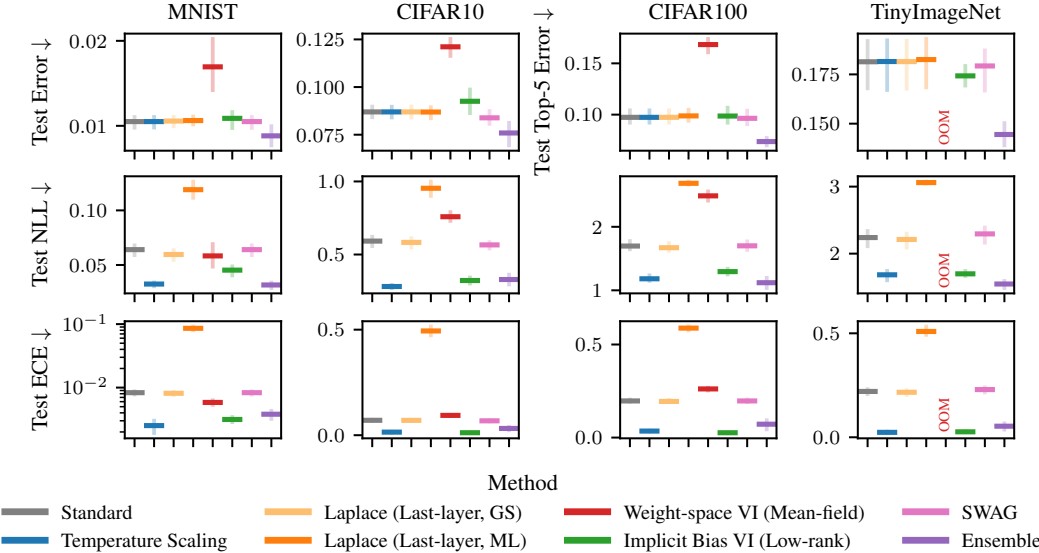

Figure S10: *In-distribution generalization and uncertainty quantification (Standard parametrization).*

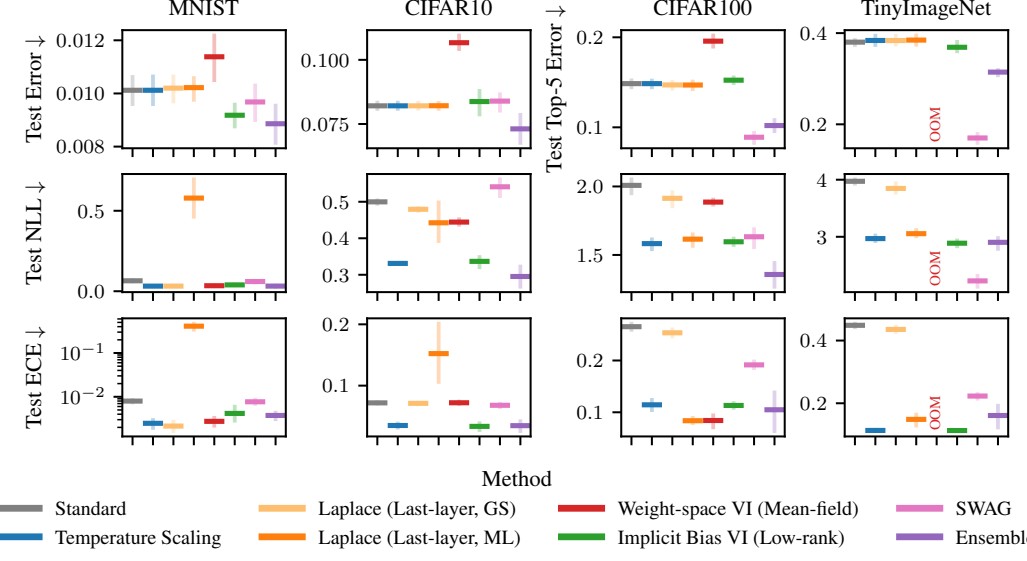

Figure S11: *In-distribution generalization and uncertainty quantification (Maximal Update parametrization).*

### S3.3.3  ROBUSTNESS TO INPUT CORRUPTIONS

Besides the benchmark in Figure S11, we also evaluated the models trained using the Maximal Update parametrization on the corrupted datasets. The results can be found in Figure S12.

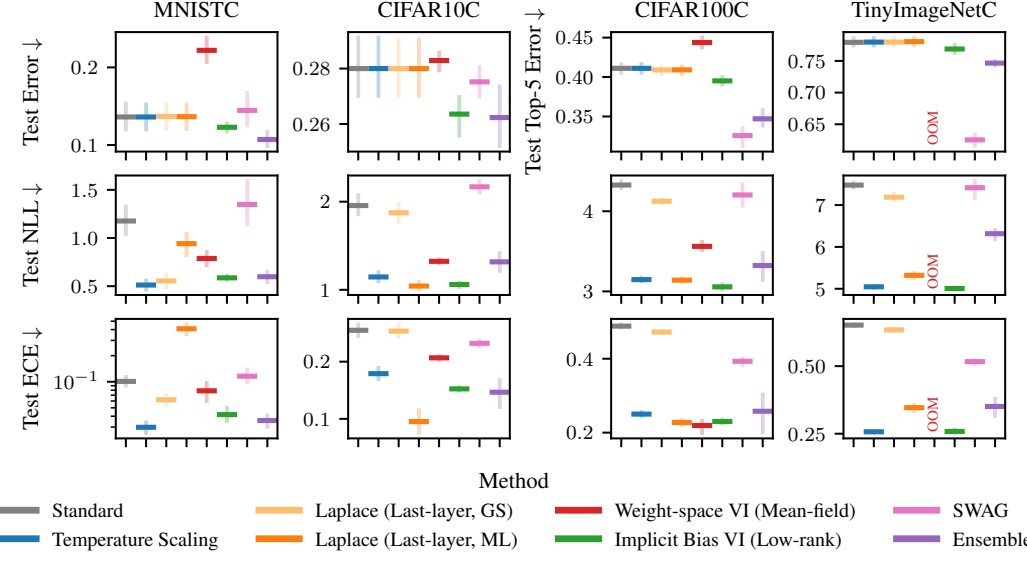

Figure S12: *Generalization on robustness benchmark problems (Maximal Update parametrization).*

### S3.3.4  COMPARISON TO GENERALIZED VI WITH 2-WASSERSTEIN REGULARIZATION

Theorems 1 and 2 characterize the implicit bias of gradient descent for an overparametrized linear model as a preference for distributions minimizing the expected loss, which are closest in 2-Wasserstein distance to the initialization. Given this characterization, by the KKT conditions there exists a Lagrange multiplier $\lambda \geq 0$ such that the optimal variational parameters $\boldsymbol{\theta}_\star^{\mathrm{GD}}$ define a stationary point of the following unconstrained optimization objective:

$$\bar{\ell}_r(\boldsymbol{\theta}) = \bar{\ell}(\boldsymbol{\theta}) + \lambda \, \mathrm{W}_2^2(q_{\boldsymbol{\theta}}, p) \,. \tag{S123}$$

In other words, Implicit Bias VI is equivalent to Generalized VI (GVI) with a 2-Wasserstein regularizer and some regularization strength $\lambda \geq 0$ for overparametrized *linear* models.

**Experiment Results**   To understand the difference in performance between IBVI and Generalized VI with a 2-Wasserstein regularizer for *deep neural networks*, we trained models via the GVI objective in Equation (S123) for different regularization strengths $\lambda \geq 0$ with the same setup as in Section 5. The results on in-distribution test data can be found in Figure S13 and the results for corrupted test data are in Figure S14. Both on in- and out-of-distribution data GVI performs similar or worse than IBVI for all regularization strengths we tested in terms of test error. IBVI and GVI perform roughly similar in terms of uncertainty quantification with GVI only performing better for regularization strengths that harm accuracy.

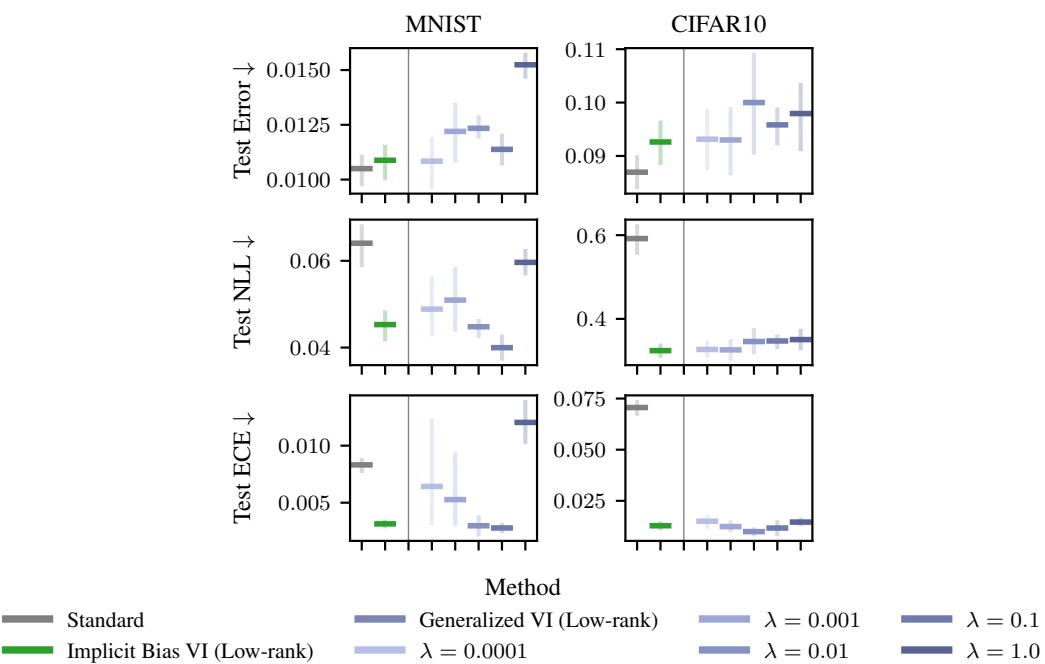

Figure S13: *In-distribution generalization and uncertainty quantification of IBVI and GVI.*

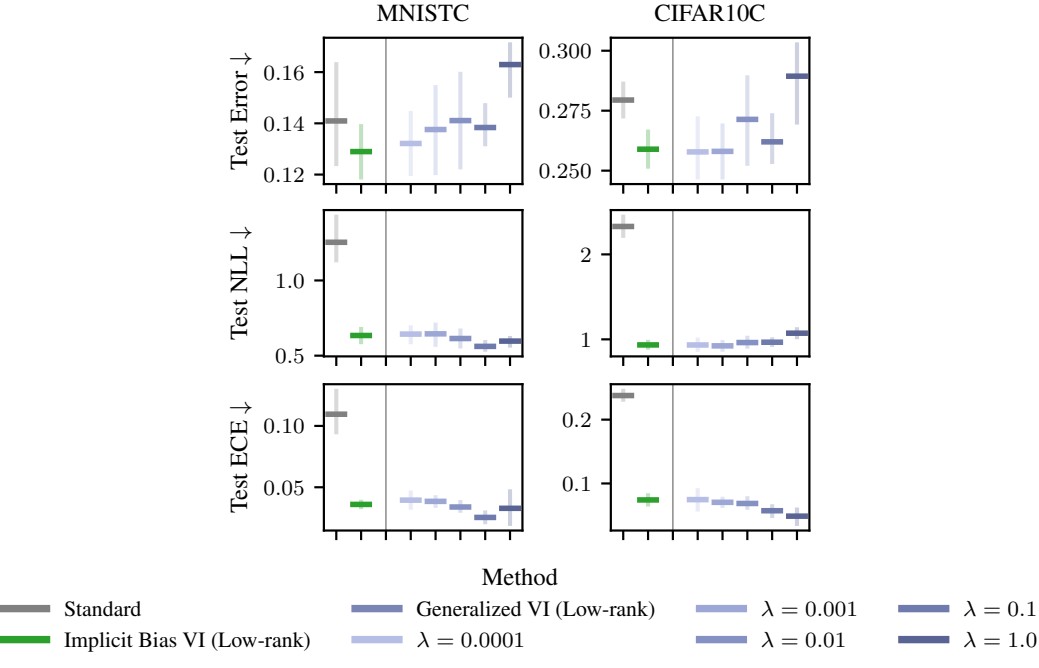

Figure S14: *Out-of-distribution generalization and uncertainty quantification of IBVI and GVI.*

