# OpenReview forum: "Variational Deep Learning via Implicit Regularization"
_ICLR.cc/2026/Conference — ICLR 2026 Poster_

### Official Review · Reviewer_uPWm · 2025-10-18

**Soundness:** 3
**Presentation:** 4
**Contribution:** 4
**Rating:** 6
**Confidence:** 5

**Summary:**

This paper proposes a method to get the benefits of Bayesian deep learning—like improved robustness and uncertainty estimates—without the usual computational cost.

Instead of adding an explicit regularization term (like KL divergence) to the loss function, they rely on the implicit regularization of SGD. They initialize a variational distribution over the network's weights to match a prior, and then train by minimizing only the data-fitting loss.

The key insight is that SGD naturally finds the solution that is closest to its starting point. This behavior acts as a powerful regularizer, achieving strong performance with minimal overhead. They provide proof that this technique actually yields the divergence minimiser for Wasserstein distance in the linear case for overparametrised models.

They provide empirical results showcasing that their proposed methodology offers competitive performance on a broad range of datasets.

**Strengths:**

1. Very sound approach and presentation, very much like the simplicity of just not having an extra divergence term to deal with the numerical issues of.
2. Theorem 1 is very nice, gives an intuition as to why this could work for much larger models.
3. Experiments convincing, comprehensive across diverse tasks and datasets

**Weaknesses:**

1. This method seems to be possibly limited to mean field variational posteriors, and only considers the Wasserstein distance as the divergence term.
2. Missing baseline comparison, would be nice if you could compare against a variational method that uses Wasserstein distance explicitly in the "ELBO", possibly https://proceedings.neurips.cc/paper_files/paper/2022/file/18210aa6209b9adfc97b8c17c3741d95-Paper-Conference.pdf or any other paper that proposes Wasserstein distance explicitly as a divergence term.

**Questions:**

1. Can it possibly be generalised to other family of variational posteriors and not just Wasserstein distance for the divergence term?
2. Did you try any other parameterisations of the mean field covariance term?
3. Would appreciate an ablation study showing that this offers similar performance compared to using an explicit Wasserstein distance divergence term.

---

> ### Author Response · Authors · 2025-11-20
> **Author Response**
>
> Thank you for reviewing our paper and the time you invested in providing feedback! We answered your questions below. If anything remains unclear, we would be happy to clarify during the discussion period. We also ran the additional experiment you requested which can be found in Section S3.3.4 of the updated paper. If this rebuttal has changed your initial assessment of our work, we would appreciate if you would consider updating your score. Thank you!
>
> ### Response to Questions
> > This method seems to be possibly limited to mean field variational posteriors, and only considers the Wasserstein distance as the divergence term.
> >
> > Q: Can it possibly be generalised to other family of variational posteriors and not just Wasserstein distance for the divergence term?
>
> **Variational Family**:
> Our proposed approach of learning a variational distribution by optimizing purely the expected loss without a divergence term can be in principle used for any variational family. However, the exact characterization of the implicit regularization for overparametrized linear models in Theorems 1 and 2 only holds for a multivariate Gaussian variational family with a factorized covariance - **not** a mean-field covariance.
> In our experiments we use a variational family consisting of a Gaussian first and last layer with low-rank covariances.
> It would be interesting to investigate in future work whether a different variational family with stochastic intermediate layers would perform even better in practice.
>
> **Implicit Regularization resulting in 2-Wasserstein regularizer**:
> This is a great question. Using a different optimizer or optimization hyperparameters generally results in different implicit regularization. For example, Gunasekar et al. (2018) [1] show that for standard overparametrized linear models trained with mirror descent with respect to some Mahalanobis distance results in minimum norm regularization to the initialization in the chosen Mahalanobis distance (see Theorem 1). We conjecture that one could extend our results similarly to the corresponding Wasserstein distance induced by the chosen Mahalanobis distance. We focused on (stochastic) gradient descent in this work given its ubiquity as an optimizer in deep learning.
>
> > Q: Did you try any other parameterisations of the mean field covariance term?
>
> To clarify, all instances of our method use a factorized covariance for the Gaussian layers, not a mean-field covariance (i.e., the covariance is factorized, not the Gaussian distribution). We believe there is potential for improvement by, for example, using a mean-field or Kronecker-factorized covariance in hidden layers, paired with a factorized covariance for the output layer as suggested by Theorems 1 and 2. Using a different factorization of the covariance in the output layer would presumably lead to a different regularizer than 2-Wasserstein.
>
> > Q: Would appreciate an ablation study showing that this offers similar performance compared to using an explicit Wasserstein distance divergence term.
>
> We added a **new experiment comparing IBVI to Generalized VI with a Wasserstein regularizer (GVI)** on MNIST and CIFAR-10 to the paper  (see Section S3.3.4 of the supplementary material).
> We trained a LeNet5 and ResNet34 architecture via the GVI loss with a 2-Wasserstein distance using five different regularization strengths $\lambda \in \\{1e0, \dots ,1e-4\\}$ and evaluated its performance on in- and out-of-distribution data in the same way we did in our main experiment in Section 5.
> We found **IBVI performs similar or better than GVI with respect to test error** both in- (see Figure S13) and out-of-distribution (see Figure S14).
> IBVI and GVI are roughly equivalent in terms of uncertainty quantification, with GVI only providing slightly better uncertainty quantification for large regularization strength $\lambda$, which comes at the cost of higher test error.
>
>
> [1] S. Gunasekar, J. Lee, D. Soudry, and N. Srebro. “Characterizing Implicit Bias in Terms of Optimization Geometry”. In: International Conference on Machine Learning (ICML). 2018. URL: https://arxiv.org/abs/1802.08246

---

> > ### Author Response · Authors · 2025-11-20
> > **Re: Reviewer response**
> >
> > Thank you again for your review and the suggestion for the ablation!

---

### Official Review · Reviewer_NjWy · 2025-10-23

**Soundness:** 3
**Presentation:** 3
**Contribution:** 3
**Rating:** 8
**Confidence:** 3

**Summary:**

- this paper exploits the implicit regularization effect of (stochastic) gradient descent (SGD) for Bayesian deep learning.
- In particular, the authors characterize the implicit bias induced by SGD for regression and binary classification task in overparameterized linear models.
- Experiments show competitive performance compared to state-of-the-art Bayesian deep learning baselines.

**Strengths:**

- The paper is well written and structured.
- The authors propose to training a Bayesian neural network using only the expected loss, removing the divergence term from the variational objective. They then show that among the variational parameters (assumed to follow a Gaussian distribution) that minimize the expected loss, SGD converges to the one closest to the prior in terms of Wasserstein-2 distance -- analogous to how SGD converges to minimum-norm solution in overparameterised linear network.
- The experimental results demonstrate that this implicit bias achieves competitive test error and uncertainty quantification across different datasets, thereby empirically confirming the proposed effect.
- Overall, I enjoyed reading this paper.

**Weaknesses:**

- The theoretical analysis is limited to overparameterized linear model (one layer). Extending the proof to more than 2 layers might not be straightforward.
- The proof assumes Gaussian variational distribution, and the theorems do not generalize to other distributions which might be more effective for other problem settings/ datasets.

**Questions:**

See the weaknesses.

**Details Of Ethics Concerns:**

I have no concerns.

---

> ### Author Response · Authors · 2025-11-20
> **Author Response**
>
> Thank you for your time and effort in reviewing our paper! We're glad you enjoyed reading it. We answered the questions you raised in your review below.
>
> ### Response to Questions
>
> > Q: The theoretical analysis is limited to overparameterized linear model (one layer). Extending the proof to more than 2 layers might not be straightforward.
>
> We agree that extending the theoretical results to multiple layers (and to non-linear models) is not straightforward. Even in the non-probabilistic setting, only recently has there been progress in characterizing the implicit bias of two-layer ReLU networks [1].
> An angle for analyzing deeper networks could be through the Edge of Stability (EoS) framework. In the non-probabilistic case, [2] analyzes the stability of the minima found by implicit bias and the regularity of the corresponding functions. Furthermore, [3] show that by adding noise to the gradients, the optimization procedure converges to flatter minima, resulting in more regular functions. In our case, since we draw samples throughout training, the gradients are inherently noisy; however, we also optimize the noise levels (through training the covariance of the weights), in contrast to the fixed injected noise considered in [2]. Hence, we believe our framework lies somewhere in between these two regimes.
> We view this as an interesting direction for future research.
>
> > Q: The proof assumes Gaussian variational distribution, and the theorems do not generalize to other distributions which might be more effective for other problem settings/ datasets.
>
> Correct; our theoretical results currently assume a Gaussian variational distribution. Extending them to more general families is not straightforward. However, Gaussians are surprisingly expressive in Bayesian deep learning contexts. Recent works (e.g. [4]) show that one can approximate complex, multi-modal predictive distributions when using even simple Gaussian distributions in hidden layers.
>
> [1] Kou, Y., Chen, Z., & Gu, Q. (2023). Implicit Bias of Gradient Descent for Two-layer ReLU and Leaky ReLU Networks on Nearly-orthogonal Data. In Advances in Neural Information Processing Systems (NeurIPS 2023). URL: https://arxiv.org/abs/2310.18935
>
> [2] Mulayoff, R., Michaeli, T., & Soudry, D. (2021). The Implicit Bias of Minima Stability: A View from Function Space. In Advances in Neural Information Processing Systems (NeurIPS 2021). URL: https://proceedings.neurips.cc/paper_files/paper/2021/file/944a5ae3483ed5c1e10bbccb7942a279-Paper.pdf
>
> [3] Ghosh, A., Cong, B., Yokota, R., Ravishankar, S., Wang, R., Tao, M., Khan, M. E., & Möllenhoff, T. (2025). Variational Learning Finds Flatter Solutions at the Edge of Stability. In Advances in Neural Information Processing Systems (NeurIPS 2025). URL: https://arxiv.org/abs/2506.12903
>
> [4] Sharma, M., Farquhar, S., Nalisnick, E., & Rainforth, T. (2023). Do Bayesian neural networks need to be fully stochastic? In Proceedings of the 26th International Conference on Artificial Intelligence and Statistics (AISTATS 2023). URL: https://arxiv.org/abs/2211.06291

---

> > ### Comment · Reviewer_NjWy · 2025-11-27
> >
> > I thank the authors for the clarification. I would like to keep the score unchanged.

---

### Official Review · Reviewer_QjSV · 2025-10-31

**Soundness:** 3
**Presentation:** 3
**Contribution:** 3
**Rating:** 6
**Confidence:** 4

**Summary:**

he authors propose and analyze a variant of the so-called “generalized variational inference” objective (e.g., Eq. 1 of Knoblauch et al.) that drops that regularization term, arguing that the problem is already well-posed from implicit rather than explicit regularization. The well-posedness derives from the fact that in overparameterized models, the set of global optima is quite large (not just a point mass), and so SGD tends to converge to the member of the set of global optima which is nearest to the initialization. Because of this, the problem is implicitly regularized to the prior already (b/c the prior is used for initialization), so explicit regularization is unnecessary. The authors formalize this result for linear regression and binary classification by showing the implicit regularization takes the form of a Wasserstein distance from the the variational distribution to the prior.

**Strengths:**

- The ideas of the paper are central, yet deep, and well articulated by the authors
- To my knowledge, the contributions are novel in the literature and interestingly meld together the ideas of Bayesian ensembles of models with variational inference and optimization. I find the takeaway message of the paper to be quite “unifying”.
- Extensibility to two different problem settings (regression and classification) augments the impact of the work.
- The authors provide sufficient background to make the work accessible to a broader audience

**Weaknesses:**

- The selling point of the paper falls short of the results: deep learning is not included in the main results of S4, which assume overparameterized linear models. I still find the contribution interesting even in this context, but I’m unsure how much the results can be said to explain the success of “deep learning”. I understand the discussion on page 5 and the CIFAR experiment tie it more closely to practical deep learning, but the theoretical results do not explain the success of deep learning in the same way that the results of, for example, the neural tangent kernel (NTK) does. The lack of a clear connection to deep learning undermines the contribution somewhat.
- The main experimental results (Fig. 4) do not demonstrate clear outperformance of the method relative to the competitors. However, I commend the breadth of the competing approaches considered.

**Questions:**

- It seems (cf. line 45 and line 2228) that the loss function used by the authors is the negative log likelihood. Considering that the proposed method only minimizes this, and nothing else, it’s surprising that other approaches can outperform on this exact metrics (Test NLL for MNISTC, for example, in Figure 5). Can the authors provide any insight on this phenomenon?
- Taking implicit regularization as well-established, can the authors comment more on the desirability of the solutions found by this fact? Is the solution closest to the prior advantageous with respect to generalizability, etc.?
- The authors may consider citing other works that use the negative log likelihood loss, perhaps to comment on differences. For example, the class of neural posterior estimation (NPE) methods comes to mind. Can you elaborate on similarities or differences to this class of methods?

---

> ### Author Response · Authors · 2025-11-19
> **Author Response**
>
> Thank you for reviewing our paper and giving detailed feedback! We responded to the questions you raised below.
>
> ### Response to Questions
>
> > Q: It seems (cf. line 45 and line 2228) that the loss function used by the authors is the negative log likelihood. Considering that the proposed method only minimizes this, and nothing else, it’s surprising that other approaches can outperform on this exact metrics (Test NLL for MNISTC, for example, in Figure 5). Can the authors provide any insight on this phenomenon?
>
> As you point out, our method trains solely via the (expected) negative log likelihood (i.e., there is no explicit regularization). Many other baselines we compare against use the negative log likelihood as a loss function in addition to explicit regularization.
> On in-distribution test data (Figure S10) only Temperature Scaling and Ensembles achieve slightly better Test NLL than our approach, both baselines which are also trained solely via the negative log likelihood.
> On *corrupted* data (e.g. MNISTC as you correctly point out) other methods can achieve lower Test NLL, because the test distribution $p(y, x)$ has changed and therefore additional inductive bias from an explicit regularizer can be helpful, if well-chosen.
>
> > Q: Taking implicit regularization as well-established, can the authors comment more on the desirability of the solutions found by this fact? Is the solution closest to the prior advantageous with respect to generalizability, etc.?
>
> There are works showing the value of a good function-space prior (e.g., [1,2]). However, these methods can be expensive to train and it is not always easy to elicit a helpful prior that does not worsen in-distribution performance. Our approach, which inherits the prior from the initialization and training procedure, does not hurt in-distribution performance and works well in experiments without extensive prior elicitation.
>
> [1] Ba-Hien Tran, Simone Rossi, Dimitrios Milios, Maurizio Filippone. *All You Need is a Good Functional Prior for Bayesian Deep Learning*. JMLR, 2022. URL: https://arxiv.org/abs/2011.12829
>
> [2] Tim G. J. Rudner, Sanyam Kapoor, Shikai Qiu, Andrew Gordon Wilson. *Function-Space Regularization in Neural Networks: A Probabilistic Perspective*. ICML, 2023. URL: https://arxiv.org/abs/2312.17162
>
> > Q: The authors may consider citing other works that use the negative log likelihood loss, perhaps to comment on differences. For example, the class of neural posterior estimation (NPE) methods comes to mind. Can you elaborate on similarities or differences to this class of methods?
>
> NPE is mostly used when the likelihood is intractable (simulation-based inference). In our setting, the likelihood is assumed to be tractable and directly used as the loss function. Note that in NPE the NLL refers to the reverse cross entropy, i.e. the expectation is of the variational distribution with respect to samples from the true joint, whereas in our setting it's the opposite. It is also worth mentioning that NPE typically scales prohibitively with the dimension of the parameter space, which means in practice it is typically used in at most hundreds of dimensions (e.g., [3] considers a Fashion MNIST example with a 784-dimensional parameter). To apply NPE for variational deep learning it would have to scale to millions of parameters.
>
> [3] Marvin Schmitt, Valentin Pratz, Ullrich Kothe, Paul-Christian Burkner, Stefan T. Radev. *Consistency Models for Scalable and Fast Simulation-Based Inference*. NeurIPS, 2024. URL: https://arxiv.org/abs/2312.05440

---

> > ### Comment · Reviewer_QjSV · 2025-11-26
> > **Thanks**
> >
> > Thanks to the authors for the response, I've increased my score to reflect their answers to my questions.

---

### Meta-Review · Area_Chair_91x9 · 2026-01-06

**Summary:**

The reviewers’ main concerns centered on the limited theoretical scope. This is because the formal analysis is restricted to overparameterized linear models and Gaussian variational families, and on the strength of the connection to deep learning practice. In particular, questions were raised about whether the results can fully explain the success of deep learning beyond linear settings, and whether the lack of explicit regularization could be detrimental under distribution shift. At the same time, reviewers consistently agreed that the paper presents a novel, well-motivated approach with clear theoretical insights and competitive empirical performance. The rebuttal clarified empirical observations, strengthened the positioning relative to related work, and addressed missing experimental comparisons. Overall, the remaining concerns primarily point to future extensions rather than shortcomings of the current contribution, supporting a positive recommendation.

**Reviewer Concerns:**

- Clarification of empirical behavior under NLL minimization (Reviewer QjSV):
The rebuttal clearly explained why methods trained solely with the NLL can be outperformed on corrupted or shifted data by explicitly regularized baselines, resolving the apparent contradiction noted in the original review.


- Missing baseline comparison to explicit Wasserstein VI (Reviewer uPWm):
The newly added ablation comparing IBVI to generalized VI with an explicit Wasserstein regularizer directly addressed this concern and strengthened the empirical claims.

- Expressiveness of Gaussian variational families (Reviewer NjWy):
The rebuttal cited recent work showing that even simple Gaussian distributions can yield rich predictive behavior, mitigating concerns about practical limitations.

Outstanding concerns:
Theoretical scope limited to linear / shallow models: All reviewers acknowledged that extending the theory to deep, non-linear networks remains open. However, this was consistently framed as an expected and challenging future direction rather than a deficiency of the current work.

Generality beyond Gaussian variational families: While the approach itself is broadly applicable, a full theoretical characterization for other variational families is not yet available.

These remaining issues do not undermine the correctness or significance of the present contribution and are appropriate avenues for future research.

**Reviewer Scores:**

- Reviewer QjSV:
Unchanged, as the rebuttal clarified the reviewer’s questions and improved confidence in the work, but did not fundamentally alter the reviewer’s view regarding the theoretical scope relative to deep learning.

- Reviewer NjWy:
Unchanged, since the reviewer already had a positive assessment and the rebuttal mainly confirmed their original understanding without introducing substantially new evidence.

- Reviewer uPWm:
Unchanged, as although the rebuttal addressed several technical concerns and added new experiments, the reviewer’s overall cautious stance about generality and scope remained.

---

### Decision · Program_Chairs · 2026-01-26

Accept (Poster)